# Reference-free cell type deconvolution of multi-cellular pixel-resolution spatially resolved transcriptomics data

Brendan F. Miller[1,2], Feiyang Huang [1,2], Lyla Atta [1,2], Arpan Sahoo[1,3] & Jean Fan [1,2,3✉]

Recent technological advancements have enabled spatially resolved transcriptomic profiling but at multi-cellular pixel resolution, thereby hindering the identification of cell-type-specific spatial patterns and gene expression variation. To address this challenge, we develop STdeconvolve as a reference-free approach to deconvolve underlying cell types comprising such multi-cellular pixel resolution spatial transcriptomics (ST) datasets. Using simulated as well as real ST datasets from diverse spatial transcriptomics technologies comprising a variety of spatial resolutions such as Spatial Transcriptomics, 10X Visium, DBiT-seq, and Slide-seq, we show that STdeconvolve can effectively recover cell-type transcriptional profiles and their proportional representation within pixels without reliance on external single-cell transcriptomics references. STdeconvolve provides comparable performance to existing reference-based methods when suitable single-cell references are available, as well as potentially superior performance when suitable single-cell references are not available. STdeconvolve is available as an open-source R software package with the source code available at https://github.com/JEFworks-Lab/STdeconvolve.

[1] Center for Computational Biology, Whiting School of Engineering, Johns Hopkins University, Baltimore, MD 21211, United States. [2] Department of Biomedical Engineering, Johns Hopkins University, Baltimore, MD 21218, United States. [3] Department of Computer Science, Johns Hopkins University, Baltimore, MD 21218, United States. ✉email: jeanfan@jhu.edu

Delineating the spatial organization of transcriptionally distinct cell types within tissues is critical for understanding the cellular basis of tissue function[1]. Recent technologies have enabled spatial transcriptomic (ST) profiling within tissues at multi-cellular pixel-resolution[2]. As such, these ST measurements represent cell mixtures that may comprise multiple cell types. This lack of single-cell resolution hinders the characterization of cell-type-specific spatial organization and gene expression variation.

To address this challenge, several reference-based, supervised deconvolution approaches have recently been developed to estimate the proportion of cell types within ST pixels. Of these, SPOTlight[3] uses cell-type marker genes derived from a single-cell RNA-sequencing (scRNA-seq) reference to seed a non-negative matrix factorization. RCTD[4] uses the cell-type-specific mean expression of marker genes derived from a scRNA-seq reference to build a probabilistic model of the contribution of each cell type to the observed gene counts in each pixel. SpatialDWLS[5] uses cell-type signature genes derived from a scRNA-seq reference to first enrich for cell types likely to be in each pixel, then applies a dampened weighted least squares approach to infer the cell-type composition. As such, these approaches rely on the availability of a suitable single-cell reference, which may present limitations if such a reference does not exist due to budgetary, technical[6], or biological limitations[7]. While the rise of scRNA-seq references through atlasing efforts such as the BRAIN Initiative Cell Census Network[8], the Human BioMolecular Atlas Program, and Human Cell Atlas[9] may help alleviate such limitations particularly for healthy tissues, processing independent tissue samples or different sections of the same tissue may still result in systematically different gene expression quantifications due to batch effects as well as inter- and intra-sample heterogeneity. Additionally, difficulties dissociating and capturing certain cell types via single-cell sequencing may result in missing or inconsistent cell types between scRNA-seq references and ST datasets[10,11]. Further, scRNA-seq references and ST datasets may be affected by different perturbations manifesting as distinct transcriptional differences affecting reference-based deconvolution accuracy and subsequent biological interpretations. As such, a reference-free deconvolution approach provides an alternative strategy for deconvolving cell types when an appropriate reference is not available.

Here, we developed STdeconvolve (available at https://github.com/JEFworks-Lab/STdeconvolve and as Supplementary Software) as a reference-free, unsupervised approach for deconvolving multi-cellular pixel-resolution ST data (Fig. 1). STdeconvolve builds on latent Dirichlet allocation (LDA), a generative statistical model commonly used in natural language processing for discovering latent topics in collections of documents. In the context of natural language processing, given a count matrix of words in documents, LDA infers the distribution of words for each topic and the distribution of topics in each document. In the context of ST data, given a count matrix of gene expression in multi-cellular ST pixels, STdeconvolve applies LDA to infer the putative transcriptional profile for each cell type and the proportional representation of each cell type in each multi-cellular ST pixel ("Methods"). While LDA has previously been applied in the context of deconvolving cell types in bulk RNA-seq data[12,13], STdeconvolve further leverages several unique aspects of ST data in its application of LDA (Supplementary Note 1). Briefly, these unique aspects of ST data include (i) the limited number of cells and cell types represented in each ST pixel, (ii) the limited impact of batch effects on the measured gene expression across pixels, (iii) the large number of pixels compared to cell types, and (iv) the likely heterogeneity of cell-type proportional distribution across pixels in tissues. Leveraging these aspects, STdeconvolve feature selects for genes likely to be informative of latent cell types to improve the application of LDA to ST data. Specifically, STdeconvolve selects for significantly overdispersed genes, or genes with higher-than-expected expression variance across ST pixels[14] ("Methods"). In addition, as the application of LDA requires the number of transcriptionally distinct cell types, $K$, to be set a priori, STdeconvolve provides several data-driven metrics to guide the estimation of an appropriate $K$ ("Methods", Supplementary Note 2).

## Results

### STdeconvolve accurately recovers cell-type proportions and transcriptional profiles in simulated ST data.

As a proof of concept, we first evaluated the performance of STdeconvolve in recovering the proportional representations of cell types and their transcriptional profiles using simulated ST data. We simulated ST data by aggregating the gene expression of cells from single-cell resolution multiplex error-robust fluorescence in situ hybridization (MERFISH) data of the mouse medial preoptic area (MPOA)[15] within spatially contiguous pixels. Previously, MERFISH was previously applied to map the spatial distribution of 135 select genes within MPOA brain tissue. These select 135 genes were chosen to distinguish between major non-neuronal cell types as well as neuronal subtypes. Imaging-based cell segmentation was performed and the counts of genes per cell were quantified to achieve single-cell resolution spatially resolved transcriptomic profiling. Subsequent transcriptional clustering analysis on the single-cell resolution gene expression measurements identified 9 major cell types, including excitatory and inhibitory neurons. Further clustering found that these excitatory and inhibitory neurons could be subdivided into 69 finer neuronal subtypes.

To simulate multi-cellular pixel-resolution ST data, we aggregated the single-cell resolution MERFISH data into 100 μm² pixels (Fig. 2a, Supplementary Fig. S1A, B, "Methods"). Given the already limited 135 gene panel chosen to distinguish between cell types, additional feature selection for this dataset was not necessary (Supplementary Note 3). Applying STdeconvolve, we identified $K = 9$ cell types and deconvolved their proportional representation and transcriptional profiles in each simulated pixel (Fig. 2b, Supplementary Figs S1C, S2A, Supplementary Methods). To infer the identities of the deconvolved cell types for benchmarking purposes, we matched their deconvolved transcriptional profiles with the transcriptional profiles of ground truth cell types by testing for enrichment of ground truth cell-type-specific marker genes (Methods, Supplementary Fig. S2B). We observed strong correlations between the transcriptional profiles of each deconvolved cell-type and matched ground truth cell type across genes (Fig. 2c). Likewise, we observed strong correlations between the proportions of each deconvolved cell type and matched ground truth cell type across simulated pixels (Fig. 2d). We further quantified this performance using the root-mean-square error (RMSE) of the deconvolved cell type proportions compared to ground truth across simulated pixels ("Methods", Fig. 2e). In this manner, STdeconvolve can accurately recover the proportional representation and transcriptional profiles of major cell types.

### STdeconvolve achieves competitive performance to reference-based, supervised deconvolution approaches.

We next sought to compare the performance of STdeconvolve to existing supervised reference-based deconvolution approaches SPOTlight, RCTD, and spatialDWLS using our simulated 100 μm² resolution ST data of the MPOA. As described previously, these approaches require a single-cell transcriptomics reference for deconvolution.

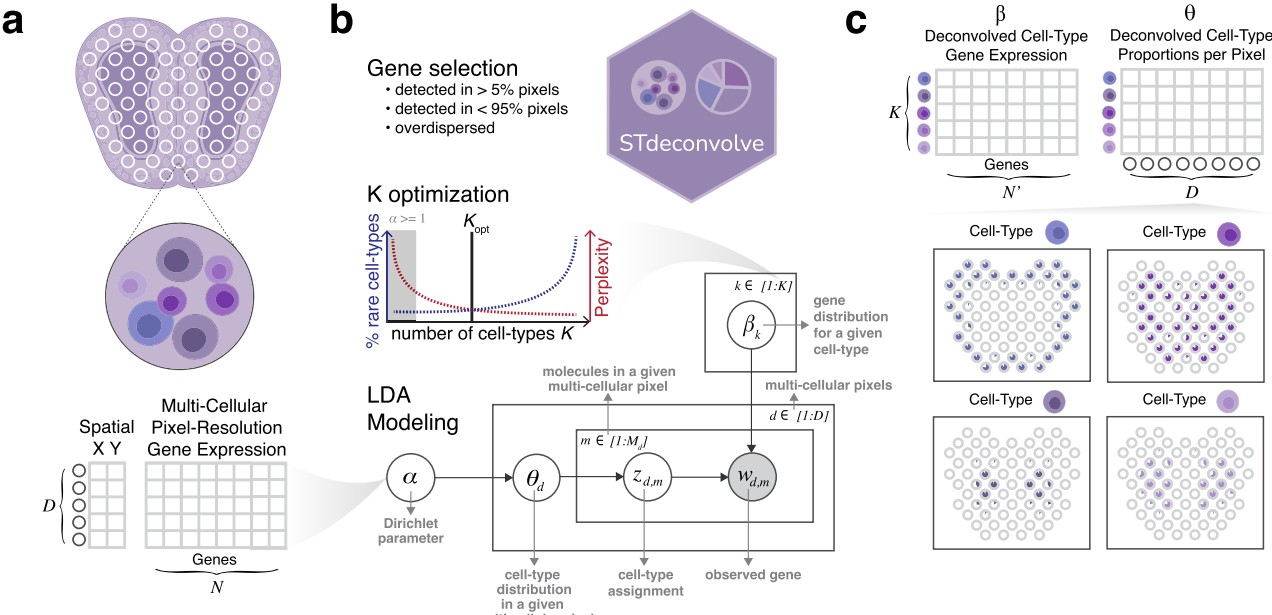

**Fig. 1 Overview of STdeconvolve. a** STdeconvolve takes as input a spatial transcriptomics (ST) gene counts matrix of $D$ pixels (rows) by $N$ genes (columns). A matrix of spatial coordinates for each of the $D$ pixels can also be used for visualization. **b** STdeconvolve first feature selects genes for deconvolution, such as genes with counts in more than 5% and less than 95% of the pixels, and overdispersed across the pixels. STdeconvolve then guides the selection of the optimal number of cell types to be deconvolved, $K$. STdeconvolve finally applies LDA modeling. A graph representation of LDA modeling and the parameters to be learned is shown. Shaded circle indicates observed variables and clear circles indicate latent variables. **c** STdeconvolve outputs two matrices: (1) $\beta$, the deconvolved transcriptional profile matrix of $K$ cell types over $N'$ feature selected genes, and (2) $\theta$, the proportions of $K$ cell types across the $D$ pixels. The proportions of deconvolved cell types can then be visualized across the pixels.

As an ideal single-cell transcriptomics reference, we used the original single-cell MERFISH data that was used to construct the simulated ST data (Supplementary Fig. S3A, Supplementary Methods). We again quantified the performance of each approach using the RMSE of the deconvolved cell-type proportions compared to ground truth across simulated pixels. In general, we find the performance of STdeconvolve to be comparable to these reference-based deconvolution approaches when such an ideal single-cell transcriptomics reference is used (Fig. 2e, f).

One potential limitation of such existing reference-based deconvolution approaches is their reliance on a suitable single-cell transcriptomics reference. We thus sought to evaluate the performance of these reference-based deconvolution approaches when a suitable single-cell reference is not available. To this end, we removed excitatory and inhibitory neuronal cell types to simulate a less suitable single-cell transcriptomics reference (Supplementary Methods). We then deconvolved the simulated ST data of the MPOA using each reference-based deconvolution approach with this new less suitable reference and computed the RMSE across pixels. Because STdeconvolve does not use a reference, its performance does not change. However, the performance for all reference-based deconvolution approaches resulted in a significantly higher RMSE (Diebold–Mariano $p$ value $< 2.2 \times 10^{-16}$) than STdeconvolve (Fig. 2g). Likewise, pixels previously comprised of neurons were now erroneously predicted by reference-based deconvolution approaches to be comprised primarily of immature oligodendrocytes (Supplementary Fig. S3B). In addition, we evaluated the performance of each reference-based deconvolution approach after removing rarer ependymal cells from the single-cell transcriptomics reference. Again, given this less suitable single-cell transcriptomics reference, pixels previously comprised of ependymal cells were now erroneously predicted by reference-based deconvolution approaches to be comprised primarily of astrocytes (Supplementary Fig. S3C). Thus, the performance of reference-based deconvolution approaches is sensitive to differences in cell-type composition between the ST data and the single-cell transcriptomics reference used.

Likewise, such an ideal single-cell transcriptomics reference that optimally matches the cell-type composition and measurement sensitivities of the ST data to be deconvolved may not be available. Therefore, this ideal MERFISH MPOA single-cell transcriptomics reference likely provides an upper bound on performance for reference-based deconvolution approaches. To provide a more realistic evaluation of performance for reference-based deconvolution approaches, we sought to deconvolve our simulated ST data of the MPOA using a scRNA-seq reference from a mouse brain atlasing effort[16]. Again, as a reference-free deconvolution approach, the performance of STdeconvolve does not change. However, again, the performance for all reference-dependent methods resulted in a significantly higher RMSE (Diebold–Mariano $p$ value $< 2.2 \times 10^{-16}$) than STdeconvolve (Fig. 2h, Supplementary Methods). Thus, STdeconvolve achieves comparable performance to reference-based, supervised deconvolution approaches when an ideal single-cell transcriptomics reference is used, and potentially better performance when an ideal single-cell transcriptomics reference is not available.

**STdeconvolve recovers perturbation specific gene expression profiles.** Though reference-based deconvolution approaches may accurately recover cell-type proportions in ST data, they currently do not deconvolve cell-type-specific gene expression profiles. Nonetheless, perturbations may induce cell-type-specific transcriptional changes in ST data that would not be identifiable by current reference-based deconvolution approaches unless perturbation-matched single-cell transcriptomics references are used. While the availability of scRNA-seq references grows due to single-cell atlasing initiatives, these datasets primarily represent collections of healthy tissues[8,9,17–19]. As such, there is a particular

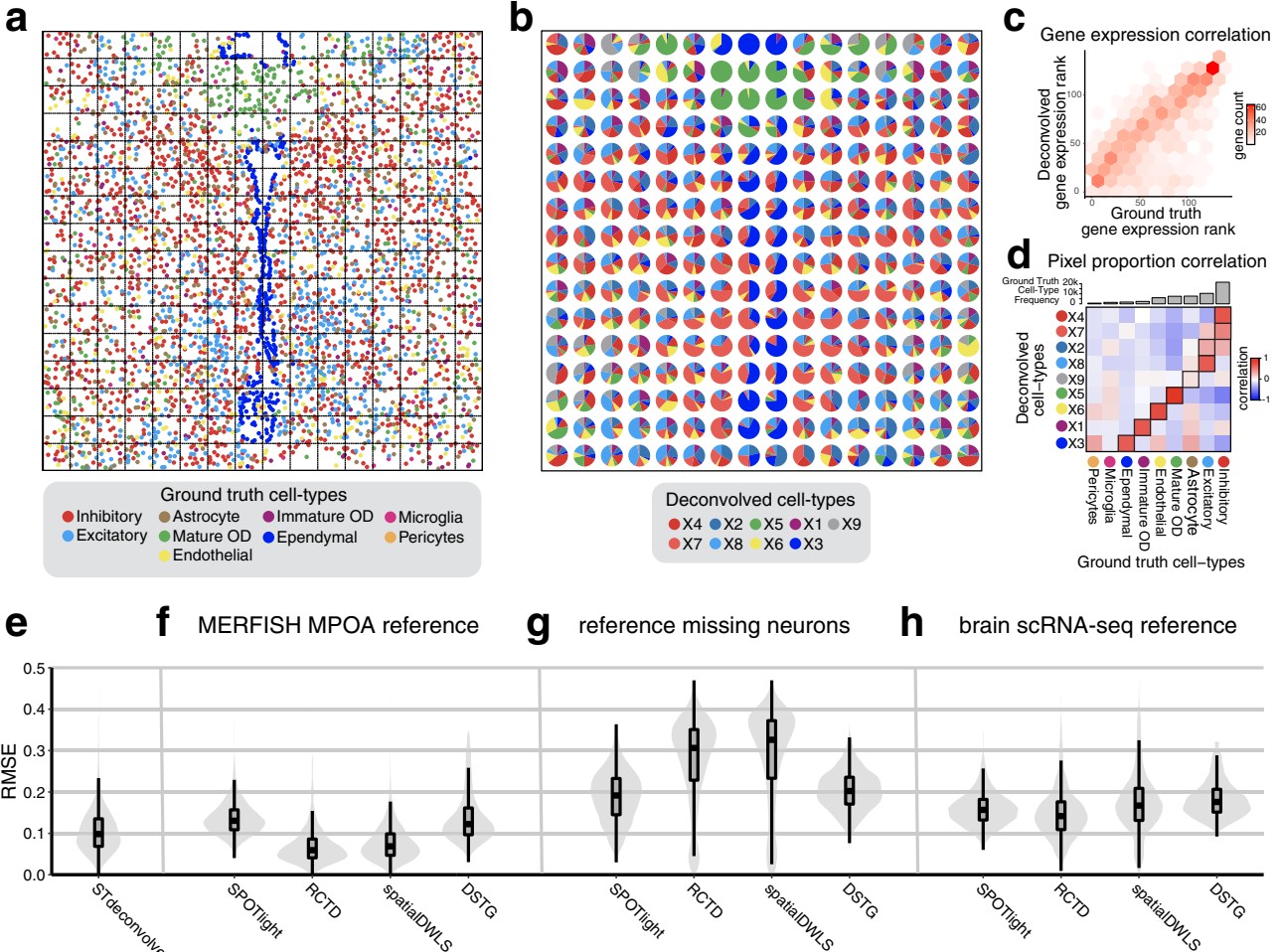

**Fig. 2 Deconvolution of simulated ST data. a** Ground truth single-cell resolution MERFISH data of one section of the MPOA partitioned into 100 μm² pixels (black dashed squares). Each dot is a single cell colored by its ground truth cell type label. **b** Proportions of deconvolved cell types from STdeconvolve represented as pie charts for each simulated pixel. **c** The ranking of each gene based on its expression level in the deconvolved cell-type transcriptional profiles compared to its gene rank in the matched ground truth cell-type transcriptional profiles. **d** Heatmap of Pearson's correlations between the deconvolved and ground truth cell type proportions across simulated pixels. Ground truth cell types are ordered by their frequencies in the ground truth dataset. Matched deconvolved and ground truth cell types are boxed. **e** Root-mean-square-error (RMSE) of the deconvolved cell-type proportions ($n = 3072$ pixels) compared to ground truth for STdeconvolve, **f** for supervised deconvolution approaches using the ideal single cell transcriptomics MERFISH MPOA reference, **g** for supervised deconvolution approaches using the single cell transcriptomics MERFISH MPOA reference with missing neurons, and **h** for supervised deconvolution approaches using a brain single-cell RNA-seq reference. Boxplots indicate median (middle line), 25th, 75th percentile (box) and 5th and 95th percentile (whiskers).

scarcity of suitable scRNA-seq references available for reference-based deconvolution of ST data in the context of disease and other perturbations.

In contrast to current reference-based deconvolution approaches, STdeconvolve can estimate cell-type transcriptional profiles in a manner that is not constrained by the expression profiles of specific cell types defined in single-cell transcriptomics references. We therefore sought to explore the potential of STdeconvolve in detecting these perturbation-driven cell-type-specific gene expression changes using simulated ST data from mixtures of single cells assayed by scRNA-seq (Fig. 3a, Supplementary Methods). Briefly, we took advantage of scRNA-seq data previously collected from mammary tissues of aged and young mice[20]. Previous transcriptional clustering analysis revealed a subpopulation of macrophages with age-associated gene expression changes. Specifically, aged macrophages upregulated *Cd274* and *Clec4d*, and downregulated *Coro1a* compared to young macrophages. Therefore, we simulated ST data of aged tissue using mixtures of aged macrophages and other luminal cells and

ST data of young tissue using mixtures of young macrophages and other luminal cells (Fig. 3b). We then sought to evaluate the ability of STdeconvolve to recover these age-associated gene expression changes in macrophages (Supplementary Methods). Applying STdeconvolve using $K = 2$ cell types to the simulated ST data of both aged and young tissue, we found that the deconvolved transcriptional profiles were highly correlated with the matched ground truth gene expression profiles from scRNA-seq in all cases (Supplementary Fig. S4A, B). Further, when we compared the deconvolved transcriptional profiles of aged versus young macrophages, we were able to identify upregulated genes included *Cd274* and *Clec4d*, and downregulated genes included *Coro1a*, consistent with the original study (Fig. 3c). Thus, STdeconvolve can potentially recover perturbation-driven cell-type-specific gene expression changes in ST data.

**Deconvolution provides distinct insights compared to clustering analysis.** Generally, we note that deconvolution of multi-

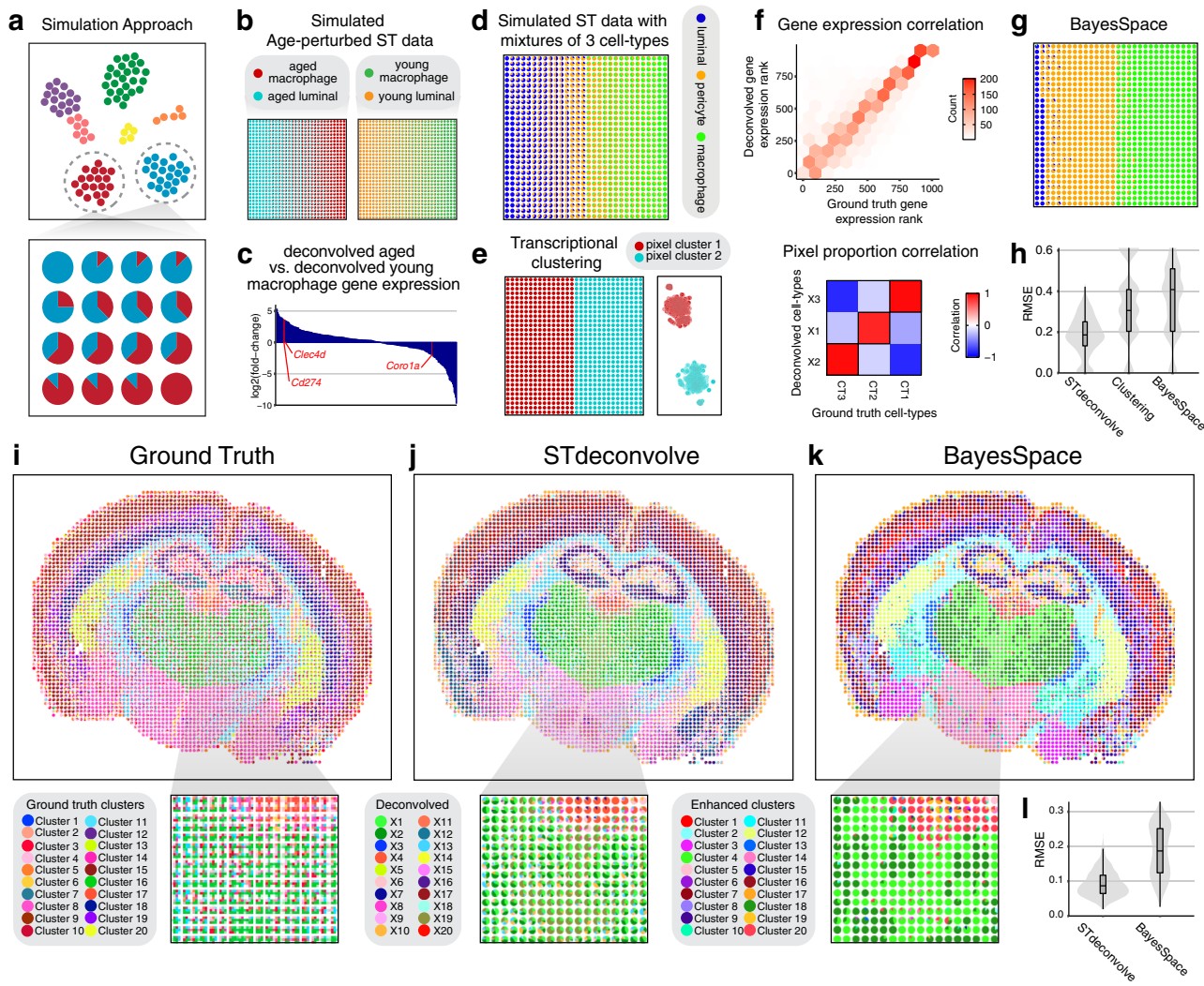

**Fig. 3 Comparing clustering versus deconvolution analysis for ST data. a** Overview of simulation approach. Starting from a single-cell RNA-seq clustering result visualized as a 2D t-SNE embedding with cells colored by cell type (top), gene expression counts from cells are combined to simulate cell-type mixtures, with the proportions of cell types are represented as pie charts for each arbitrary spatial pixel (bottom). **b** Simulated ST datasets of aged and young tissues using mixtures of aged macrophages with aged luminal cells and young macrophages with young luminal cells respectively represented as pie charts for each simulated ST pixel. **c** Bar chart of $\log_2$ fold-change for deconvolved aged macrophage versus deconvolved young macrophage gene expression. Select genes are highlighted in red. **d** Simulated ST dataset with 3 cell types represented as pie charts for each simulated ST pixel. **e** Clustering analysis results of simulated ST dataset with 3 cell types. Pie chart proportional representation (left) and t-SNE representation (right). **f** Deconvolution results for the simulated ST dataset with 3 cell types by STdeconvolve. The ranking of each gene based on its expression level in the deconvolved cell-type transcriptional profiles compared to its gene rank in the matched ground truth cell-type transcriptional profiles (top). Heatmap of Pearson's correlations between the deconvolved cell types proportions and ground truth cell types proportions across simulated pixels (bottom). **g** BayesSpace enhanced resolution clustering results for the simulated ST dataset with 3 cell types represented as pie charts. **h** Root-mean-square-error (RMSE) of the deconvolved cell-type proportions ($n = 900$ pixels) compared to ground truth for the simulated ST dataset with three cell types. **i** Ground truth cell-type proportions derived from single-cell resolution MERFISH data of the mouse brain partitioned into 100 $\mu m^2$ pixels. **j** Deconvolved cell-type proportions for the mouse brain by STdeconvolve. **k** Enhanced resolution clustering for the mouse brain by BayesSpace. Inset highlights an interior region corresponding approximately to the thalamus. **l** Root-mean-square-error (RMSE) of the deconvolved cell-type proportions ($n = 716$ pixels) compared to single-cell clustering for the MERFISH mouse brain data for the inset interior region corresponding approximately to the thalamus. Boxplots indicate median (middle line), 25th, 75th percentile (box) and 5th and 95th percentile (whiskers).

cellular pixel resolution ST data can provide distinct insights from clustering analysis. To demonstrate this, we again simulated ST data using mixtures of single cells assayed by scRNA-seq (Supplementary Methods). Specifically, we simulated ST pixels comprised of mixtures of either luminal cells and pericytes or pericytes and macrophages (Fig. 3d). Applying clustering analysis to these ST pixels, we identified 2 clusters corresponding to either mixtures of luminal cells and pericytes or mixtures of pericytes and macrophages (Fig. 3e). In contrast, applying STdeconvolve

with $K = 3$, we were able to recover the proportional representations of luminal cells, pericytes, and macrophages as well as their original cell-type-specific transcriptional profiles (Fig. 3f).

Such differences between deconvolution and clustering analysis extends to resolution enhancing clustering approaches such as BayesSpace[21]. Briefly, BayesSpace utilizes a spatial prior that encourages spatially neighboring pixels to cluster into the same transcriptional cluster. Enhanced resolution clustering is obtained after subdividing each pixel and modeling the expression profiles

of the subpixels as additional latent parameters estimated in the Bayesian model. Applying BayesSpace with 3 clusters to our simulated ST data, we obtained 3 spatially discrete clusters corresponding to different mixtures of luminal cells and pericytes and mixtures of pericytes and macrophages (Fig. 3g, Supplementary Methods). Compared to STdeconvolve, both regular transcriptional clustering and resolution enhanced clustering with BayesSpace exhibited significantly higher RMSE (Diebold-Mariano $p$ value $< 2.2 \times 10^{-16}$) (Fig. 3h).

To further demonstrate the difference between deconvolution and clustering analysis for ST data, we again simulated ST data using a single-cell resolution MERFISH dataset of a coronal section of the mouse brain. We analyzed the single-cell resolution transcriptional profiles to identify 20 transcriptionally distinct cell types and again simulated multi-cellular pixel-resolution ST data by aggregating the single cells into 100 μm$^2$ pixels (Fig. 3i, Supplementary Fig. S5A, B, Supplementary Methods). The organization of cell types within the mouse brain is highly complex with many regions including the thalamus at the central region of this coronal section being composed of mixtures of multiple transcriptionally distinct cell types. We thus sought to evaluate whether STdeconvolve could better recover the proportional representation of cell types compared to resolution enhanced clustering with BayesSpace. Applying both STdeconvolve and BayesSpace, we generally recover the cell-type pixel proportions and visually recapitulate the spatial organization of cell types within various brain structures (Fig. 3j, k, Supplementary Methods). However, focusing in on the central region of the coronal section encompassing the thalamus, we indeed saw a visual difference between the spatial organization of cell types recovered by deconvolution via STdeconvolve compared to resolution enhanced clustering via BayesSpace (Fig. 3j, k inset). Quantifying performance, BayesSpace exhibited significantly higher RMSE compared to STdeconvolve (Diebold-Mariano $p$ value $< 2.2 \times 10^{-16}$) as a whole (Supplementary Fig S5C, though more discernably in the thalamus region (Fig. 3l). Taken together, deconvolution approaches such as STdeconvolve can reveal cell types and patterns not evident through clustering and resolution enhanced clustering approaches alone when applied to multi-cellular pixel resolution data for pixels containing heterogeneous mixtures of cell types.

**STdeconvolve characterizes the spatial organization of transcriptionally distinct cell types in real ST data.** Having demonstrated the capacity of STdeconvolve to recover cell-type proportions and transcriptional profiles in simulated ST data, we next sought to evaluate the performance of STdeconvolve by analyzing real 100 μm$^2$ resolution ST data of the mouse main olfactory bulb (MOB)[22]. The MOB consists of multiple bilaterally symmetric and transcriptionally distinct cell layers due to topographically organized sensory inputs[23]. While previous clustering analysis of ST data of the MOB revealed coarse spatial organization of coarse cell layers, finer structures such as the rostral migratory stream (RMS) could not be readily observed (Supplementary Fig. S6A, B). We applied STdeconvolve to identify $K = 12$ cell types (Fig. 4a, Supplementary Fig. S6C, Supplementary Methods) that either overlapped with or further split coarse cell layers previously identified from clustering analysis (Supplementary Fig. S6D). In particular, deconvolved cell-type X7 overlapped with the granule cell layer previously identified from clustering analysis and was spatially placed where the RMS is expected[24] (Fig. 4b). Upregulated genes in its deconvolved transcriptional profile, including *Nrep*, *Sox11*, and *Dcx*, are known to be associated with neuronal differentiation and upregulated in neuronal precursor cells within the RMS[25] (Fig. 4c,

Supplementary Fig. S6E). Higher resolution ISH staining of these genes further demarcates a region within the granule cell layer where the RMS is expected[19] (Fig. 4d). This suggests that deconvolved cell-type X7 may correspond to the neuronal precursor cell-type within the RMS unidentified from clustering analysis.

To further evaluate the biological reproducibility of deconvolved cell types, we applied STdeconvolve independently to 3 additional biological replicates of ST data of the MOB (Supplementary Methods). In each biological replicate, STdeconvolve consistently identified approximately 12 cell types (Supplementary Fig. S7A). Transcriptional profiles between deconvolved cell types were also highly correlated across biological replicates (Supplementary Fig. S7B–D). This suggests that STdeconvolve can reliably deconvolve consistent cell types, even across biological replicates.

As noted previously using simulated ST data, the performance of reference-based deconvolution approaches is sensitive to differences in cell-type composition between the single-cell transcriptomics reference and the ST data to be deconvolved. To demonstrate this with real ST data, we first compared STdeconvolve and reference-based deconvolution approaches using an appropriate MOB scRNA-seq reference[26] (Supplementary Methods). We found strong correlations between cell-type proportions estimated by STdeconvolve and the evaluated reference-based deconvolution approaches with a high degree of correspondence among all evaluated methods (Supplementary Fig. S8A, B). Notably, the proportion and transcriptional profile of deconvolved cell-type X8 identified by STdeconvolve to be enriched in the olfactory nerve layer correlated strongly with the proportion of olfactory ensheathing cells (OECs) identified by reference-based deconvolution approaches.

Next, to simulate a less suitable scRNA-seq reference, we removed OECs from the MOB scRNA-seq reference and again evaluated the performance of reference-based deconvolution approaches given this new scRNA-seq reference without OECs (Supplementary Methods). Again, as a reference-free deconvolution approach, the results of STdeconvolve do not change. However, for some reference-based deconvolution approaches, given this new reference without OECs, pixels in the olfactory nerve layer previously comprised of OECs were now predicted to be comprised of N2 cells (Supplementary Fig. S9A). Although we do not know the ground truth cell-type composition of this olfactory nerve layer, we have reasons to believe that this placement of N2 cells is erroneous. First, when a scRNA-seq reference with OECs was used, reference-based deconvolution approaches generally estimated N2 cells to be relatively rare. However, when a scRNA-seq reference without OECs was used, these reference-based deconvolution approaches substantially increased their estimated abundance of N2 cells (Supplementary Fig. S9B). Second, while the transcriptional profiles of OECs and N2 cells are highly correlated (Supplementary Fig. S9C), the two cell types exhibit significant transcriptionally differences. For example, top differentially upregulated genes in OECs are highly expressed in the olfactory nerve layer (Supplementary Fig. S8C) whereas top differentially upregulated genes in N2 cells are not well detected in the olfactory nerve layer (Supplementary Fig. S9D). This lack of detection of N2 cell marker genes within the olfactory nerve layer coupled with the rarity of N2 cells in the original reference-based deconvolution with OECs suggests that the placement of N2 cells in the olfactory nerve layer by reference-based deconvolution approaches when using a reference without OECs is erroneous.

Further, a single-cell transcriptomics reference may not always exist for the same tissue from which ST data was generated, prompting the use of a reference from a related but inherently

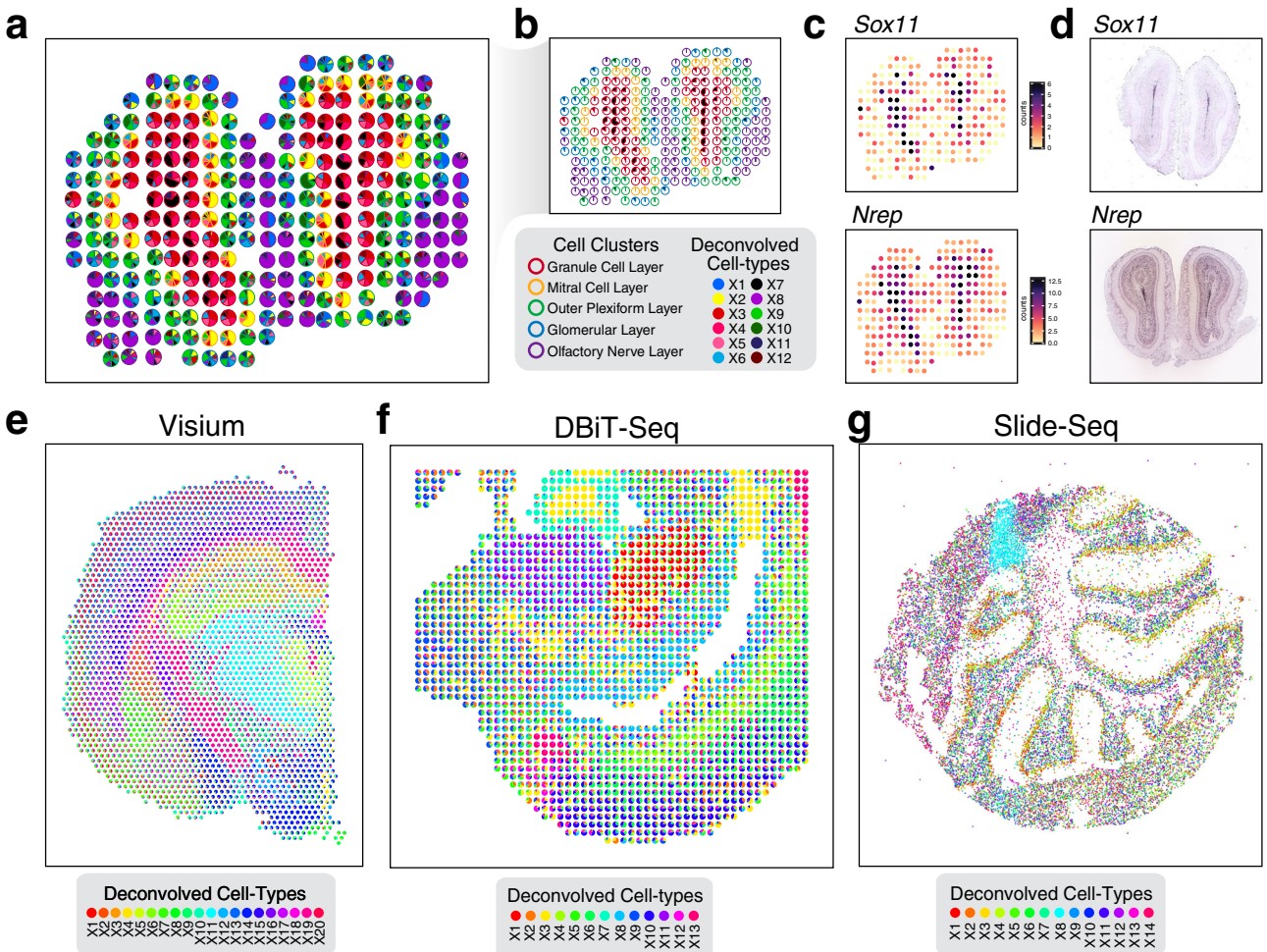

**Fig. 4 Deconvolution of ST data of varying resolution from multiple technologies by STdeconvolve. a** Deconvolved cell-type proportions for ST data of the MOB, represented as pie charts for each ST pixel. Pixels are outlined with colors based on the pixel transcriptional cluster assignment corresponding to MOB coarse cell layers. **b** Highlight of deconvolved cell-type X7. Pixel proportion of deconvolved cell-type X7 are indicated as black slices in pie charts. Pixels are outlined with colors as in (**a**). **c** Gene counts in each pixel of the MOB ST dataset for deconvolved cell-type X7's select top marker genes *Sox11* and *Nrep*. **d** Corresponding ISH images for *Sox11* and *Nrep* from the Allen Brain Atlas[19]. **e** Deconvolved cell-type proportions for Visium data of the mouse brain. **f** Deconvolved cell-type proportions for DBiT-seq data of the lower body of an E11 mouse embryo. **g** Deconvolved cell-type proportions for Slide-seq data of the mouse cerebellum.

different tissue source. To evaluate the potential effect of using a single-cell transcriptomics reference from a different tissue source on reference-based deconvolution approaches, we sought to deconvolve the MOB ST data using the scRNA-seq reference from the mouse brain described previously. Given this mouse brain reference, pixels in the olfactory nerve layer previously comprised of OECs were now predicted to be comprised of vascular leptomeningeal cells (VLMC) (Supplementary Fig. S10A). Again, although we do not know the ground truth cell-type composition of this olfactory nerve layer, top differentially upregulated genes in VLMCs are not well detected in the MOB (Supplementary Fig. S10B) and are therefore likely not truly present. Taken together, all this suggests that reference-based deconvolution approaches are sensitive to the cell types represented in the single-cell transcriptomics reference that is used, which may lead to inaccurate results and spurious cell-type assignments when a suitable reference is not available.

**STdeconvolve is applicable across diverse ST dataset resolutions and technologies.** We anticipate that continual technological improvements will enhance the resolution of ST data.

Already, ST technologies such as Visium (10X Genomics), Slide-seq[27], and DBiT-seq[28] have achieved resolution that can range from 50 μm² to 10 μm². Therefore, we sought to evaluate the performance of STdeconvolve on higher resolution ST data using both simulated as well as real ST data from higher resolution ST technologies including Visium, Slide-seq, and DBiT-seq.

First, to simulate higher resolution ST data, we again aggregated single-cell resolution MERFISH data of the MPOA into 50, 20, and 10 μm² resolution pixels. Applying STdeconvolve, we observed similarly strong correlations between the deconvolved cell-type transcriptional profiles and proportions with the ground truth (Supplementary Fig. S11A–D). Although the number of cells in each multi-cellular pixel did decrease as the resolution of the pixel increased as expected, we note that even higher resolution pixels may still contain multiple cells representing multiple cell types (Supplementary Fig. S11E, F). Thus, deconvolution may still be applicable to higher resolution ST data and STdeconvolve can still accurately deconvolve cell types within these higher resolution multi-cellular pixels.

Encouraged by STdeconvolve's ability to recover cell types in simulated high-resolution ST data, we then applied STdeconvolve to real high-resolution multi-cellular ST data from several

different technologies. First, we applied STdeconvolve to 50 μm² resolution ST data of a coronal section of the mouse brain from 10X Visium. Briefly, for 10X Visium, mRNAs from tissue sections are captured onto an array of DNA barcoded spots, resulting in RNA-sequencing measurements with gridded 2D spatial positional information. We applied STdeconvolve to identify $K = 20$ cell types that exhibit spatially distinct patterns that demarcate known brain structures such as the isocortex and fiber tracts (Fig. 4e, Supplementary Fig. S12, Supplementary Methods).

We next applied STdeconvolve to 25 μm² resolution ST data of the lower body of the E11 mouse embryo from DBiT-seq. Briefly, for DBiT-seq, parallel microfluidic channels are used to deliver DNA barcodes to the surface of a tissue to enable direct barcoding of mRNAs in situ, resulting in RNA-sequencing measurements in a 2D mosaic of spatial pixels. Previously, the authors identified 13 transcriptionally and spatially distinct features in the E11 mouse embryo including the atrium, ventricle, liver, and blood vessels containing erythrocyte coagulation. Applying STdeconvolve with $K = 13$, we identify deconvolved cell types that corresponded with similar spatially distinct features in agreement with the original findings (Fig. 4f, Supplementary Fig. S13A, Supplementary Methods). Moreover, the top genes in the deconvolved cell-type-specific transcriptional profiles contained the expected marker genes of the matching features, such as *Myh6* for the atrium, *Myh7* for the ventricle, *Apoa2* for the liver, and *Hba.a2* for the blood vessels containing erythrocyte coagulation in agreement with the original findings (Supplementary Fig. S13B).

Finally, we applied STdeconvolve to 10 μm² resolution ST data of the mouse cerebellum from Slide-seq. Briefly, for Slide-seq, mRNAs from tissue sections are captured onto densely packed barcoded beads, resulting in RNA-sequencing measurements with 2D spatial positional information. Previously, RCTD was also applied to this Slide-seq dataset with a matched Drop-seq scRNA-seq reference of the mouse cerebellum[29] to identify beads representing a distinct layers of Purkinje neurons and Bergmann glia. Applying STdeconvolve, we identified $K = 14$ cell types (Fig. 4g, Supplementary Methods) whose transcriptional profiles correlated strongly with cell types from the scRNA-seq dataset of the mouse cerebellum (Supplementary Fig. S14A). In particular, we found that the deconvolved transcriptional profiles of cell-type X4 and cell-type X2 correlated strongly with the transcriptional profiles of Purkinje neurons and Bergmann glia. Likewise, the deconvolved proportional representation of cell-type X4 and cell-type X2 also agreed significantly (Fisher's Exact *p* value < 2.2 × $10^{-16}$) with the predicted proportions of Purkinje neuron and Bergmann glia from RCTD (Supplementary Fig. S14B, C). Taken together, these results indicate that STdeconvolve can be applicable to a range of multi-cellular resolution ST technologies.

As the resolution of ST data improves, the number of spatially resolved pixels and cell types represented in the data will presumably also increase. We therefore sought to evaluate the scalability of STdeconvolve in anticipation of these increasingly larger datasets. To this end, we benchmarked the runtime and total memory usage by STdeconvolve when deconvolving varying numbers of cell types using varying numbers of genes across varying numbers of pixels (Methods). We found that both the runtime and memory usage by STdeconvolve increased linearly with the number of pixels and genes in the input dataset (Supplementary Fig. S15A) and is comparable to existing reference-based deconvolution methods when applied to current ST datasets[5]. Likewise, runtime scales with the number of deconvolved cell types $K$ in the input dataset though memory usage remains stable (Supplementary Fig. S15B). To enhance runtime efficiency, STdeconvolve has built in parallelization. In this manner, we anticipate that STdeconvolve will be amenable to larger ST data.

**STdeconvolve identifies immune infiltrates in breast cancer.** Finally, to demonstrate the potential of an unsupervised, reference-free deconvolution approach, we applied STdeconvolve to 100 μm² resolution ST data of 4 breast cancer sections[30]. Here, a matched scRNA-seq reference was not available and using a scRNA-seq reference from another breast cancer sample may be inappropriate due to potential inter-tumoral heterogeneity[31]. Transcriptional clustering of the ST pixels previously identified 3 transcriptionally distinct clusters that corresponded to 3 histological regions of the tissue: ductal carcinoma in situ (DCIS), invasive ductal carcinoma (IDC), and non-malignant[30] (Fig. 5a, Supplementary Fig. 16A, B). However, the tumor microenvironment is a complex milieu of many additional cell types[32]. We thus applied STdeconvolve to identify potential additional cell types and interrogate their spatial organization, resulting in $K = 15$ identified cell types (Fig. 5b, Supplementary Fig. S16C, Supplementary Methods). Of these, deconvolved cell types X3 and X13 pixel proportions corresponded spatially with pixels annotated as the non-malignant and DCIS regions, respectively (Supplementary Fig. S16D). Likewise, the deconvolved expression profiles for X3 and X13 included *KRT1*, a keratin gene specifically expressed in mammary myoepithelial cells[33], and *PRSS23*, a serine protease associated with proliferation of breast cancer cells[34], respectively, consistent with the non-malignant and DCIS annotations (Supplementary Fig. S17). Interestingly, the deconvolved expression profile for cell-type X15 included immune genes such as *CD74* and *CXCL10* (Fig. 5c–e, Supplementary Fig. S18). Gene set enrichment analysis also suggested that genes in the deconvolved expression profile for cell-type X15 are significantly enriched in immune processes such as T cell activation (Supplementary Data 1, Supplementary Methods). This suggests that deconvolved cell-type X15 may correspond to immune infiltrates. Further, we find a significant the number of pixels with a high proportion of deconvolved cell-type X15 corresponding to IDC regions (Fisher's exact *p* value = 0.001257) based on previous clustering and pathology annotations. In contrast, we do not see a significant number of pixels with a high proportion of deconvolved cell-type X15 corresponding to DCIS regions (Fisher's exact *p* value = 0.5625). This is consistent with previous observations that when comparing pure DCIS and IDC, infiltration of immune cells was significantly higher in IDC to pure DCIS[35,36].

The spatial organization of immune cells within tumors has been previously implicated to be relevant in breast cancer prognosis[37]. In particular, whether immune cells are infiltrated or excluded from the tumor is associated with tumor microenvironments that stratify patient outcomes[38]. To evaluate whether STdeconvolve may be able to distinguish infiltrated versus excluded spatial organization of immune cells in tumors, we simulated ST data representing infiltrated and excluded spatial organizations using mixtures of single cells assayed by scRNA-seq (Fig. 5f, Supplementary Methods). In both the simulated infiltrated and excluded cases, we find that STdeconvolve can effectively recover the cell-type transcriptional profiles (Fig. 5g) and enable the quantification of immune infiltration to help distinguish between infiltrated versus excluded spatial organization of immune cells (Fig. 5h). Therefore, we anticipate that STdeconvolve may be able to assist in deconvolving cell types in heterogeneous cancer tissues to recover potentially clinically interesting spatial organizational patterns.

## Discussion
Multi-cellular pixel-resolution ST technologies have enabled high-throughput transcriptomic profiling of small mixtures of cells within tissues but accurate identification of the underlying cell types within each pixel is critical for elucidating cell-type-

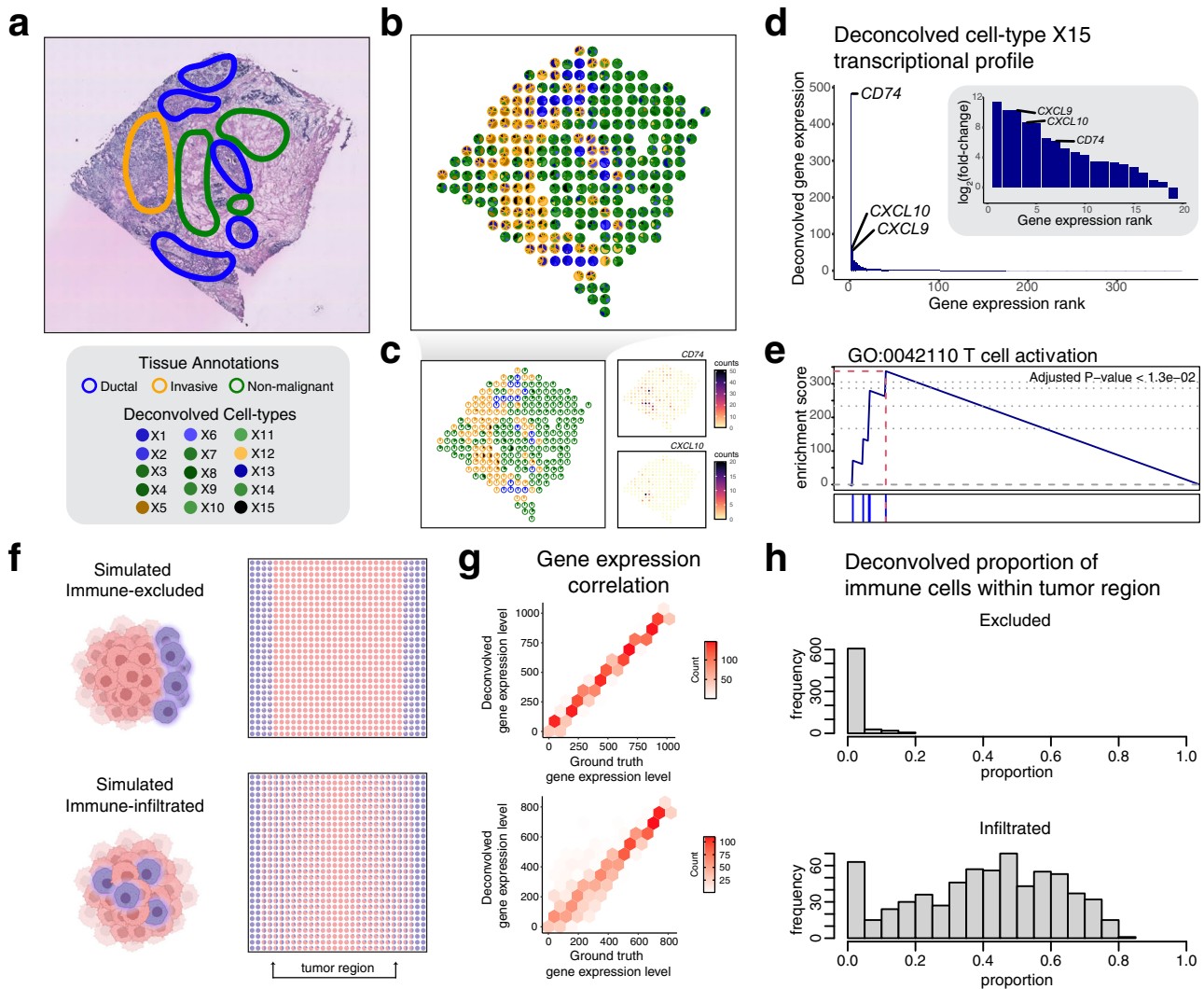

**Fig. 5 STdeconvolve characterizes the spatial organization of immune cells in real and simulated breast cancer ST data. a** An H&E-stained image of the breast cancer tissue with pathological annotations adapted from Yoosuf et al.[30]. **b** Deconvolved cell-type pixel proportions for ST data of a breast cancer tissue section, represented as pie charts. Pixels are outlined with colors based on the pixel transcriptional cluster assignment corresponding to 3 pathological annotations. **c** Highlight of deconvolved cell-type X15. Pixel proportion of deconvolved cell-type X15 are indicated as black slices in pie charts. Pixels are outlined with colors as in (**b**). Select genes corresponding cell-type X15's select top marker genes are shown. **d** Barplot of the deconvolved transcriptional profile of cell-type X15 ordered by magnitude. Inset represents the $\log_2$ fold-change of the deconvolved transcriptional profile of cell-type X15 with respect to the mean expression of the other 14 deconvolved cell-type transcriptional profiles. Select highly expressed and high fold-change genes are labeled. **e** Gene set enrichment plot for significantly enriched GO term "T cell activation" for deconvolved cell-type X15. **f** Simulated ST datasets of an immune-excluded tumor sample (top) and immune-infiltrated tumor sample (bottom) using mixtures of single cells represented as pie charts for each simulated ST pixel. **g** Deconvolution results for the simulated ST data by STdeconvolve. The ranking of each gene based on its expression level in the deconvolved-cell-type transcriptional profiles compared to its gene rank in the matched ground truth cell-type transcriptional profiles for the simulated immune-excluded tumor sample (top) and immune-infiltrated tumor sample (bottom). **h** Histogram of the deconvolved proportion of immune cells in the tumor region defined in (**f**) for the simulated immune-excluded tumor sample (top) and immune-infiltrated tumor sample (bottom).

specific spatial organizational patterns and gene expression variation. Although several deconvolution methods have already been developed to address this challenge, they currently rely on suitable single-cell transcriptomics references. As we have shown, this reliance on single-cell transcriptomics references constrains the spatial mapping of cell types to those in the reference, which may present limitations if there are missing cell types, mismatched cell types, perturbations, and batch effect differences between the single-cell transcriptomics reference and ST data to be deconvolved. Here, we have presented STdeconvolve, a reference-free computational approach to deconvolve cell-type proportions and their transcriptional profiles in multi-cellular pixel-resolution ST data. We have

demonstrated that STdeconvolve can accurately recover underlying cell-type proportions and their transcriptional profiles across a range of different ST technologies and resolutions. STdeconvolve further provides competitive performance to reference-based deconvolution approaches when an ideal single-cell transcriptomics reference is available and potentially better performance in more realistic circumstances where such an ideal reference is not available. Additionally, we showed the advantage of deconvolution over clustering-based analysis methods to interrogate heterogeneous mixtures of cell types. Likewise, using simulated ST data of aged-perturbed tissues, we showed that STdeconvolve can recover perturbation-driven cell-type-specific gene expression changes. Finally, we applied

STdeconvolve to identify putative immune infiltration in real and simulated breast cancer ST data.

Though we have shown that STdeconvolve can effectively recover cell-type proportions and transcriptional profiles in simulated and real ST data, its use of LDA modeling relies on several underlying assumptions, which may present limitations when these assumptions are not satisfied. Notably, the performance of LDA in accurately deconvolving cell types depends on the size of the dataset with respect to the number of pixels and the number of genes[39]. As such, deconvolution accuracy generally decreases for ST data containing fewer than 10 pixels (Supplementary Fig. S19). While we have generally found the number of pixels in most ST datasets to be well beyond 10 pixels after quality control filtering, the application of ST to profile tissue slivers or other thin structures covering only a few pixels may present challenges to deconvolution by STdeconvolve. Further, LDA modeling attempts to identify tightly occurring, and ideally non-overlapping groups of genes in the pixels as cell types. In this manner, if genes do not exhibit variability across pixels due to a homogeneous or uniform proportional representation of cell types across pixels (Supplementary Fig. S20), STdeconvolve may fail to deconvolve transcriptionally distinct cell types. Likewise, if the gene expression in the ST data is too sparse with high rates of stochastic drop-outs[40], then the LDA model may struggle to identify distinct groups of co-expressed genes and as such STdeconvolve may also struggle to deconvolve transcriptionally distinct cell types as well. Still, when such failures happen, STdeconvolve will indicate to users when distinct cell types cannot be adequately deconvolved.

Although we have demonstrated the applicability of STdeconvolve to current high-resolution multi-cellular pixel resolution ST data, as the resolution of ST data continues to increase, sub-cellular pixel-resolution ST technologies will also become more accessible. Already, a number sub-cellular pixel-resolution ST technologies have emerged[41–45]. As the capture efficiency at this resolution and likewise the biological questions of interest may differ substantially from multi-cellular ST data, we anticipate that new methods specifically suited for sub-cellular resolution ST data will be needed. Thus, STdeconvolve may not be best suited for analysis of such sub-cellular resolution ST data. Still, as we have noted previously, even as the resolution of ST data increases, some pixels may still contain multiple cells representing multiple cell types, suggesting that deconvolution may still be necessary. Likewise, the number of cells present in a pixel ultimately will depend on cell size, which can vary depending on the organism, tissue, and/or disease state being profiled. Ultimately, we believe that there will be a need to balance between resolution and throughput of ST technologies depending on the biological question of interest. The potentially larger tissue regions able to be covered by multi-cellular pixel resolution ST data may still be of interest and thus still require deconvolution. We anticipate that STdeconvolve will be applicable to data from a variety of current and future ST technologies as well as potentially inferred ST data[46] to reveal cell-type-specific spatial organizational patterns and transcriptional changes. In general, we foresee that reference-free deconvolution approaches such as STdeconvolve will contribute to the interrogation of the spatial relationships between transcriptionally distinct cell types in heterogeneous tissues.

## Methods

**STdeconvolve overview**. STdeconvolve uses LDA[47], a generative probabilistic model, to deconvolve the latent cell types contained within multi-cellular pixels of spatially resolved transcriptome (ST) measurements. In this context, each pixel is defined as a mixture of $K$ cell types represented as a multinomial distribution of cell-type probabilities ($\theta$), and each cell type is defined as a probability distribution over the genes ($\beta$) present in the ST dataset.

*LDA modeling*. The ST dataset is represented as a $D \times N$ matrix of discrete gene expression counts for each pixel $d$ and gene $n$. The total number of unique molecules, or total gene expression, in a given pixel $d$ is $M_d$.

As a generative probabilistic model, the LDA model generates a set of new pixels as follows:

For each pixel $d \in [1{:}D]$:

a. draw a cell-type distribution $\theta_d \sim Dir\,(\alpha)$, where $\theta_d$ is a multinomial distribution of length $K$ drawn from a uniform Dirichlet distribution with scaling parameter $\alpha$.

b. for each observed molecule $m$ in $M_d$:

  i. draw a cell-type assignment $z_{d,m} \sim mult(\theta_d) \in [1{:}K]$
  ii. based on this cell-type assignment, draw a gene $w_{d,m} \sim mult(\beta_{z_{d,m}}) \in [1{:}N]$

The central goal is to identify the posterior distribution of the latent parameters, $\theta$ and $\beta$, given the input data, i.e., the observed gene expression in the ST dataset. For each pixel $d$, the posterior distribution is defined as:

$$p(\theta_d, \mathbf{z} | \mathbf{w}, \alpha, \beta) = \frac{p(\theta_d, \mathbf{z}, \mathbf{w} | \alpha, \beta)}{p(\mathbf{w} | \alpha, \beta)} \tag{1}$$

where $\mathbf{z}$ is the vector of $M_d$ cell types assigned to each unique molecule in pixel $d$, and $\mathbf{w}$ is the vector of $M_d$ genes assigned to each molecule in pixel $d$. A variational expectation-maximization approach is used to estimate the values of the latent parameters[47,48]. Multiplying the marginal probabilities of the individual pixels gives the probability of the entire ST dataset. By default, $\beta$ is initialized with 0 for all cell types and genes, and $\alpha$ as $50/K$.

The resulting estimated $\theta$ and $\beta$ matrices represent the deconvolved proportions of cell types in each pixel and the gene expression profiles for each cell type, scaled to a library size of 1. $\beta$ represents a $K \times N$ gene-probability (i.e., expression) matrix for each cell type $k$ and each gene $n$ with each row summing to 1. The $\beta$ matrix can be multiplied by a scaling factor of one million to be more like conventional counts-per-million expression values for interpretability. $\theta$ represents a $D \times K$ pixel-cell-type proportion matrix for each pixel $d$ and each cell type $k$. LDA modeling in STdeconvolve is implemented through the 'topicmodels' R package[48].

Of note, LDA assumes for each cell type that there is a group of genes highly co-expressed with high probability. Therefore, STdeconvolve selects for genes more likely to be highly co-expressed within cell types, which can improve cell-type deconvolution.

*Selection of genes for LDA model*. Latent cell types are best discovered by LDA modeling if cell-type-specific marker genes are included in the input ST data while genes whose expression is shared across cell types are excluded. Therefore, to filter for genes that are more likely to be specifically expressed in particular cell types to improve cell-type deconvolution by LDA, STdeconvolve first removes genes that are not detected in a sufficient number of pixels. By default, genes detected in less than 5% of pixels are removed. Likewise, STdeconvolve also removes genes that are expressed in all pixels. By default, genes detected in 100% of pixels are removed. STdeconvolve then selects for significantly overdispersed genes, or genes with higher-than-expected expression variance across pixels, as a means to detect transcriptionally distinct cell types[14]. We assume that the proportion of cell types will vary across pixels and thus differences in their cell-type-specific transcriptional profiles manifest as overdispersed genes across pixels in the dataset.

If there are too many genes included in the input ST data, LDA may also struggle to identify non-overlapping clusters composed of distinct combinations of co-expressed genes. In these circumstances, users may modulate the number of informative genes included in the input matrix to ensure LDA convergence. By default, only the top 1000 most overdispersed genes are retained in the input ST data because we note that deconvolution accuracy in general stabilizes for larger numbers of informative features (Supplementary Fig. S22).

Additional gene filtering or cell-type-specific marker genes to include in the input ST data may also be augmented by the user.

*Selection of LDA model with optimal number of cell types*. The number of cell types $K$ in the LDA model must be chosen a priori. To determine the optimal number of cell types $K$ to choose for a given dataset, we fit a set of LDA models using different values for $K$ over a user defined range of positive integers greater than 1. We then compute the perplexity of each fitted model:

$$\text{Perplexity}(D) = \exp\left\{ -\frac{\log\big(p(D)\big)}{\sum\limits_{d=1}^{D} \sum\limits_{n=1}^{N} c_{d,n}} \right\} \tag{2}$$

where $p(D)$ is the likelihood of the dataset and $c_{d,n}$ is the gene count, or expression level, of gene $n$ in pixel $d$. We can interpret $p(D)$ as the posterior likelihood of the dataset conditional on the cell-type assignments using the final estimated $\theta$ and $\beta$. The lower the perplexity, the better the model represents the real dataset. Thus, the trend between choice of $K$ and the respective model perplexity can then serve as a

guide. By default, the perplexity is computed by comparing $p(D)$ to the entire input dataset used to estimate $\theta$ and $\beta$.

In addition, STdeconvolve also reports the trend between $K$ and the number of deconvolved cell types with mean pixel proportions <5% (as default). We chose this default threshold based on the difficulty of STdeconvolve and reference-based deconvolution approaches to deconvolve cell types at low proportions, (i.e., "rare" cell types) (Supplementary Note 2). We note that as $K$ is increased for fitted LDA models, the number of such "rare" cell types generally increases. Such rare deconvolved cell types are often distinguished by fewer distinct transcriptional patterns in the data and may represent non-relevant or spurious subdivisions of primary cell types. We can use this metric to help set an upper bound on $K$.

Generally, perplexity decreases and the number of "rare" deconvolved cell types increases as $K$ increases. Given these model perplexities and number of "rare" deconvolved cell types for each tested $K$, the optimal $K$ can then be determined by choosing the maximum $K$ with the lowest perplexity while minimizing number of "rare" deconvolved cell types. To further guide the choice of $K$, an inflection point ("knee") is derived from the maximum second derivative of the plotted $K$ versus perplexity plot and $K$ versus number of "rare" deconvolved cell types.

Still, for a given $K$, the fitted LDA model may fail to identify distinct cell types e.g., the distribution of cell-type proportions in each pixel is uniform. In such a situation, the Dirichlet distribution shape parameter $\alpha$ of the LDA model will be $\geq 1$ and STdeconvolve will indicate to the user that the fitted LDA model for a particular $K$ has an $\alpha$ above this threshold by graying out these $Ks$ in the trend plot.

Ultimately, the choice of $K$ is left up to the user and can be chosen taking into consideration prior knowledge of the biological system.

**Simulating ST data from single-cell resolution spatially resolved MERFISH data**. MERFISH data of the mouse medial preoptic area (MPOA) was obtained from the original publication[15]. Normalized gene expression values were converted back to counts by dividing by 1000 and multiplying by each cell's absolute volume. Datasets for an untreated female animal containing counts for 135 genes assayed by MERFISH were used. Genes with non-count expression intensities assayed by sequential FISH were omitted. Counts of blank control measurements were also removed. Cells were previously annotated as being one of 9 major cell types (astrocyte, endothelial, microglia, immature or mature oligodendrocyte, ependymal, pericyte, inhibitory neuron, excitatory neuron). Cells originally annotated as "ambiguous" were removed from the dataset to ensure the ground truth was composed of cells with distinguishable cell types. Because certain cell types may be enriched in specific regions of the MPOA, we combined 12 tissue sections across the anterior and posterior regions to ensure that all expected cell types would be well represented in the final simulated ST dataset. After filtering, the final dataset contained 59651 cells representing 9 total cell types and counts for the 135 genes.

To simulate a multi-cellular pixel resolution ST dataset from such single-cell resolution spatially resolved MERFISH data, we generated a grid of squares, each square with an area of 100 $\mu m^2$. Each square was considered a simulated pixel and the gene counts of cells whose x–y centroid was located within the coordinates of a square pixel were summed together. A grid of square pixels was generated for each of the 12 tissue sections separately and the simulated pixels for all 12 tissue sections were subsequently combined into a single ST dataset. For a given tissue section, the bottom edge of the grid was the lowest y-coordinate of the cell centroids and the left edge of the grid was the lowest x-coordinate. Square boundaries were then drawn from each of these edges in 100 $\mu m^2$ increments until the position of the farthest increment from the origin was greater than the highest respective cell centroid coordinate. After generating the grid, square pixels whose edges formed one of the outside edges of the grid were discarded in order to remove simulated pixels, which by virtue of their placement, encompassed space outside of the actual tissue sample. The retained pixels covered 49,142 out of the original 59651 cells in the 12 tissue sections. This resulted in a simulated ST dataset with 3072 pixels by 135 genes. We used the original cell type labels of each cell to compute the ground truth proportions in each simulated pixel. Likewise, to generate the ground truth transcriptional profiles of each cell type, we averaged the gene counts for cells of the same cell type from the original 59,651 cells and normalized the resulting gene count matrix to sum to 1 for each cell type. To simulate pixels of 50, 20, and 10 $\mu m^2$, an identical approach was taken using the same cells except those square boundaries were drawn from each edge in 50, 20, or 10 $\mu m^2$ increments.

**Annotation and matching of deconvolved and ground truth cell types**. Each deconvolved cell type was first matched with the ground truth cell type that had the highest Pearson's correlation between their transcriptional profiles. This was done by computing the Pearson's correlation between every combination of deconvolved and ground truth cell-type transcriptional profiles.

The assignment of deconvolved cell types to ground truth cell types was confirmed by testing for enrichment of differentially upregulated genes of the ground truth cell types in the deconvolved cell-type transcriptional profiles. To determine the differentially upregulated genes of the ground truth cell types, ground truth transcriptional profiles were converted to counts per thousand and low expressed genes, defined as those with average expression values less than 5, were removed. For each ground truth cell type, the log$_2$ fold change of each remaining gene with respect to the average expression across the other ground truth cell types was computed. Differentially upregulated genes were those with log$_2$ fold-change >1. We performed rank-based gene set enrichment analysis of the ground truth upregulated gene sets in each deconvolved cell-type transcriptional profile using the 'liger' R package[49]. A match to a ground truth cell type was confirmed and assigned if the ground truth gene set had the lowest gene set enrichment adjusted $p$ value that was at least <0.05, followed by the highest positive edge score[50], and then highest positive enrichment score to break ties.

**Deconvolution of additional simulated and real ST data**. Deconvolution of simulated and real ST data using STdeconvolve in addition to deconvolution of simulated and real ST data using supervised reference-based deconvolution approaches with various single-cell transcriptomics references is further detailed in Supplementary Methods.

**Comparison of deconvolution approaches**. How each supervised deconvolution approach was run is further detailed in the Supplementary Methods. To compare the performance between deconvolution methods, the RMSE was computed for each pixel between the deconvolved and matched ground truth cell-type proportions for each pixel in the ST dataset:

$$\text{RMSE} = \sqrt{\frac{\sum_{k=1}^{K} (\hat{y}_k - y_k)^2}{K}} \qquad (3)$$

where $K$ is the number of cell types, $\hat{y}_k$ is the predicted cell-type proportion for the cell type $k$, and $y_k$ is the ground truth cell-type proportion for the cell type $k$. To assess whether the distribution of pixel RMSEs was significantly lower for STdeconvolve compared to other methods, a one-sided Diebold–Mariano test[51] was used.

**Runtime and memory evaluation**. Using the Visium dataset described in Supplementary Methods 'Deconvolution of 10X Visium data with STdeconvolve', we generated an input ST dataset of 2702 pixels and feature selected for the top 1000 most significant overdispersed genes. Runtime of STdeconvolve was measured on randomly drawn subsets of input data. Five subsets are drawn with 2702 pixels and 50, 100, 200, 400, and 1000 genes, respectively. Another five subsets are drawn with the 1000 top overdispersed genes and 50, 100, 200, 400, and 1000 pixels, respectively. All subsets are deconvolved with cell type number ($K$) between 4 and 20 as input parameters. Runtime was measured using the R package 'microbenchmark'[52] (v1.4–7). Memory usage of STdeconvolve was measured using a similar sub-setting procedure and the R package 'profmem'[53] (v0.6.0). Total memory allocation was measured, which provides an upper bound for peak memory usage. Runtime and memory analyses were run on a machine with i7-6600U 2.60GHz CPU with 8GM of RAM.

**Reporting summary**. Further information on research design is available in the Nature Research Reporting Summary linked to this article.

## Data availability

Cell centroid coordinates, gene counts, and metadata of the MERFISH dataset of the mouse medial preoptic area[15] are available for download at https://datadryad.org/stash/dataset/doi:10.5061/dryad.8t8s248/. Cell volume data for animal FN7, datasets 171021_FN7_2_M22_M26 and 171023_FN7_1_M22_M26, was also obtained from the original authors. Cell centroid coordinates, gene counts, and metadata of Slice 2, replicate 1 of the MERFISH dataset of the mouse coronal section of the cortex are available as part of the Vizgen Data Release V1.0. May 2021 https://info.vizgen.com/mouse-brain-data. Gene count matrices and H&E images for all MOB replicates are available for download at https://www.spatialresearch.org/resources-published-datasets/doi-10-1126science-aaf2403/. Gene count matrices and H&E images for all breast cancer tissue replicates are available for download at https://www.spatialresearch.org/resources-published-datasets/doi-10-1126science-aaf2403/. 10X Visium data of the mouse coronal section of the cortex (V1_Adult_Mouse_Brain - Adult Mouse Brain (Coronal)) available for download at https://www.10xgenomics.com/resources/datasets/mouse-brain-section-coronal-1-standard-1-1-0. DBiT-seq dataset of E11 mouse embryo lower body (GSM4364242_E11-1L) gene count matrices available for download at https://www.ncbi.nlm.nih.gov/geo/query/acc.cgi?acc=GSE137986. Slide-seq of the mouse cerebellum (Puck_180819_12) available for download at https://singlecell.broadinstitute.org/single_cell/study/SCP354/slide-seq-study#study-download. Single-cell gene count matrices and cell metadata used in simulations available for download at https://www.ncbi.nlm.nih.gov/geo/query/acc.cgi?acc=GSE150580. scRNA-seq data of the mouse olfactory bulb[26] available for download at https://www.ncbi.nlm.nih.gov/geo/query/acc.cgi?acc=GSE121891. Drop-seq single-cell RNA-seq of mouse cerebellum available for download at http://dropviz.org/.

## Code availability

STdeconvolve is available as an open-source R software package[54] with the source code available as Supplemental Software and on GitHub at https://github.com/JEFworks-Lab/STdeconvolve. Additional documentation, instructions to generate the minimum

datasets to reproduce the analyses, and tutorials are available at https://jef.works/STdeconvolve/.

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

## Acknowledgements

Research reported in this publication was supported by the National Institute of General Medical Sciences of the National Institutes of Health under Award Number R35-GM142889.

## Author contributions

J.F. conceived the research. B.F.M. led computational work under the guidance of J.F. F.H., L.A., and A.S. contributed to computational work under the guidance of B.F.M. and J.F. All authors participated in interpretation and writing of the manuscript.

## Competing interests

The authors declare no competing interests.
