## [Peer Review File · Nature Communications]

Reviewers' Comments:

Reviewer #1:

Remarks to the Author:

Miller et al describe methods for deconvolving spatial transcriptomics (SpT) data using an approach based not on pre-defined references (as done by some of the current literature) but rather on modelling and parameter selection driven by the data. Given a gene-wise count matrix representing SpT data the proposed method, STdeconvolve, comprises two analysis steps. The first step determines the number of transcriptionally distinct types and the second infers the composition in each pixel, in terms of the above types.

This is an interesting work from the ML and data science perspective as it addresses an unsupervised approach to a complicated inference task. However – in light of the great progress that has taken place in SpT, where do we expect the “pixel” deconvolution task to still be relevant? One answer may come from SpT inferred from cheaper technologies like H&E or other staining approaches, including TMA. If so then such data should be clearly addressed and this specific application domain better emphasized.

Another potential application domain may be data based on FISH or other imaging approaches (similar to those used for the simulations, but somehow at a lesser resolution).

The above is our main suggestion for how to improve the manuscript.

More comments and suggestions follow below:

1. The related work and discussion is incomplete. There has been great progress in SpT - 10x Genomics have very high-resolution SpT (see Visium) with 5 cells in each barcoded spot. There is no mention of this advanced technology. How do the proposed methods stand in light of this technology? Do they apply? Are they required or bring any benefit? Will they be beneficial in the future? (5 cells per spot hardly merit deconvolution attempts - I doubt it can do much to improve upon the already high resolution given inevitable estimation errors - how sensitive is your method, i.e. will it add any benefit at these levels?).

As stated above - SpT inferred from WSIs, TMAs etc., as suggested by several studies, are not discussed and can be relevant for the methods described as the results can come in varying resolution levels.

2. Even if deconvolution is assumed beneficial for spots of 5 cells each, the authors should also address the scalability of the method to such high resolution datasets (e.g. in Visium you have thousands of genes measured across thousands of coordinates).

3. The 'pixel' terminology is confusing. Should really be defined in the beginning (as is the case with 'multi-pixel') and used consistently from there on. At least the authors can state that henceforth 'pixel' is used instead of 'multi cellular pixel' for brevity. For example, “STdeconvolve takes as input a spatial transcriptomics (ST) dataset of D pixels (rows)” - should it not be 'multi-cellular pixel'?

4. Please clarify how you chose the 'multi-pixels' where relevant (e.g. Fig 2 A) and whether you tested other resolutions on the same data or not. If yes – what were the conclusions? If not – why not? This is a great opportunity to test sensitivity to resolution by reporting a graph of performance per resolution (or similar) on the same data.

5. Is correlation in Fig. 2 sensitive enough to measure deconvolution performance? Additional statistics that are less 'aggregative' and coarse in nature should be considered to measure performance of deconvolution at the multi-pixel level against a ground-truth. Correlation offers one point-of-view. Specialized methods like statistical enrichment and more sensitive distance based approaches (such as considering the KL divergence at every pixel, in Fig 2B, for example) can be useful for performance evaluation.

6. Again – applying the methods to inferred SpT can be more relevant than applying it to direct SpT data as the latter is becoming near-single-cell SpT very quickly.

7. How were the ground truth cell types for Fig 2 determined? Where there 9 such types to start with? Are they marker based?

Minor

Title is slightly misleading - should be 'multi-pixel' (or similar), not pixel.

Figure 1D LDA modeling should be improved.

Figure 2D is too small and pixelated to clearly see what is written on the top left.

Not sure what Fig2 G and H demonstrate. Maybe clarify.

When comparing to available ground truths - it would be very interesting to demonstrate your performance and sensitivity to different 'region' size selections for the same data. Are you more accurate with higher resolutions?

Reviewer #2:

Remarks to the Author:

The transcript from Miller et al. presents a spatially resolved transcriptomic cell type deconvolution approach that unlike related methods does not rely on a single-cell transcriptomic reference. The approach is based on latent dirichlet allocation (LDA) that is typically used for topic modelling in text mining applications. Although LDA has been applied for cell type identification in single-cell RNA sequencing data, the application of LDA to deconvolve cell types in multicellular pixels of spatial transcriptomic data is elegant and to the extent that this reviewer is aware original. The authors combined previously published approaches and R functions (such as the identification of overdispersed genes and LDA) into a new R package for users to apply STdeconvolve to spatial transcriptomic data. The authors should be commended for a well-documented R package including example data. This reviewer had no problem installing and using the R package based on the available vignettes.

The authors further used previously published data sets to evaluate their approach and to compare it to two related methods that rely on single-cell transcriptomic references (SPOTlight and RCTD). Further, the authors highlighted the advantage of a reference-free approach applied to cancer samples without adequate single-cell references due to substantial inter-tumor heterogeneity. To demonstrate the application of STdeconvolve to cancer samples, the authors used spatial transcriptomic data of four breast cancer sections and drew a bold conclusion concerning possible clinical relevance. Despite the apparent advantages of reference-free cell type deconvolution, the authors did not sufficiently benchmark their method compared to the range of existing methods. They also did not mention relevant aspects like computational performance and scaling. There are a few recently published methods that are relevant to this work that were not considered or referenced by the authors.

Major concerns:

1. How does STdeconvolve compare to the recently published SpatialDWLS? This publication was not referenced by the authors even though it is highly related and includes a much more comprehensive benchmarking to existing methods than is provided by the authors for STdeconvolve. Can the authors provide a similarly comprehensive benchmarking that includes more existing methods and also considers computational performance and scaling with number of pixels?
2. How does STdeconvolve compare to the recently published spatial spot deconvolution method BayesSpace? Like STdeconvolve, BayesSpace does not require a single-cell reference and has the further advantage of providing subspot spatial resolution allowing for subsequent spatial analyses.
3. How significant do the authors expect their method to be in light of continuously improving ST methods with higher resolution? For example, DBiT-seq has a smallest achievable pixel size of about 5 μ m and recently published Seq-Scope achieves subcellular resolution already. Could such pixel deconvolution methods still be useful in such a scenario?
4. The authors claim that they can identify potentially clinically relevant spatial features because they identified immune cells at the invasive breast cancer margin. How did the authors define the interface of the cancerous and non-malignant regions? How did they calculate an enrichment of immune cells at this interface?

Clinically, the level of lymphocyte infiltration into the tumor mass is highly relevant but the simple presence of some immune cells adjacent to the tumor mass does not necessarily capture that information. Can the authors show that they can identify different levels of immune infiltration or differentiate tumors with an immune excluded vs infiltrated microenvironment? Further, can the authors based on STdeconvolve identify immunosuppressive environments based on the specific immune cell types that are present?

Minor concerns:

1. For the proof of concept, previously annotated cell types from a published MERFISH dataset provided the ground truth. How much does the evaluation and comparison of these methods depend on the quality and granularity of these previously defined cell types?

2. It is very hard to see the H&Es under the scatter-pie chart visualizations (E.g. in figure 3b). Can the authors provide a different visualization or quantification of that their cell types align with the histological structures?

Reviewer #3:

Remarks to the Author:

Reference-free cell-type deconvolution of pixel-resolution spatially resolved transcriptomics data

Miller et.al., in this work, have developed an unsupervised, reference-free approach for deconvolving multi-cellular pixel resolution of spatially resolved transcriptome data. In comparison to other recent approaches, their methodology is unique in that it is devoid of references. When sufficient single-cell references are available, the authors claim that their method recapitulates expected biology and gives comparable performance to existing supervised approaches, as well as potentially higher performance when acceptable single-cell references are not available. The paper is well written and could be accepted upon revision of the issues listed below.

1. STdeconvolve first feature selects for gene likely to be informative of transcriptionally distinct cell-types (line 45-46):

a. Is the approach sensitive to the number of genes (in the first step: feature selection of informative genes) How many minimum and maximum genes are required?

2. Further partitioning the ground-truth excitatory and inhibitory cell-types into additional subtypes (76 total) based on the previous annotation found that these deconvolved cell types correlated with specific combinations of neuronal subtypes (line 65-68)

a. It seems that the MERFISH paper identified ~70 different neuronal populations (PMID:30385464). It is not clear how the 9-ground truth cell-types (Figure 2D) were selected if the original data (ground truth) has 76.

b. For figure 3E, can the cluster (right vertical labels) be identified only based on correlation? If so, then it's hard to judge which K value (K=9 or K=76) is performing better? For instance, X7 which is an inhibitory neuron appears to be similarly correlated to both excitatory and inhibitory (Figure S3 D). Similarly, X3 rather than X9 looks more like astrocytes (Figure S3 B). Can a graph similar to Figure S14 be provided for different Ks? Can the cluster identification (in addition to correlation) be marker-assisted? For instance, Gad1 and Gad2 profile for the inhibitory cluster. Sl17a7 for Excitatory cluster.

3. We evaluated the performance of each approach using the root-mean-square-error of the deconvolved cell-type proportion to ground truth across simulated pixels. In general, we find the performance of STdeconvolve to be comparable to SPOTlight and RCTD (lines 79-82).

a. Can you provide the p-value of the difference between the three methods in Figure S5? How does the RSME plot look when you consider K=76 and ST dataset at 20um (described in lines 51-

76)?

b. Please show all figures for SPOTlight and RCTD similar to Figure S3 and S4.

c. Recently, graph-based semi-supervised algorithms have been developed (PMID:33480403, 33303016). Please show how STdeconvolve stacks up against them.

d. Although there are budgetary constraints (line 40), the availability of numerous scRNAseq datasets online makes it simple to obtain reference data for SPOTlight and RCTD. Naturally, the references will not be as ideal as the ones used by the authors (i.e., from the same source). In this regard, STdeconvolve, being reference-free, will be of great advantage. The authors must demonstrate, however, that STdeconvolve surpasses SPOTlight and RCTD when the source reference is not available (i.e., reference scRNAseq data not from the same study).

Reviewer #4:

Remarks to the Author:

This paper proposes to apply Latent Dirichlet Allocation (LDA) to perform an unsupervised deconvolution of spatial transcriptomics data without any single-cell reference data. It is noteworthy that they accurately inferred the presence of the rostral migratory stream in the olfactory bulb that other methods previously failed to detect. However, applying LDA to decompose bulk RNA-seq data has been already considered by Dey et al. (PLoS Genetics, 2017, <https://doi.org/10.1371/journal.pgen.1006599>), so the novelty of this manuscript is questionable. The authors should cite this work and explain the novelty of their contribution. Furthermore, as detailed below, some of the assumptions of LDA do not seem to hold in the spatial transcriptomics case, but these issues are not addressed in the manuscript. Lastly, more comprehensive benchmarking needs to be performed to demonstrate the improvement over other approaches.

Major Comments:

1. The method has been applied to only MERFISH data, so it is unclear how it would perform when applied to whole-transcriptome spatial data with much more features. Benchmarking using single-cell-resolution whole-transcriptome data from XYSeq or sci-Space would be worthwhile. Also, the method could be benchmarked on multi-cellular whole-transcriptome data from Slide-Seq or 10X Visium.

2. In addition to SPOTlight and RCTD, there are other deconvolution methods, including cell2location, SpatialDWLS, stereoscope, and DSTG. The benchmarking should be expanded to include these methods and other settings/data types.

3. Is the spatial structure of ST data used anywhere? If not, the problem that STdeconvolve attempts to solve seems to be in the purview (except for possibly minor sample size considerations) of general unsupervised reference-free joint deconvolution of multiple bulks, for which several methods exist (e.g., CDSeq, TOAST, UNDO, BayesCCE) and against which STdeconvolve should be benchmarked. (BayesCCE was developed for methylation data, but features essentially the same generative model as it applies for RNA-Seq, and so should be tried.) Moreover, if spatial structure is indeed not accounted for, then any kind of single-cell dataset (not only ST) could be artificially divided up into different pixels, used as ground truth data and hence benchmarked against. These simulated pixels could also be manually tuned to satisfy more strongly or weakly LDA's assumptions (e.g., sparsity), with resultant effect on performance analyzed.

4. The spatial correlation between pixels violates the LDA assumption. This issue should be addressed.

5. The theta and beta inference appear to be inconsistent; that is, even in the limit of infinite data, they might be misestimated. Intuitively, this should be due to the normalization of all cell types'

expression vectors onto the simplex (i.e., normalizing their library sizes to unity)—in the worst case two cell types are perfect multiples of each other (in which case every gene is differentially expressed!), but the current LDA setup seems unable to tell them apart. More generally, in the large data limit, the fraction of reads coming from cell type k for a fixed gene g should be (for a single pixel d):

$$\theta_d(k) * e_g(k) / (\sum_{j=1}^K \theta_d(j) e_g(j)),$$

where $e(k)$ is the unnormalized expression vector of cell type k . Since the true $\beta(k) = e(k) / \|e(k)\|_1$, the same quantity under LDA becomes

$$(\theta_d(k) * e_g(k) / \|e(k)\|_1) / (\sum_{j=1}^K \theta_d(j) e_g(j) / \|e(j)\|_1),$$

which (unless the library sizes are all identical to begin with) is different from the above.

6. Analyzing the robustness of supervised methods to perturbations is important, and its results should indeed inform the value of unsupervised methods, but might profit from a more thorough discussion. E.g., if I understand correctly, Figure S5B kicks the two most abundant cell types out of the reference, for which significant reductions in accuracy aren't surprising. What happens if less abundant cells are removed?

7. There is no discussion of runtimes or the computational resources needed to run the algorithm.

8. In the breast cancer application, would it be possible to apply RCTD and SPOTlight using single-cell reference datasets from a different breast cancer sample in order to quantitatively compare the performance, rather than qualitatively accounting for the tumor microenvironment heterogeneity?

Minor Comments:

1. The total bifurcation of results in the main text and all methodological discussion in the supplement seems rather odd. I would prefer a short overview of the model and its strengths/weaknesses in the main text.

2. Please clarify how genes are selected for LDA. How were spatial variability and its expectation calculated? Based on the description in "Gene selection for the LDA model," it seems that only the genes with global spatial expression variability are selected, but genes with local spatial expression variability could be important as well, and the selection of genes should be based on their cell-type-specific DE, rather than spatial DE, if the goal is to deconvolve cell types.

3. How many genes do we need to distinguish between cell types accurately?

4. For the MOB dataset, are there particular reasons why replicates 2,5,8 and 12 were chosen? Why not deconvolve and report summary statistics for all replicates?

5. In a similar vein, aggregating multiple replicates would be an interesting way to explore sample size related concerns somewhat orthogonal to the varying of pixel numbers done in supplementary note 2.

6. When comparing proportions, the mean L1 metric might be both more intuitive (mean estimation error per cell type) and rigorous (through its relation to the TV distance) than RMSE.

7. It would be helpful to see how the error in proportion inference depends on space or other properties of the ground truth distributions. I.e., is there any property of the ground truth mixture that guarantees accurate inference?

8. Are the N2 and OEC clusters transcriptionally similar? If they are, then the behavior of the supervised methods is expected and, indeed, arguably desirable.

9. The model performance measured by transcriptional profile correlation in Fig S3B seems rather strange. Why does ground truth immature OD have a relatively large correlation with every deconvolved cell type?
10. Does Fig 2C correspond to some aggregation of 9 cell types (if so, how was the aggregation of gene ranks performed) or just a single cell type?
11. The model performance quantified by the pixel cell-type proportion in Fig 2D is not as good as that in Fig 1 of the RCTD paper (Cable et al., 2021). Could you explain why astrocyte and pericyte are mapping to X3 together with ependymal, while Microglia does not have a mapped prediction?
12. In order to properly evaluate the correlation structures in Figure 2D, S3B, etc., it would be helpful to display the correlations between true cell types as well.
13. Instead of Fig 3C and Fig S6E, it might be better to show volcano plots with p-values to indicate the DE significance.
14. Could you include the "proportion" evaluation of Figure S3C for the two supervised methods, too?
15. Figure S6D: Does deconvolving the data with $K=5$ reproduce the ground truth clusters?
16. Producing Figure S7 for the MERFISH data as well might help to interpret it.
17. Fig S13 is difficult to interpret and it makes STdeconvolve look poor in comparison to the other methods. I felt this was somewhat insufficient to explore the implications of deconvolving with closely related cell types.
18. In lines 174-180 of Online Methods describes a method comparison based on RMSE of individual cell types: I may have missed where its results are reported.

Point by Point Response – Overview

We sincerely thank the editor and the reviewers for their insightful and constructive feedback in helping us improve this manuscript. We have now revised the manuscript to address all the points raised by the reviewers, organized herein as a point-by-point response. Throughout this point-by-point response, reviewer comments are shown in **blue**, with our responses in **green**, and changes to the manuscript in **black**.

Briefly, our manuscript was originally submitted in a Brief Communication format. We have now expanded the manuscript to a full Article format. This revised version now includes assessments of additional datasets from various spatially resolved transcriptomics technologies as well as additional benchmarks against supervised and semi-supervised, reference-based deconvolution approaches.

Reviewer 1

Overview 1-0: Miller et al describe methods for deconvolving spatial transcriptomics (SpT) data using an approach based not on pre-defined references (as done by some of the current literature) but rather on modelling and parameter selection driven by the data. Given a gene-wise count matrix representing SpT data the proposed method, STdeconvolve, comprises two analysis steps. The first step determines the number of transcriptionally distinct types and the second infers the composition in each pixel, in terms of the above types.

This is an interesting work from the ML and data science perspective as it addresses an unsupervised approach to a complicated inference task. However – in light of the great progress that has taken place in SpT, where do we expect the “pixel” deconvolution task to still be relevant?

One answer may come from SpT inferred from cheaper technologies like H&E or other staining approaches, including TMA. If so then such data should be clearly addressed and this specific application domain better emphasized. Another potential application domain may be data based on FISH or other imaging approaches (similar to those used for the simulations, but somehow at a lesser resolution). The above is our main suggestion for how to improve the manuscript.

We thank the reviewer for bringing up the recent great progress in spatially resolved transcriptomics technology development. Indeed, these advanced technologies are now enabling spatially resolved measurements at higher spatial resolutions ranging from 50 μm^2 to 10 μm^2 .

We do anticipate that pixel deconvolution tasks in general will still be relevant to analyzing these datasets. First, we note that even at these higher resolutions, pixels may still comprise multiple cells that represent multiple cell-types. To demonstrate this, we simulated higher resolution ST data using single-cell resolution MERFISH data of the MPOA and evaluated the number of ground truth cells and cell-types across the simulated pixels. Although the number of cells in each multi-cellular pixel did decrease as the resolution of the pixel increased as expected, we note that even higher resolution pixels may still contain multiple cells representing multiple cell-types. Therefore, pixel deconvolution would still be applicable. We have included these results in a new section called “**STdeconvolve is applicable across diverse ST dataset resolutions and technologies**” (Lines 320 to 384) in the revised manuscript, with relevant excerpts provided below for the reviewer’s reference.

We anticipate that continual technological improvements will enhance the resolution of ST data. ...Therefore, we sought to evaluate the performance of STdeconvolve on higher resolution ST data...First, to simulate higher resolution ST data, we again aggregated single-cell resolution MERFISH data of the MPOA into 50, 20, and 10 μm^2 resolution pixels....Although the number of cells in each multi-cellular pixel did decrease as the resolution of the pixel increased as expected, we note that even higher resolution pixels may still contain multiple cells representing multiple cell-types (Supplementary Figure S11E-F). Thus, deconvolution may still be applicable to higher resolution ST data and STdeconvolve can still accurately deconvolve cell-types within these higher resolution multi-cellular pixels.

E**F**
Supplementary Figure S11E-F. E) Distributions of number of individual cells contained within pixels of each simulated resolution of the MERFISH MPOA ST dataset. Numbers above each dataset indicate the total number of pixels for each simulated ST data. F) Stacked bar plots showing the fraction of pixels that contain a distinct number of different cell-types for each simulated resolution of the MERFISH MPOA ST dataset. This was done for pixels that contain at 2 or more individual cells. Numbers within each box indicate the total number of pixels with that many distinct cell-types and number above the bars indicate the total number of pixels with 2 or more individual cells.

Second, we would like to emphasize that deconvolution provides distinct insights from clustering analysis. Given that even at high resolutions, pixels may contain multiple cells comprising multiple cell-types, we now include simulations to demonstrate that clustering multi-cellular pixels can give differing results than deconvolution. Briefly, we simulated ST data comprised of mixtures of either luminal cells and pericytes or pericytes and macrophages (Figure 3D, Supplementary Methods). Applying clustering analysis to these ST pixels, we identified 2 clusters corresponding to either mixtures of luminal cells and pericytes or mixtures of pericytes and macrophages (Figure 3E). In contrast, applying STdeconvolve with $K=3$, we were able to recover the proportional representations of luminal cells, pericytes, and macrophages as well as their original cell-type specific transcriptomic profiles (Figure 3F). Such differences between deconvolution and clustering analysis extends to resolution enhancing clustering approaches as well as real tissue settings. We have included these results in a new section called **“Deconvolution provides distinct insights compared to clustering analysis”** (Lines 203 to 244) in the revised manuscript, with relevant excerpts provided below for the reviewer’s reference.

Generally, we note that deconvolution of multi-cellular pixel resolution ST data can provide distinct insights from clustering analysis. To demonstrate this, we again simulated ST data using mixtures of single cells assayed by scRNA-seq (Supplementary Methods). Specifically, we simulated ST pixels comprised of mixtures of either luminal cells and pericytes or pericytes and macrophages (Figure 3D). Applying clustering analysis to these ST pixels, we identified 2 clusters corresponding to either mixtures of luminal cells and pericytes or mixtures of pericytes and macrophages (Figure 3E). In contrast, applying STdeconvolve with $K=3$, we were able to recover the proportional representations of luminal cells, pericytes, and macrophages as well as their original cell-type specific transcriptional profiles (Figure 3F).

Figure 3D-F. Comparing clustering versus deconvolution analysis for ST data. D) Simulated ST dataset with 3 cell-types represented as pie charts for each simulated ST pixel. E) Clustering analysis results of simulated ST dataset with 3 cell-types. Pie chart proportional representation (left) and tSNE representation (right). F) Deconvolution results for the simulated ST dataset with 3 cell-types by STdeconvolve. The ranking of each gene based on its expression level in the deconvolved-cell-type transcriptional profiles compared to its gene rank in the matched ground truth cell-type transcriptional profiles (top). Heatmap of Pearson's correlations between the deconvolved cell-types proportions and ground truth cell-types proportions across simulated pixels (bottom).

Still, moving forward, we anticipate that there will be a need to balance between resolution and throughput of ST datasets depending on the biological question of interest. Therefore, the comparative larger tissue region covered by multi-cellular pixel resolution ST data may still make such data useful and thus still require pixel deconvolution. We have now included a discussion of these thoughts in the revised manuscript (Lines 479-482), provided below for the reviewer's reference.

Ultimately, we believe that there will be a need to balance between resolution and throughput of ST technologies depending on the biological question of interest. The potentially larger tissue regions able to be covered by multi-cellular pixel resolution ST data may still be of interest and thus still require deconvolution.

The reviewer further brings up an interesting potential application for pixel deconvolution for spatially resolved transcriptomics inferred from cheaper technologies like H&E or other staining approaches, including TMA. While we agree that such applications could be highly impactful due to its broader financial accessibility, we also anticipate that such applications will need to be coupled with additional error modeling to accommodate potential error propagation due to the additional inference step. Therefore, we foresee that new methods will be needed to be specifically tailored to such analyses of inferred ST data. We have now included a brief discussion of this potential in the revised manuscript (Lines 482-485), provided below for the reviewer's reference.

We anticipate that STdeconvolve will be applicable to data from a variety of current and future ST technologies as well as potentially inferred ST data¹ to reveal cell-type specific spatial organizational patterns and transcriptional changes.

Major Comment 1-1: The related work and discussion is incomplete. There has been great progress in SpT - 10x Genomics have very high-resolution SpT (see Visium) with 5 cells in each barcoded spot. There is no mention of this advanced technology. How do the proposed methods stand in light of this technology? Do they apply? Are they required or bring any benefit? Will they be beneficial in the future? (5 cells per spot hardly merit deconvolution attempts - I doubt it can do much to improve upon the already high resolution given inevitable estimation errors -

how sensitive is your method, i.e. will it add any benefit at these levels?). As stated above - SpT inferred from WSIs, TMAs etc., as suggested by several studies, are not discussed and can be relevant for the methods described as the results can come in varying resolution levels.

We thank the reviewer for raising this important point. As mentioned above, while the resolution of spatial technologies is increasing, we do anticipate that pixel deconvolution approaches in general will still be warranted. Nonetheless, it is still important to assess our approach across a range of different resolutions and technologies. To this end, we have now applied STdeconvolve to newer ST technologies as the reviewer has suggested, which include 10X Visium at $50 \mu\text{m}^2$ pixel resolution, DBiT-seq at $25 \mu\text{m}^2$ pixel resolution, and Slide-seq at $10 \mu\text{m}^2$ pixel resolution. We show that our deconvolution analysis is able to recapitulate previously reported findings and identify known biological features, indicating that STdeconvolve is applicable across different high resolution ST technologies. The results are presented in a new section called “**STdeconvolve is applicable across diverse ST dataset resolutions and technologies**” (Lines 320 to 384) in the revised manuscript, with relevant text provided below for the reviewer’s reference.

We anticipate that continual technological improvements will enhance the resolution of ST data. Already, ST technologies such as Visium (10X Genomics), Slide-seq², and DBiT-seq³ have achieved resolution that can range from $50 \mu\text{m}^2$ to $10 \mu\text{m}^2$. Therefore, we sought to evaluate the performance of STdeconvolve on higher resolution ST data using both simulated as well as real ST data from higher resolution ST technologies including Visium, Slide-seq, and DBiT-seq... Encouraged by STdeconvolve’s ability to recover cell-types in simulated high-resolution ST data, we then applied STdeconvolve to real high-resolution multi-cellular ST data from several different technologies. First, we applied STdeconvolve to $50 \mu\text{m}^2$ resolution ST data of a coronal section of the mouse brain from 10X Visium⁴. Briefly, for 10XVisium, mRNAs from tissue sections are captured onto an array of DNA barcoded spots, resulting in RNA-sequencing measurements with gridded 2D spatial positional information. We applied STdeconvolve to identify $K=20$ cell-types that exhibit spatially distinct patterns that demarcate known brain structures such as the isocortex and fiber tracts (Figure 4E, Supplementary Figure S12, Supplementary Methods).

Figure 4E. Deconvolution of ST data of varying resolution from multiple technologies by STdeconvolve. E) Deconvolved cell-type proportions for Visium data of the mouse brain.

Supplementary Figure S12. STdeconvolve pixel proportion of select deconvolved cell-types in 10X Visium data of the mouse brain. A) STdeconvolve pixel proportions for select deconvolved cell-types. B) Visually corresponding annotated brain region from the Allen Brain Atlas⁵.

We next applied STdeconvolve to $25 \mu\text{m}^2$ resolution ST data of the lower body of the E11 mouse embryo from DBiT-seq. Briefly, for DBiT-seq, parallel microfluidic channels are used to deliver DNA barcodes to the surface of a tissue to enable direct barcoding of mRNAs *in situ*, resulting in RNA-sequencing measurements in a 2D mosaic of spatial pixels. Previously, the authors identified 13 transcriptionally and spatially distinct features in the E11 mouse embryo including the atrium, ventricle, liver, and blood vessels containing erythrocyte coagulation. Applying STdeconvolve with $K=13$, we identify deconvolved cell-types that corresponded with similar spatially distinct features in agreement with the original findings (Figure 4F, Supplementary Figure S13A, Supplementary Methods). Moreover, the top genes in the deconvolved cell-type specific transcriptional profiles contained the expected marker genes of the matching features, such as *Myh6* for the atrium, *Myh7* for the ventricle, *Apoa2* for the liver, and *Hba.a2* for the blood vessels containing erythrocyte coagulation in agreement with the original findings (Supplementary Figure S13B).

Figure 4F. Deconvolution of ST data of varying resolution from multiple technologies by STdeconvolve. F) Deconvolved cell-type proportions for DBiT-seq data of the lower body of an E11 mouse embryo.

Supplementary Figure S13. STdeconvolve pixel proportions and transcriptional profiles of select deconvolved cell-types in DBiT-seq data of an E11 mouse embryo lower tail section. A) Pixel proportions of select deconvolved cell-types X7, X11, X1, and X3 corresponding to the atrium, ventricle, fetal liver, and erythrocyte coagulation, respectively, annotated in the previous publication³. B) Deconvolved transcriptional profiles for cell-types X7, X11, X1, and X3 with the expression of the top gene in each deconvolved transcriptional profile visualized in the original tissue (inset).

Finally, we applied STdeconvolve to 10 μm^2 resolution ST data of the mouse cerebellum from Slide-seq. Briefly, for Slide-seq, mRNAs from tissue sections are captured onto densely packed barcoded beads, resulting in RNA-sequencing measurements with 2D spatial positional information. Previously, RCTD was also applied to this Slide-seq dataset with a matched Drop-seq scRNA-seq reference of the mouse cerebellum⁶ to identify beads representing a distinct layers of Purkinje neurons and Bergmann glia. Applying STdeconvolve, we identified $K=14$ cell-types (Figure 4G, Supplementary Methods) whose transcriptional profiles correlated strongly with cell-types

from the scRNA-seq dataset of the mouse cerebellum (Supplementary Figure S14A). In particular, we found that the deconvolved transcriptional profiles of cell-type X4 and cell-type X2 correlated strongly with the transcriptional profiles of Purkinje neurons and Bergmann glia. Likewise, the deconvolved proportional representation of cell-type X4 and cell-type X2 also agreed significantly (Fisher's Exact p -value $< 2.2 \times 10^{-16}$) with the predicted proportions of Purkinje neuron and Bergmann glia from RCTD (Supplementary Figure S14B-C). Taken together, these results indicate that STdeconvolve can be applicable to a range of multi-cellular resolution ST technologies.

Figure 4G. Deconvolution of ST data of varying resolution from multiple technologies by STdeconvolve. G) Deconvolved cell-type proportions for Slide-seq data of the mouse cerebellum.

Supplementary Figure S14. STdeconvolve predicted cell-type proportions of Purkinje neurons and Bergmann glia in Slide-seq data of the mouse cerebellum compared to RCTD. A) Pearson's correlation between the deconvolved transcriptional profiles of the 14 STdeconvolve cell-types and best matched cell-types in the mouse

cerebellum scRNA-seq reference⁶. B) Beads predicted to contain Purkinje neurons (cyan) or Bergmann glia (magenta) by either RCTD (left) or STdeconvolve (right). C) Stacked bar plots indicating the fractions of beads predicted to contain Purkinje neurons (left) or Bergmann glia (right) by RCTD and STdeconvolve.

As mentioned previously, while the number of cells in each multi-cellular pixel does decrease at higher resolutions, we note that even higher resolution pixels may still contain multiple cells representing multiple cell-types. In particular, we have further simulated higher resolution ST data using our single-cell resolution MERFISH data of the MPOA. In our simulated $50 \mu\text{m}^2$ ST data, there was an average of approximately 4 cells per pixel, in close agreement with the average number of cells reported for 10X Visium. However, most of these multi-cellular pixels contained more than 2 different cell-types (Supplementary Figure S11E-F). Specifically, we have now demonstrated using simulation that when pixels are comprised of mixtures of cell-types, deconvolution can provide distinct insights from clustering analysis. These results are presented in a new section called “**Deconvolution provides distinct insights compared to clustering analysis**” (Lines 203 to 244) in the revised manuscript. As such, we anticipate that even with 5 or so cells per pixel, deconvolution may still be beneficial.

Finally, to address the accuracy of STdeconvolve for deconvolving data at higher resolutions, we have now included a new set of simulations and results confirming that deconvolution at higher resolutions by STdeconvolve is still accurate by comparing the Root-Mean-Squared-Error (RMSE) between predicted and ground truth cell-type pixel proportions of 9 major cell-types across simulated ST data of the MPOA using single-cell resolution MERFISH data, at different pixel resolutions. These results are presented in the revised manuscript (Lines 327-330) and included below for the reviewer’s reference.

First, to simulate higher resolution ST data, we again aggregated single-cell resolution MERFISH data of the MPOA into 50 , 20 , and $10 \mu\text{m}^2$ resolution pixels. Applying STdeconvolve, we observed similarly strong correlations between the deconvolved cell-type transcriptional profiles and proportions with the ground truth (Supplementary Figure S11A-D).

Supplementary Figure S11D. Root-mean-square-error (RMSE) of the deconvolved cell-type pixels proportions compared to ground truth cell-types at each resolution of the simulated ST dataset.

We further thank the reviewer for bringing up a point about ST data inferred from WSIs, TMAs etc. We agree that STdeconvolve, and deconvolution approaches in general, will have use not only for real ST data but also for inferred ST data. Indeed, STdeconvolve only needs a pixel by gene count matrix in order to perform deconvolution. Thus, as long as the data is partitioned into “coordinates” or pixels, STdeconvolve may be applied. We now discuss this potential application of STdeconvolve and reference one of these studies in the revised Discussion (Lines 482-485), which we also provide relevant excerpts below for the reviewer’s reference.

We anticipate that STdeconvolve will be applicable to data from a variety of current and future ST technologies as well as potentially inferred ST data¹ to reveal cell-type specific spatial organizational patterns and transcriptional changes.

Major Comment 1-2: Even if deconvolution is assumed beneficial for spots of 5 cells each, the authors should also address the scalability of the method to such high-resolution datasets (e.g. in Visium you have thousands of genes measured across thousands of coordinates).

We thank the reviewer for bringing up this important point about scalability. To assess the scalability of STdeconvolve to larger ST datasets, we have now assessed both the runtime and total memory usage of STdeconvolve in terms of varying the number of genes and number of pixels. We find that the runtime increases linearly with either the number of pixels or genes, as well as the number of cell-types to be deconvolved. As a specific example, deconvolution of 12 cell-types using all 2072 pixels and 1000 feature selected genes in a typical deconvolution analysis of a 10X Visium dataset completed in approximately 30 minutes on a personal computer, which we believe is a reasonable time cost and is comparable with previously published deconvolution approaches⁷. Further, we now allow for parallelization, which may further improve scalability. Importantly, the total memory cost was also minimal, with total memory reaching approximately 170 MBs for the full sized 10X Visium analysis. These results are now presented as Supplementary Figure S15 and we have provided the relevant text (Lines 373-384) below for the reviewer's reference.

As the resolution of ST data improves, the number of spatially resolved pixels and cell-types represented in the data will presumably also increase. We therefore sought to evaluate the scalability of STdeconvolve in anticipation of these increasingly larger datasets. To this end, we benchmarked the runtime and total memory usage by STdeconvolve when deconvolving varying numbers of cell-types using varying numbers of genes across varying numbers of pixels (Methods). We found that both the runtime and memory usage by STdeconvolve increased linearly with the number of pixels and genes in the input dataset (Supplementary Figure S15A) and is comparable to existing reference-based deconvolution methods when applied to current ST datasets⁷. Likewise, runtime scales with the number of deconvolved cell-types K in the input dataset though memory usage remains stable (Supplementary Figure S15B). To enhance runtime efficiency, STdeconvolve has built in parallelization. In this manner, we anticipate that STdeconvolve will be amenable to larger ST data.

Supplementary Figure S15. Runtime and memory usage by STdeconvolve. A) Runtime as a function of dataset size in terms of the number of pixels (top) or genes (bottom), and number of cell-types K . For scaling pixels, the top

1000 most significant overdispersed genes were selected and kept constant. For scaling genes, all 2702 pixels in the dataset were used and kept constant. B) Memory usage as a function of dataset size in terms of scaling pixels (top) or genes (bottom), and number of cell-types K . Again, for scaling pixels, the top 1000 most significant overdispersed genes were selected and kept constant. For scaling genes, all 2702 pixels in the dataset were used and kept constant.

Major Comment 1-3: The 'pixel' terminology is confusing. Should really be defined in the beginning (as is the case with 'multi-pixel') and used consistently from there on. At least the authors can state that henceforth 'pixel' is used instead of 'multi cellular pixel' for brevity. For example, "STdeconvolve takes as input a spatial transcriptomics (ST) dataset of D pixels (rows)" - should it not be 'multi-cellular pixel'?

We agree with the reviewer's sentiment that the term "pixel" is confusing. We have now clarified and emphasized our focus on multi-cellular pixels. Instances in the text of "pixel" have been changed to "multi-cellular pixel" where appropriate both in the text and figures.

Major Comment 1-4: Please clarify how you chose the 'multi-pixels' where relevant (e.g. Fig 2 A) and whether you tested other resolutions on the same data or not. If yes – what were the conclusions? If not – why not? This is a great opportunity to test sensitivity to resolution by reporting a graph of performance per resolution (or similar) on the same data.

We agree with the reviewer that this is an important opportunity to assess performance across ST dataset resolutions. First, we have clarified in the text that the simulated ST multi-cellular pixels shown in Figure 2B are based on $100 \mu\text{m}^2$ resolution (Lines 106-108). We have now also tested additional resolutions of simulated ST data of the MPOA using single-cell resolution MERFISH data at 50, 20, and $10 \mu\text{m}^2$. We find that both the deconvolved transcriptional profiles and predicted pixel proportions of the 9 deconvolved cell-types are highly correlated with the 9 major ground truth cell-types. Additionally, the RMSEs between STdeconvolve predicted pixel proportions of 9 major cell-types across the different pixel resolutions are similar and do not increase, suggesting that STdeconvolve is accurate even at high multi-cellular pixel resolution. These results are now included in a new section "**STdeconvolve is applicable across diverse ST dataset resolutions and technologies**" (Lines 327-330) and provided below for the reviewer's reference.

First, to simulate higher resolution ST data, we again aggregated single-cell resolution MERFISH data of the MPOA into 50, 20, and $10 \mu\text{m}^2$ resolution pixels. Applying STdeconvolve, we observed similarly strong correlations between the deconvolved cell-type transcriptional profiles and proportions with the ground truth (Supplementary Figure S11A-D).

Supplementary Figure S11A-C. Comparison of STdeconvolve cell-types to ground truth cell-types in high resolution pixel MERFISH MPOA ST datasets. A) Deconvolution accuracy in $50 \mu\text{m}^2$ simulated pixels. Left: The ranking of each gene based on its expression level in the transcriptional profiles of the deconvolved cell-types, compared to its gene rank in the transcriptional profile of the matched ground truth cell-type. Middle: Pearson's correlation between the transcriptional profiles of the 9 ground truth cell-types in MERFISH MPOA data and 9 deconvolved cell-types by STdeconvolve. Right: Pearson's correlation between the pixel proportions of the 9 ground truth cell-types in the MERFISH MPOA data and 9 deconvolved cell-types by STdeconvolve. B) Deconvolution accuracy in $20 \mu\text{m}^2$ simulated pixels. C) Deconvolution accuracy in $10 \mu\text{m}^2$ simulated pixels.

Major Comment 1-5: Is correlation in Fig. 2 sensitive enough to measure deconvolution performance? Additional statistics that are less 'aggregative' and coarse in nature should be considered to measure performance of deconvolution at the multi-pixel level against a ground-truth. Correlation offers one point-of-view. Specialized methods like statistical enrichment and more sensitive distance-based approaches (such as considering the KL divergence at every pixel, in Fig 2B, for example) can be useful for performance evaluation.

We thank the reviewer for raising this point and agree that multiple points-of-view should be used to assess the deconvolution accuracy. We quantitate the deconvolution performance of the different deconvolution approaches by calculating the RMSE of a given approach's predicted cell-type proportions for each pixel. Approaches can then be compared by testing for statistical differences in the resulting RMSE distributions. We agree that methods such as RMSE, which are more sensitive to distances like the differences between ground truth and predicted cell-type

proportions, are more useful for evaluating performance between different approaches. Additionally, we note that RMSE has also been used as a means to benchmark deconvolution approaches in previous work, such as spatialDWLS⁷.

In addition to the RMSE, we also utilize the Pearson's correlation between the transcriptional profiles or pixel proportions of predicted and ground truth cell-types. While this metric is more aggregative (compares cell-types overall instead of within individual pixels, for example), it provides additional information not available by the RMSE. For instance, our heatmaps of the Pearson's correlation of cell-type pixel proportions can help indicate if there are cell-types for which a given approach is erroneously, or correctly, classifying. Likewise, the Pearson's correlations between transcriptional profiles can inform us if an erroneous classification might be due to transcriptional similarities between cell-types. Furthermore, in cases in which cell-types are missing from a reference, it becomes difficult to fairly compare the Pearson's correlations quantitatively between approaches because the correlations are effectively only for the cell-types that were in the reference. For example, with respect to the MPOA, in scenarios where cell-types were removed from a reference, the remaining cell-types were well correlated with their corresponding ground truth cell-type and thus the correlations were high. However, if one were to look across the pixels, clearly the deconvolution accuracy of cell-type pixel proportions was erroneous, even if the relative levels of predicted cell-type proportions across pixels were correlated with the ground truth proportions. In this way, we have used RMSE to give a quantitative measure of the overall performance, and Pearson's correlation to provide insight into what might be contributing to this performance.

Finally, we have now also included additional visual representations of the deconvolved results, included below for the reviewer's reference. We hope that these multiple metrics as well as visual representations will provide readers with sufficient points of views for performance evaluation.

Supplementary Figure S3. Cell-type predictions of supervised and semi-supervised deconvolution approaches across simulated MERFISH MPOA ST dataset pixels with different single-cell transcriptomics references. A) Deconvolved pixel proportions of cell-types by each supervised and semi-supervised deconvolution approach using

the MERFISH MPOA single-cell transcriptomics data as the reference. B) Deconvolved pixel proportions of cell-types by each supervised and semi-supervised deconvolution approach using the MERFISH MPOA single-cell transcriptomics data with missing neurons as the reference. C) Deconvolved pixel proportions of cell-types by each supervised and semi-supervised deconvolution approach using the MERFISH MPOA single-cell transcriptomics data with missing ependymal cells as the reference.

Major Comment 1-6: Again – applying the methods to inferred SpT can be more relevant than applying it to direct SpT data as the latter is becoming near-single-cell SpT very quickly.

We thank the reviewer for bringing up this interesting application of inferred spatial transcriptomics data. Although Spatial Transcriptomics technologies are becoming more popular, their application to tissues can still be expensive and technically challenging. Conversely, inferring spatial gene expression from more ubiquitous and attainable whole slide tissue images, such as H&E-stained images, may prove to be an attractive alternative. We agree that STdeconvolve, and deconvolution approaches in general, will have use not only for real ST data but also for inferred ST data. With respect to WSIs such as H&E, these are typically segmented into smaller tiles, which can effectively act as multi-cellular pixels. STdeconvolve only needs a pixel by gene count matrix in order to perform deconvolution. Thus, as long as the data is partitioned into “coordinates” or pixels, STdeconvolve may be applied. In this way, STdeconvolve in conjunction with approaches that infer gene expression signature in the image tiles may help increase the resolution and inference of the spatial organization of cell-types in these tissue samples. We discuss the potential application of STdeconvolve to cases in which the spatial gene expression is inferred in the revised Discussion (Lines 482-485).

Again, while ST technologies resolutions are approaching single cell, we anticipate that there will be a balance between resolution and throughput. Likewise, near single-cell ST technologies may not be applicable in every case and likely will be expensive and technically challenging. For example, the comparative larger tissue region covered by multi-cellular pixel resolution ST data may still make such data useful and thus still require pixel deconvolution. Even in cases with high resolution, we note that multi-cellular pixels can still contain multiple distinct cell types, thus prompting the need for deconvolution approaches.

Major Comment 1-7: How were the ground truth cell types for Fig 2 determined? Where there 9 such types to start with? Are they marker based?

We thank the reviewer for raising this point of clarification. The simulated ST data of the MPOA was built using single-cell resolution MERFISH data. In the original publication, single-cell RNA-seq and clustering analysis was first performed to identify cell-types in the MPOA. From there, a panel of 135 genes was specifically chosen to optimally distinct between 7 major classes of non-neuronal cell-types in addition to excitatory and inhibitory neuronal subtypes. Spatially resolved MERFISH data was then collected for this select panel of 135 genes and clustering analysis was subsequently performed to identify 9 major cell-types. In this manner, the ground truth cell-types were determined based on the original publication’s clustering analysis. We now clarify this point on lines 95-105 and provide the relevant text below for the reviewer’s reference.

We simulated ST data by aggregating the gene expression of cells from single-cell resolution multiplex error-robust fluorescence *in situ* hybridization (MERFISH) data of the mouse medial pre-optic area (MPOA)⁸ within spatially contiguous pixels. Previously, MERFISH was previously applied to map the spatial distribution of 135 select genes within MPOA brain tissue. These select 135 genes were chosen to distinguish between major non-neuronal cell-types as well as neuronal subtypes. Imaging-based cell segmentation was performed and the counts of genes per cell were quantified to achieve single-cell resolution spatially resolved transcriptomic profiling. Subsequent transcriptional clustering analysis on the single-cell resolution gene expression measurements identified 9 major cell-types, including excitatory and inhibitory neurons.

Minor Comment 1-1: Title is slightly misleading - should be ‘multi-pixel’ (or similar), not pixel.

We thank the reviewer for the suggestion, and we have changed the title to: “Reference-free cell-type deconvolution of multi-cellular pixel-resolution spatially resolved transcriptomics data”.

Minor Comment 1-2: Figure 1D LDA modeling should be improved.

We thank the reviewer for the suggestion. We have expanded upon the LDA modeling panel in Figure 1. Each parameter in the LDA plate model is now described in terms of what it represents with respect to the input ST dataset. Further, parameter symbol assignments have been changed to connect the LDA model diagram with features of the ST data. We have provided the revised Figure 1 for the reviewer’s reference.

Figure 1. Overview of STdeconvolve. A) STdeconvolve takes as input a spatial transcriptomics (ST) gene counts matrix of D pixels (rows) by N genes (columns). A matrix of spatial coordinates for each of the D pixel can also be used for visualization. B) STdeconvolve first feature selects genes for deconvolution, such as genes with counts in more than 5% and less than 95% of the pixels, and overdispersed across the pixels. STdeconvolve then guides the selection of the optimal number of cell-types to be deconvolved, K . STdeconvolve finally applies LDA modeling. A graph representation of LDA modeling and the parameters to be learned is shown. Shaded circle indicates observed variables and clear circles indicate latent variables. C) STdeconvolve outputs two matrices: (1) β , the deconvolved transcriptional profile matrix of K cell-types over N' feature selected genes, and (2) θ , the proportions of K cell-types across the D pixels. The proportion of deconvolved cell-types can then be visualized across the pixels.

Minor Comment 1-3: Figure 2D is too small and pixelated to clearly see what is written on the top left.

We agree with the reviewer and have increased the font sizes and resolution to improve visibility.

Figure 2D. Deconvolution of simulated ST data. D) Ground truth cell-types are ordered by their frequencies in the ground truth dataset. Matched deconvolved and ground truth cell-types are boxed.

Minor Comment 1-4: Not sure what Fig2 G and H demonstrate. Maybe clarify.

We thank the reviewer for raising this point of clarification. These figures were meant to provide evidence that deconvolved cell-type X7 of the MOB ST dataset may correspond to the neuronal precursor cell-type within the RMS unidentified from clustering analysis. This was supported by the fact that top expressed genes in cell-type X7's deconvolved transcriptional profile included *Nrep*, *Sox11*, and *Dcx*, which are known to be associated with neuronal differentiation and upregulated in neuronal precursor cells within the RMS⁹. Figure 2G indicated that *Sox11* and *Nrep* were both expressed in the central region of the granule cell layer where cell-type X7 was predicted to be. Figure 2H presented independent and orthogonal evidence of higher resolution ISH staining of these genes in a region within the granule cell layer where the RMS is expected⁵. These figures have now been moved to Figure 4C and D in the revised manuscript. Further, these points have been clarified in the main results (Lines 257-265) and we have also provided below for the reviewer's reference.

In particular, deconvolved cell-type X7 overlapped with the granule cell layer previously identified from clustering analysis and was spatially placed where the RMS is expected¹⁰ (Figure 4B). Upregulated genes in its deconvolved transcriptional profile, including *Nrep*, *Sox11*, and *Dcx*, are known to be associated with neuronal differentiation and upregulated in neuronal precursor cells within the RMS⁹ (Figure 4C, Supplementary Figure S6E). Higher resolution ISH staining of these genes further demarcates a region within the granule cell layer where the RMS is expected⁵ (Figure 4D). This suggests that deconvolved cell-type X7 may correspond to the neuronal precursor cell-type within the RMS unidentified from clustering analysis.

Figure 4B-D. Deconvolution of ST data of varying resolution from multiple technologies by STdeconvolve. B) Highlight of deconvolved cell-type X7. Pixel proportion of deconvolved cell-type X7 are indicated as black slices in pie charts. Pixels are outlined with colors as in A). C) Gene counts in each pixel of the MOB ST dataset for deconvolved cell-type X7's select top marker genes *Sox11* and *Nrep*. D) Corresponding ISH images for *Sox11* and *Nrep* from the Allen Brain Atlas⁵.

Minor Comment 1-5: When comparing to available ground truths - it would be very interesting to demonstrate your performance and sensitivity to different 'region' size selections for the same data. Are you more accurate with higher resolutions?

We thank the reviewer for the suggestion and interesting idea. We have now evaluated the performance of STdeconvolve to different region sizes represented by different numbers of pixels. We note that the deconvolution accuracy does depend on the number of pixels in the ST dataset. We found that deconvolution accuracy generally decreased when the input dataset was less than 10 pixels, suggesting that STdeconvolve may not perform well for region sizes covering fewer than 10 pixels. While we have generally found the number of pixels in most ST datasets to be well beyond 10 pixels after quality control filtering, the application of ST to profile tissue slivers or other thin structures covering only a few pixels may present challenges to deconvolution by STdeconvolve. These thoughts have been added to the Discussion (Lines 453-459) and we have also provided below for the reviewer's reference.

Notably, the performance of LDA in accurately deconvolving cell-types depends on the size of the dataset with respect to the number of pixels and the number of genes¹¹. As such, deconvolution accuracy generally decreases for ST data containing fewer than 10 pixels (Supplementary Figure S19). While we have generally found the number of pixels in most ST datasets to be well beyond 10 pixels after quality control filtering, the application of ST to profile tissue slivers or other thin structures covering only a few pixels may present challenges to deconvolution by STdeconvolve.

Supplementary Figure S19. Accuracy of deconvolution by STdeconvolve based on the number of pixels in the input dataset. Each point is a replicate in which a random sample of pixels from the simulated MERFISH MPOA ST was used as the input into STdeconvolve. For each replicate, the mean RMSE was computed by averaging the RMSEs of the difference in deconvolved cell-type and ground truth cell-type proportions for each pixel. The solid line represents the mean RMSE of the replicates for each given number of pixels sampled.

To evaluate whether STdeconvolve is more accurate at higher resolutions, we have also included simulated ST data of the MPOA at increasing pixel resolutions including 100, 50, 20, and 10 μm^2 . Using the 9 previously annotated major cell-types of the single-cell MERFISH data as ground truth, we deconvolved $K=9$ cell-types for each resolution dataset and computed the RMSE for each pixel. We then compared the RMSE distributions for each pixel resolution. In general, the deconvolution accuracy of STdeconvolve stays the same, suggesting that our deconvolution approach is robust at higher pixel resolutions. This analysis has been added to the main Results (Lines 327-335) and we have also provided text below for the reviewer's reference.

First, to simulate higher resolution ST data, we again aggregated single-cell resolution MERFISH data of the MPOA into 50, 20, and 10 μm^2 resolution pixels. Applying STdeconvolve, we observed similarly strong correlations between the deconvolved cell-type transcriptional profiles and proportions with the ground truth (Supplementary Figure S11A-D). Although the number of cells in each multi-cellular pixel did decrease as the resolution of the pixel increased as expected, we note that even higher resolution pixels may still contain multiple cells representing multiple cell-types (Supplementary Figure S11E-F). Thus, deconvolution may still be applicable to higher resolution ST data and STdeconvolve can still accurately deconvolve cell-types within these higher resolution multi-cellular pixels.

Reviewer 2

Overview 2-0: The transcript from Miller et al. presents a spatially resolved transcriptomic cell type deconvolution approach that unlike related methods does not rely on a single-cell transcriptomic reference. The approach is based on latent Dirichlet allocation (LDA) that is typically used for topic modelling in text mining applications. Although LDA has been applied for cell type identification in single-cell RNA sequencing data, the application of LDA to deconvolve cell types in multicellular pixels of spatial transcriptomic data is elegant and to the extent that this reviewer is aware original. The authors combined previously published approaches and R functions (such as the identification of overdispersed genes and LDA) into a new R package for users to apply STdeconvolve to spatial transcriptomic data. The authors should be commended for a well-documented R package including example data. This reviewer had no problem installing and using the R package based on the available vignettes.

We thank the reviewer for their kind words and are thrilled they had no issues installing and running the R package.

Overview 2-1: The authors further used previously published data sets to evaluate their approach and to compare it to two related methods that rely on single-cell transcriptomic references (SPOTlight and RCTD). Further, the authors highlighted the advantage of a reference-free approach applied to cancer samples without adequate single-cell references due to substantial inter-tumor heterogeneity. To demonstrate the application of STdeconvolve to cancer samples, the authors used spatial transcriptomic data of four breast cancer sections and drew a bold conclusion concerning possible clinical relevance. Despite the apparent advantages of reference-free cell type deconvolution, the authors did not sufficiently benchmark their method compared to the range of existing methods. They also did not mention relevant aspects like computational performance and scaling. There are a few recently published methods that are relevant to this work that were not considered or referenced by the authors.

Major Comment 2-1: How does STdeconvolve compare to the recently published SpatialDWLS? This publication was not referenced by the authors even though it is highly related and includes a much more comprehensive benchmarking to existing methods than is provided by the authors for STdeconvolve.

We thank the reviewer for raising this important point about additional benchmarking, particularly against other highly related approaches. We have now included spatialDWLS as well as another recently published deconvolution approach DSTG in addition to SPOTlight and RCTD as part of the comparison between STdeconvolve and other supervised deconvolution approaches. We note that like SPOTlight and RCTD, spatialDWLS and DSTG are also reference-based deconvolution approaches.

To compare STdeconvolve with these reference-based deconvolution approaches, we again used our simulated 100 μm^2 pixel resolution ST data using our single-cell resolution MERFISH data of the MPOA. The ground truth cell-type proportions of each multi-cellular pixel was based on the 9 major cell-types previously annotated in the original MERFISH publication. We quantitated the deconvolution performance of the different deconvolution approaches by calculating the Root-Mean-Squared-Error (RMSE) of a given approach's predicted cell-type proportions for each pixel in a manner similar to how benchmarking was done in the spatialDWLS paper. With respect to these reference-based deconvolution approaches however, a single-cell transcriptomics reference was required for deconvolution. We therefore trained these approaches using three different sets of references and compared the results to our reference-free approach.

The first reference used was the original single-cell MERFISH data used to construct the simulated ST data, which optimally matched the ST dataset and served as an upper bound of performance. The second set of references was the single-cell MERFISH data but with missing cell-types. The third was an independent scRNA-seq reference, more akin to what would typically be available for such deconvolution scenarios. When we use the ideal reference, supervised approaches RCTD and spatialDWLS had slightly smaller RMSEs than STdeconvolve, however, STdeconvolve had a lower RMSE than SPOTlight. DSTG was unable to recover distinct cell-types and was omitted from further analysis. When we use less ideal single cell transcriptomics references such as references with missing cell-types or an independent scRNA-seq reference, the RMSE of all reference-based deconvolution approaches

significantly increased and were higher than that of STdeconvolve. This suggests that STdeconvolve provides potentially superior performance when suitable single-cell references are not available. We have added these findings to the Results under section “**STdeconvolve achieves competitive performance to reference-based, supervised deconvolution approaches**” (Lines 124-169) and have provided the relevant text below for the reviewer’s reference.

We next sought to compare the performance of STdeconvolve to existing reference-based, supervised and semi-supervised deconvolution approaches SPOTlight, RCTD, spatialDWLS, and DTSG using our simulated 100 μm^2 resolution ST data of the MPOA. As described previously, these approaches require a single-cell transcriptomics reference for deconvolution. As an ideal single-cell transcriptomics reference, we used the original single-cell MERFISH data that was used to construct the simulated ST data (Supplementary Figure S3A, Supplementary Methods). We again quantified the performance of each approach using the RMSE of the deconvolved cell-type proportions compared to ground truth across simulated pixels. DTSG was unable to deconvolve distinct cell-types in the data and was omitted from further comparison (Supplementary Figure S3A). In general, we find the performance of STdeconvolve to be comparable to these reference-based deconvolution approaches when such an ideal single-cell transcriptomics reference is used (Figure 2E-F).

One potential limitation of such existing reference-based deconvolution approaches is their reliance on a suitable single-cell transcriptomics reference. We thus sought to evaluate the performance of these reference-based deconvolution approaches when a suitable single-cell reference is not available. To this end, we removed excitatory and inhibitory neuronal cell-types to simulate a less suitable single-cell transcriptomics reference (Supplementary Methods). We then deconvolved the simulated ST data of the MPOA using each reference-based deconvolution approach with this new reference and computed the RMSE across pixels. Because STdeconvolve does not use a reference, its performance does not change. However, the performance for all reference-based deconvolution approaches resulted in a significantly higher RMSE (Diebold-Mariano p -value $< 2.2 \times 10^{-16}$) than STdeconvolve (Figure 2G). Likewise, pixels previously comprised of neurons were now erroneously predicted by reference-based deconvolution approaches to be comprised primarily of immature oligodendrocytes (Supplementary Figure S3B). In addition, we evaluated the performance of each reference-based deconvolution approach after removing rarer ependymal cells from the single-cell transcriptomics reference. Again, given this less suitable single-cell transcriptomics reference, pixels previously comprised of ependymal cells were now erroneously predicted by reference-based deconvolution approaches to be comprised primarily of astrocytes (Supplementary Figure S3C). Thus, the performance of reference-based deconvolution approaches is sensitive to differences in cell-type composition between the ST data and the single-cell transcriptomics reference used.

Likewise, such an ideal single-cell transcriptomics reference that optimally matches the cell-type composition and measurement sensitivities of the ST data to be deconvolved may not be available. Therefore, this ideal MERFISH MPOA single-cell transcriptomics reference likely provides an upper bound on performance for reference-based deconvolution approaches. To provide a more realistic evaluation of performance for reference-based deconvolution approaches, we sought to deconvolve our simulated ST data of the MPOA using a scRNA-seq reference from a mouse brain atlasing effort¹². Again, as a reference-free deconvolution approach, the performance of STdeconvolve does not change. However, again, the performance for all reference-dependent methods resulted in a significantly higher RMSE (Diebold-Mariano p -value $< 2.2 \times 10^{-16}$) than STdeconvolve (Figure 2H, Supplementary Methods). Thus, STdeconvolve achieves comparable performance to reference-based, supervised deconvolution approaches when an ideal single-cell transcriptomics reference is used, and potentially better performance when an ideal single-cell transcriptomics reference is not used.

Figure 2E-H. Deconvolution of simulated ST data. E) Root-mean-square-error (RMSE) of the deconvolved cell-type proportions compared to ground truth for STdeconvolve, F) for supervised deconvolution approaches using the ideal single cell transcriptomics MRFISH MPOA reference, G) for supervised deconvolution approaches using the single cell transcriptomics MRFISH MPOA reference with missing neurons, and H) for supervised deconvolution approaches using a brain single-cell RNA-seq reference.

Supplementary Figure S3. Cell-type predictions of supervised and semi-supervised deconvolution approaches across simulated MERFISH MPOA ST dataset pixels with different single-cell transcriptomics references. A) Deconvolved pixel proportions of cell-types by each supervised and semi-supervised deconvolution approach using

the MERFISH MPOA single-cell transcriptomics data as the reference. B) Deconvolved pixel proportions of cell-types by each supervised and semi-supervised deconvolution approach using the MERFISH MPOA single-cell transcriptomics data with missing neurons as the reference. C) Deconvolved pixel proportions of cell-types by each supervised and semi-supervised deconvolution approach using the MERFISH MPOA single-cell transcriptomics data with missing ependymal cells as the reference.

Major Comment 2-2: Can the authors provide a similarly comprehensive benchmarking that includes more existing methods and also considers computational performance and scaling with number of pixels?

We thank the reviewer for the suggestion and have now included additional approaches for which STdeconvolve was benchmarked against, which include spatialDWLS as an additional supervised approach along with SPOTlight and RCTD, as well as DSTG as a semi-supervised approach in the Results under section “**STdeconvolve achieves competitive performance to reference-based, supervised deconvolution approaches**” (Lines 124-169).

Additionally, to assess the scalability of STdeconvolve, we have now assessed both the runtime and total memory usage of STdeconvolve in terms of varying the number of genes and number of pixels. We find that the runtime increases linearly with either the number of pixels or genes, as well as the number of cell-types to be deconvolved. As a specific example, deconvolution of 12 cell-types using all 2072 pixels and 1000 feature selected genes in a typical deconvolution analysis of a 10X Visium dataset completed in approximately 30 minutes on a personal computer, which we believe is a reasonable time cost and is comparable with previously published deconvolution approaches⁷. Further, we now allow for parallelization, which may further improve scalability. Importantly, the total memory cost was also minimal, with total memory reaching approximately 170 MBs for the full sized 10X Visium analysis. These results are now presented as Supplementary Figure S15 and we have provided the relevant text (Lines 373-384) below for the reviewer’s reference.

As the resolution of ST data improves, the number of spatially resolved pixels and cell-types represented in the data will presumably also increase. We therefore sought to evaluate the scalability of STdeconvolve in anticipation of these increasingly larger datasets. To this end, we benchmarked the runtime and total memory usage by STdeconvolve when deconvolving varying numbers of cell-types using varying numbers of genes across varying numbers of pixels (Methods). We found that both the runtime and memory usage by STdeconvolve increased linearly with the number of pixels and genes in the input dataset (Supplementary Figure S15A) and is comparable to existing reference-based deconvolution methods when applied to current ST datasets⁷. Likewise, runtime scales with the number of deconvolved cell-types K in the input dataset though memory usage remains stable (Supplementary Figure S15B). To enhance runtime efficiency, STdeconvolve has built in parallelization. In this manner, we anticipate that STdeconvolve will be amenable to larger ST data.

Supplementary Figure S15. Runtime and memory usage by STdeconvolve. A) Runtime as a function of dataset size in terms of the number of pixels (top) or genes (bottom), and number of cell-types K . For scaling pixels, the top 1000 most significant overdispersed genes were selected and kept constant. For scaling genes, all 2702 pixels in the dataset were used and kept constant. B) Memory usage as a function of dataset size in terms of scaling pixels (top) or genes (bottom), and number of cell-types K . Again, for scaling pixels, the top 1000 most significant overdispersed genes were selected and kept constant. For scaling genes, all 2702 pixels in the dataset were used and kept constant.

Major Comment 2-3: How does STdeconvolve compare to the recently published spatial spot deconvolution method BayesSpace? Like STdeconvolve, BayesSpace does not require a single-cell reference and has the further advantage of providing subspot spatial resolution allowing for subsequent spatial analyses.

We thank the reviewer for the suggestions and have now included a comparison between STdeconvolve and BayesSpace. While BayesSpace does provide enhanced resolution clustering without a reference, we note that BayesSpace is a clustering approach and that deconvolution approaches in general may provide distinct insights from clustering analyses. To demonstrate this, we have now included a new analysis using simulated ST data comprised of mixtures of either luminal cells and pericytes or pericytes and macrophages (Figure 3D, Methods). Applying clustering analysis to these ST pixels, we identified 2 clusters corresponding to either mixtures of luminal cells and pericytes or mixtures of pericytes and macrophages (Figure 3E). In contrast, applying STdeconvolve with $K=3$, we were able to recover the proportional representations of luminal cells, pericytes, and macrophages as well as their original cell-type specific transcriptomic profiles (Figure 3F). Applying BayesSpace at enhanced resolution with 3 clusters to the same simulation also resulted in 3 spatially discrete clusters but the RMSE was significantly higher than STdeconvolve. Such differences between deconvolution and clustering analysis can extend to real tissue settings. We generated a new simulated ST dataset of a coronal section of the mouse cortex at $100 \mu\text{m}^2$ pixel resolution using a single-cell MERFISH dataset recently released by VizGen¹³. Individual cells in this dataset were transcriptionally clustered into 20 distinct groups, which served as ground truth labels and demarcated known cell-layers in the cortex. We focused on the thalamus, which is a region comprised of mixtures of multiple distinct cell-types. We note a visual difference between STdeconvolve and BayesSpace at enhanced clustering at the thalamus, and quantification confirmed a significantly higher RMSE for BayesSpace than STdeconvolve. Thus, deconvolution by STdeconvolve is able to provide distinct insights from clustering analyses and potentially higher accuracy particularly in tissue regions with heterogeneous and non-continuous mixtures of cell-types. We have included these findings in a new section called **“Deconvolution provides distinct insights compared to clustering analysis”** (Lines 203-244) and have provided the relevant text below for the reviewer’s reference.

Generally, we note that deconvolution of multi-cellular pixel resolution ST data can provide distinct insights from clustering analysis. To demonstrate this, we again simulated ST data using mixtures of single cells assayed by scRNA-seq (Supplementary Methods). Specifically, we simulated ST pixels comprised of mixtures of either luminal cells and pericytes or pericytes and macrophages (Figure 3D). Applying clustering analysis to these ST pixels, we identified 2 clusters corresponding to either mixtures of luminal cells and pericytes or mixtures of pericytes and macrophages (Figure 3E). In contrast, applying STdeconvolve with $K=3$, we were able to recover the proportional representations of luminal cells, pericytes, and macrophages as well as their original cell-type specific transcriptional profiles (Figure 3F).

Such differences between deconvolution and clustering analysis extends to resolution enhancing clustering approaches such as BayesSpace¹⁴. Briefly, BayesSpace utilizes a spatial prior that encourages spatially neighboring pixels to cluster into the same transcriptional cluster. Enhanced resolution clustering is obtained after subdividing each pixel and modeling the expression profiles of the subpixels as additional latent parameters estimated in the Bayesian model. Applying BayesSpace with 3 clusters to our simulated ST data, we obtained 3 spatially discrete clusters corresponding to different mixtures of luminal cells and pericytes and mixtures of pericytes and macrophages (Figure 3G, Supplementary Methods). Compared to STdeconvolve, both regular transcriptional clustering and resolution enhanced clustering with BayesSpace exhibited significantly higher RMSE (Diebold-Mariano p -value $< 2.2 \times 10^{-16}$) (Figure 3H).

Figure 3D-H. Comparing clustering versus deconvolution analysis for ST data. D) Simulated ST dataset with 3 cell-types represented as pie charts for each simulated ST pixel. E) Clustering analysis results of simulated ST dataset with 3 cell-types. Pie chart proportional representation (left) and tSNE representation (right). F) Deconvolution results for the simulated ST dataset with 3 cell-types by STdeconvolve. The ranking of each gene based on its expression level in the deconvolved-cell-type transcriptional profiles compared to its gene rank in the matched ground truth cell-type transcriptional profiles (top). Heatmap of Pearson’s correlations between the deconvolved cell-types proportions and ground truth cell-types proportions across simulated pixels (bottom). G) BayesSpace enhanced resolution clustering results for the simulated ST dataset with 3 cell-types represented as pie charts. H) Root-mean-square-error (RMSE) of the deconvolved cell-type proportions compared to ground truth for the simulated ST dataset with 3 cell-types.

To further demonstrate the difference between deconvolution and clustering analysis for ST data, we again simulated ST data using a single-cell resolution MERFISH dataset of a coronal section of the mouse brain¹³. We analyzed the single-cell resolution transcriptional profiles to identify 20 transcriptionally distinct cell-types and again simulated multi-cellular pixel-resolution ST data by aggregating the single cells into $100 \mu\text{m}^2$ pixels (Figure

3I, Supplementary Figure S5A-B, Supplementary Methods). The organization of cell-types within the mouse brain is highly complex with many regions including the thalamus at the central region of this coronal section being composed of mixtures of multiple transcriptionally distinct cell-types. We thus sought to evaluate whether STdeconvolve could better recover the proportional representation of cell-types compared to resolution enhanced clustering with BayesSpace. Applying both STdeconvolve and BayesSpace, we generally recover the cell-type pixel proportions and visually recapitulate the spatial organization of cell-types within various brain structures (Figure 3J-K, Supplementary Methods). However, focusing in on the central region of the coronal section encompassing the thalamus, we indeed saw a visual difference between the spatial organization of cell-types recovered by deconvolution via STdeconvolve compared to resolution enhanced clustering via BayesSpace (Figure 3J-K inset). Quantifying performance, BayesSpace exhibited significantly higher RMSE compared to STdeconvolve (Diebold-Mariano p -value $< 2.2 \times 10^{-16}$) as a whole (Supplementary Fig S5C, though more discernably in the thalamus region (Figure 3L). Taken together, deconvolution approaches such as STdeconvolve can provide distinct results from clustering and resolution enhanced clustering approaches when applied to multi-cellular pixel resolution data.

Figure 3I-L. Comparing clustering versus deconvolution analysis for ST data. I) Ground truth cell-type proportions derived from single-cell resolution MERFISH data of the mouse brain partitioned into $100 \mu\text{m}^2$ pixels. J) Deconvolved cell-type proportions for the mouse brain by STdeconvolve. K) Enhanced resolution clustering for the mouse brain by BayesSpace. Inset highlights an interior region corresponding approximately to the thalamus. L) Root-mean-square-error (RMSE) of the deconvolved cell-type proportions compared to single-cell clustering for the MERFISH mouse brain data for the inset interior region corresponding approximately to the thalamus.

Supplementary Figure S5. Single-cell transcriptional clustering of MERFISH data of a coronal section of the mouse brain. A) UMAP embedding of single cells colored by transcriptional cluster assignment. B) Spatial positions of cells colored by transcriptional cluster assignment. C) Root-mean-square-error (RMSE) of the deconvolved cell-type proportions compared to single-cell clustering for the MERFISH mouse brain data.

Major Comment 2-4: How significant do the authors expect their method to be in light of continuously improving ST methods with higher resolution? For example, DBiT-seq has a smallest achievable pixel size of about $5\mu\text{m}$ and recently published Seq-Scope achieves subcellular resolution already. Could such pixel deconvolution methods still be useful in such a scenario?

We thank the reviewer for raising this important point about the validity of deconvolution methods applied to recent advances in ST technologies profiling tissues at higher pixel resolutions. While the number of cells in each multi-cellular pixel does decrease at higher resolutions, we note that even higher resolution pixels may still contain multiple cells representing multiple cell-types, even at $10\mu\text{m}^2$, thus necessitating the need for deconvolution approaches (Supplementary Figure S11E-F). Nonetheless, it is still important to assess our approach across a range of different resolutions and technologies. To this end, we have now applied STdeconvolve to newer ST technologies as the reviewer has suggested, which include 10X Visium at $50\mu\text{m}^2$ pixel resolution, DBiT-seq at $25\mu\text{m}^2$ pixel resolution, and Slide-seq at $10\mu\text{m}^2$ pixel resolution. We show that our deconvolution analysis is able to recapitulate previously reported findings and identify known biological features, indicating that STdeconvolve is applicable across different high resolution ST technologies. The results are presented in a new section called “**STdeconvolve is applicable across diverse ST dataset resolutions and technologies**” (Lines 320 to 384) in the revised manuscript, with relevant text provided below for the reviewer’s reference.

We anticipate that continual technological improvements will enhance the resolution of ST data. Already, ST technologies such as Visium (10X Genomics), Slide-seq², and DBiT-seq³ have achieved resolution that can range from $50\mu\text{m}^2$ to $10\mu\text{m}^2$. Therefore, we sought to evaluate the performance of STdeconvolve on higher resolution ST data using both simulated as well as real ST data from higher resolution ST technologies including Visium, Slide-seq, and DBiT-seq.

First, to simulate higher resolution ST data, we again aggregated single-cell resolution MERFISH data of the MPOA into 50 , 20 , and $10\mu\text{m}^2$ resolution pixels. Applying STdeconvolve, we observed similarly strong correlations between the deconvolved cell-type transcriptional profiles and proportions with the ground truth (Supplementary Figure S11A-D). Although the number of cells in each multi-cellular pixel did decrease as the resolution of the pixel increased as expected, we note that even higher resolution pixels may still contain multiple cells representing multiple cell-types (Supplementary Figure S11E-F). Thus, deconvolution may still be applicable to higher resolution ST data and STdeconvolve can still accurately deconvolve cell-types within these higher resolution multi-cellular pixels.

Supplementary Figure S11. Comparison of STdeconvolve cell-types to ground truth cell-types in high resolution pixel MERFISH MPOA ST datasets. A) Deconvolution accuracy in 50 μm^2 simulated pixels. Left: The ranking of each gene based on its expression level in the transcriptional profiles of the deconvolved cell-types, compared to its gene rank in the transcriptional profile of the matched ground truth cell-type. Middle: Pearson's correlation between the transcriptional profiles of the 9 ground truth cell-types in MERFISH MPOA data and 9 deconvolved cell-types by STdeconvolve. Right: Pearson's correlation between the pixel proportions of the 9 ground truth cell-types in the MERFISH MPOA data and 9 deconvolved cell-types by STdeconvolve. B) Deconvolution accuracy in 20 μm^2 simulated pixels. C) Deconvolution accuracy in 10 μm^2 simulated pixels. D) Root-mean-square-error (RMSE) of the deconvolved cell-type pixels proportions compared to ground truth cell-types at each resolution of the simulated ST dataset. E) Distributions of number of individual cells contained within pixels of each simulated

resolution of the MERFISH MPOA ST dataset. Numbers above each dataset indicate the total number of pixels for each simulated ST data. F) Stacked bar plots showing the fraction of pixels that contain a distinct number of different cell-types for each simulated resolution of the MERFISH MPOA ST dataset. This was done for pixels that contain at 2 or more individual cells. Numbers within each box indicate the total number of pixels with that many distinct cell-types and number above the bars indicate the total number of pixels with 2 or more individual cells.

Encouraged by STdeconvolve's ability to recover cell-types in simulated high-resolution ST data, we then applied STdeconvolve to real high-resolution multi-cellular ST data from several different technologies. First, we applied STdeconvolve to $50 \mu\text{m}^2$ resolution ST data of a coronal section of the mouse brain from 10X Visium⁴. Briefly, for 10XVisium, mRNAs from tissue sections are captured onto an array of DNA barcoded spots, resulting in RNA-sequencing measurements with gridded 2D spatial positional information. We applied STdeconvolve to identify $K=20$ cell-types that exhibit spatially distinct patterns that demarcate known brain structures such as the isocortex and fiber tracts (Figure 4E, Supplementary Figure S12, Supplementary Methods).

Figure 4E. Deconvolution of ST data of varying resolution from multiple technologies by STdeconvolve. E) Deconvolved cell-type proportions for Visium data of the mouse brain.

Supplementary Figure S12. STdeconvolve pixel proportion of select deconvolved cell-types in 10X Visium data of the mouse brain. A) STdeconvolve pixel proportions for select deconvolved cell-types. B) Visually corresponding annotated brain region from the Allen Brain Atlas⁵.

We next applied STdeconvolve to $25 \mu\text{m}^2$ resolution ST data of the lower body of the E11 mouse embryo from DBiT-seq. Briefly, for DBiT-seq, parallel microfluidic channels are used to deliver DNA barcodes to the surface of a tissue to enable direct barcoding of mRNAs *in situ*, resulting in RNA-sequencing measurements in a 2D mosaic of spatial pixels. Previously, the authors identified 13 transcriptionally and spatially distinct features in the E11 mouse embryo including the atrium, ventricle, liver, and blood vessels containing erythrocyte coagulation. Applying STdeconvolve with $K=13$, we identify deconvolved cell-types that corresponded with similar spatially distinct features in agreement with the original findings (Figure 4F, Supplementary Figure S13A, Supplementary Methods). Moreover, the top genes in the deconvolved cell-type specific transcriptional profiles contained the expected marker genes of the matching features, such as *Myh6* for the atrium, *Myh7* for the ventricle, *Apoa2* for the liver, and *Hba.a2* for the blood vessels containing erythrocyte coagulation in agreement with the original findings (Supplementary Figure S13B).

Figure 4F. Deconvolution of ST data of varying resolution from multiple technologies by STdeconvolve. F) Deconvolved cell-type proportions for DBiT-seq data of the lower body of an E11 mouse embryo.

Supplementary Figure S13. STdeconvolve pixel proportions and transcriptional profiles of select deconvolved cell-types in DBiT-seq data of an E11 mouse embryo lower tail section. A) Pixel proportions of select deconvolved cell-types X7, X11, X1, and X3 corresponding to the atrium, ventricle, fetal liver, and erythrocyte

coagulation, respectively, annotated in the previous publication³. B) Deconvolved transcriptional profiles for cell-types X7, X11, X1, and X3 with the expression of the top gene in each deconvolved transcriptional profile visualized in the original tissue (inset).

Finally, we applied STdeconvolve to 10 μm^2 resolution ST data of the mouse cerebellum from Slide-seq. Briefly, for Slide-seq, mRNAs from tissue sections are captured onto densely packed barcoded beads, resulting in RNA-sequencing measurements with 2D spatial positional information. Previously, RCTD was also applied to this Slide-seq dataset with a matched Drop-seq scRNA-seq reference of the mouse cerebellum⁶ to identify beads representing a distinct layers of Purkinje neurons and Bergmann glia. Applying STdeconvolve, we identified $K=14$ cell-types (Figure 4G, Supplementary Methods) whose transcriptional profiles correlated strongly with cell-types from the scRNA-seq dataset of the mouse cerebellum (Supplementary Figure S14A). In particular, we found that the deconvolved transcriptional profiles of cell-type X4 and cell-type X2 correlated strongly with the transcriptional profiles of Purkinje neurons and Bergmann glia. Likewise, the deconvolved proportional representation of cell-type X4 and cell-type X2 also agreed significantly (Fisher's Exact p -value $< 2.2 \times 10^{-16}$) with the predicted proportions of Purkinje neuron and Bergmann glia from RCTD (Supplementary Figure S14B-C). Taken together, these results indicate that STdeconvolve can be applicable to a range of multi-cellular resolution ST technologies.

Figure 4G. Deconvolution of ST data of varying resolution from multiple technologies by STdeconvolve. G) Deconvolved cell-type proportions for Slide-seq data of the mouse cerebellum.

Supplementary Figure S14. STdeconvolve predicted cell-type proportions of Purkinje neurons and Bergmann glia in Slide-seq data of the mouse cerebellum compared to RCTD. A) Pearson’s correlation between the deconvolved transcriptional profiles of the 14 STdeconvolve cell-types and best matched cell-types in the mouse cerebellum scRNA-seq reference⁶. B) Beads predicted to contain Purkinje neurons (cyan) or Bergmann glia (magenta) by either RCTD (left) or STdeconvolve (right). C) Stacked bar plots indicating the fractions of beads predicted to contain Purkinje neurons (left) or Bergmann glia (right) by RCTD and STdeconvolve.

Second, as mentioned previously, we would like to emphasize that deconvolution provides distinct insights from clustering analysis. Given that even at high resolutions, pixels may contain multiple cells comprising multiple cell-types, we now include simulations to demonstrate that clustering multi-cellular pixels can give differing results than deconvolution. Such differences between deconvolution and clustering analysis extends to resolution enhancing clustering approaches as well as real tissue settings. We have included these results in a new section called **“Deconvolution provides distinct insights compared to clustering analysis”** (Lines 203 to 244) in the revised manuscript.

Finally, we acknowledge that as the resolution of ST data increases even further, sub-cellular resolution ST data will become more accessible. Already, several sub-cellular resolution ST technologies have emerged as the reviewer has noted. However, as the capture efficiency at this resolution and likewise the biological questions of interest may differ substantially from multi-cellular ST data, we anticipate that new methods specifically suited for sub-cellular resolution ST data will be needed and thus STdeconvolve may indeed not be the most relevant for analysis of such sub-cellular resolution ST data. To this end, we have now updated the language to refer to “multi-cellular pixel resolution ST data” in order to distinguish from “subcellular pixel-resolution ST data” throughout the revised manuscript.

Still, moving forward, we anticipate that there will be a need to balance between resolution and throughput of ST datasets depending on the biological question of interest. Therefore, the comparative larger tissue region covered by multi-cellular pixel resolution ST data may still make such data useful and thus still require pixel deconvolution. We have now included a discussion of these thoughts in the revised manuscript (Lines 470-482), provided below for the reviewer’s reference.

Already, a number sub-cellular pixel-resolution ST technologies have emerged¹⁵⁻¹⁹. As the capture efficiency at this resolution and likewise the biological questions of interest may differ substantially from multi-cellular ST data, we anticipate that new methods specifically suited for sub-cellular resolution ST data will be needed. Thus, STdeconvolve may not be best suited to analysis of such sub-cellular resolution ST data. Still, as we have noted previously, even as the resolution of ST data increases, some pixels may still contain multiple cells representing multiple cell-types, suggesting that deconvolution may still be necessary. Likewise, the number of cells present in a pixel ultimately will depend on cell size, which can vary depending on the organism, tissue, and/or disease state being profiled. Ultimately, we believe that there will be a need to balance between resolution and throughput of ST technologies depending on the biological question of interest. The potentially larger tissue regions able to be covered by multi-cellular pixel resolution ST data may still be of interest and thus still require deconvolution.

Major Comment 2-5: The authors claim that they can identify potentially clinically relevant spatial features because they identified immune cells at the invasive breast cancer margin. How did the authors define the interface of the cancerous and non-malignant regions? How did they calculate an enrichment of immune cells at this interface?

We thank the reviewer for raising this point. Our observation of the deconvolved cell-type X15 at the interface of the cancerous and non-malignant tissue was solely based on visual inspection rather than a more quantitative definition of the cancerous and non-malignant interface. In the revised manuscript we have now revised to a more quantitative evaluation by testing for significant colocalization of deconvolved cell-type X15 with the pathologist annotations of the pixels. Specifically, we performed a one-sided Fisher's exact test to determine if a deconvolved cell-type was significantly enriched in pixels that were also labeled with a given pathological annotation. Additional details can be found in the revised Supplementary Methods in the section "**Enrichment of deconvolved cell-types in breast cancer pathological labels**". Overall, we found a significant the number of pixels with a high proportion of deconvolved cell-type X15 corresponding to invasive ductal carcinoma (IDC) regions (Fisher's exact p -value = 0.001257). In contrast, we do not see a significant number of pixels with a high proportion of deconvolved cell-type X15 corresponding to ductal carcinoma *in situ* (DCIS), regions (Fisher's exact p -value = 0.5625). This is consistent with previous observations that when comparing pure DCIS and IDC, infiltration of immune cells was significantly higher in IDC to pure DCIS^{20,21}. These findings are presented in the revised Results on lines 408-414 and presented below for the reviewer's reference.

Further, we find a significant the number of pixels with a high proportion of deconvolved cell-type X15 corresponding to IDC regions (Fisher's exact p -value = 0.001257) based on previous clustering and pathology annotations. In contrast, we do not see a significant number of pixels with a high proportion of deconvolved cell-type X15 corresponding to DCIS regions (Fisher's exact p -value = 0.5625). This is consistent with previous observations that when comparing pure DCIS and IDC, infiltration of immune cells was significantly higher in IDC to pure DCIS^{20,21}.

Clinically, the level of lymphocyte infiltration into the tumor mass is highly relevant but the simple presence of some immune cells adjacent to the tumor mass does not necessarily capture that information. Can the authors show that they can identify different levels of immune infiltration or differentiate tumors with an immune excluded vs infiltrated microenvironment? Further, can the authors based on STdeconvolve identify immunosuppressive environments based on the specific immune cell types that are present?

We thank the reviewer for this very great suggestion. To assess whether STdeconvolve can differentiate different levels of immune infiltration, we have now included a new analysis using simulated ST datasets that represent infiltrated and excluded immune cell spatial organizations using mixtures of single cells assayed by scRNA-seq. Briefly, we simulated ST data comprised of mixtures of luminal cells and immune cells in different spatial patterns that represent either immune infiltrated or immune excluded tumor microenvironments (Figure 5F, Supplementary Methods). We find that STdeconvolve can effectively recover the cell-type transcriptional profiles (Figure 5G) and cell-type proportions of immune infiltrates to help distinguish between infiltrated versus excluded spatial organization of immune cells (Figure 5H). We have included these findings in the revised Results section "**STdeconvolve identifies immune infiltrates in breast cancer**" (Lines 386-426) and have included the relevant text below for the reviewer's reference.

To evaluate whether STdeconvolve may be able to distinguish infiltrated versus excluded spatial organization of immune cells in tumors, we simulated ST data representing infiltrated and excluded spatial organizations using mixtures of single cells assayed by scRNA-seq (Figure 5F, Supplementary Methods). In both the simulated infiltrated and excluded cases, we find that STdeconvolve can effectively recover the cell-type transcriptional profiles (Figure 5G) and enable the quantification of immune infiltration to help distinguish between infiltrated versus excluded spatial organization of immune cells (Figure 5H).

Figure 5F-H. STdeconvolve characterizes the spatial organization of immune cells in real and simulated breast cancer ST data. F) Simulated ST datasets of an immune-excluded tumor sample (top) and immune-infiltrated tumor sample (bottom) using mixtures of single cells represented as pie charts for each simulated ST pixel. G) Deconvolution results for the simulated ST data by STdeconvolve. The ranking of each gene based on its expression level in the deconvolved-cell-type transcriptional profiles compared to its gene rank in the matched ground truth cell-type transcriptional profiles for the simulated immune-excluded tumor sample (top) and immune-infiltrated tumor sample (bottom). H) Histogram of the deconvolved proportion of immune cells in the tumor region defined in (F) for the simulated immune-excluded tumor sample (top) and immune-infiltrated tumor sample (bottom).

Finally, we acknowledge that our application of STdeconvolve to a cancer setting is currently limited to one real cancer ST dataset and therefore is not sufficient to conclude that STdeconvolve is broadly able to identify clinically relevant spatial features. We have therefore reworded our claims in the revised manuscript by suggesting that “STdeconvolve may be able to assist in deconvolving cell-types in heterogeneous cancer tissues to recover potentially clinically interesting spatial organizational patterns.” (Lines 425-426).

Minor Comment 2-1: For the proof of concept, previously annotated cell types from a published MERFISH dataset provided the ground truth. How much does the evaluation and comparison of these methods depend on the quality and granularity of these previously defined cell types?

We thank the reviewer for raising this point. For the MERFISH dataset, we note that in the original publication, a panel of 135 genes was specifically chosen that was used to optimally identify different subtypes of neurons. As such, only 33 remaining genes were available to distinguish between all major cell-types⁸. Therefore, in the context of feature selection in STdeconvolve, these chosen genes represent a specific subset of overdispersed genes rather than all overdispersed genes that are skewed towards distinguishing between neurons. Thus, neuronal subtypes may appear more transcriptionally distinct than certain cell-types for which few markers were included. This reason likely explains why 4 deconvolved cell-types X2, X4, X7, and X8 by STdeconvolve corresponded to ground truth neuronal cells and non-neuronal cells like pericytes and microglia were not readily deconvolved when $K=9$ cell-

types. However, when we increased the number of deconvolved cell-types to include the number of neuronal subtypes previously annotated in the data, we were able to identify deconvolved cell-types that were highly correlated in terms of both transcriptional profiles and pixel proportions to finer neuronal subtypes as well as rare cell-types such as pericytes and microglia (Supplementary Figure 21B-C). Thus, the ability for STdeconvolve to deconvolve neuronal subtypes may not be representative of its expected performance in distinguishing between cell sub-types under less biased gene selection. Likewise, a less biased gene selection may have resulted in stronger deconvolution of non-neuronal cell-types initially, such as pericytes and microglia. These thoughts have been included as part of “**Supplementary Note 3**” (Lines 145-179 in Supplementary Materials) and relevant excerpts are included below for the reviewer’s reference.

For this MERFISH dataset, a specific panel of 135 genes previously chosen to optimally distinguish primarily between neuronal subtypes. As such, only 33 remaining genes were available to distinguish between all major cell-types⁸. Therefore, these chosen genes represent a specific subset of overdispersed genes rather than all overdispersed genes that are skewed towards distinguishing between neurons, even though many more overdispersed genes may exist for distinguishing microglia and pericytes, for example. In this manner, neuronal subtypes may appear more transcriptionally distinct than certain cell-types for which few markers were included. Indeed, when we applied STdeconvolve to the simulated MERFISH MPOA ST data limited to only these 135 genes, deconvolved cell-types such as cell-types X2 and X8 both matched to excitatory neurons while cell-types X4 and X7 both matched to inhibitory neurons. Given that excitatory and inhibitory neurons could previously be further subdivided into finer neuronal sub-types, we sought to evaluate whether these deconvolved cell-types that matched to the excitatory and inhibitory neurons could represent these additional finer neuronal sub-types. To test this hypothesis, we further partitioned the ground-truth excitatory and inhibitory cell-types into additional sub-types based on previous annotations, resulting in 76 total non-neuronal cell-types and neuronal subtypes⁸. Comparing the deconvolved transcriptional profiles of X2, X4, X7, and X8 to the ground truth transcriptional profiles of the 76 non-neuronal cell-types and neuronal subtypes, we indeed observed a correlation between the deconvolved transcriptional profiles with the ground truth transcriptional profiles of neuronal sub-types (Supplementary Figure S21A). We then sought to evaluate whether increasing the number of deconvolved cell-types could recover the finer neuronal sub-types. We therefore applied STdeconvolve with $K=76$ and were able to identify deconvolved cell-types that were highly correlated in terms of both transcriptional profiles and pixel proportions to finer neuronal subtypes as well as rare cell-types such as pericytes and microglia (Supplementary Figure 21B-C). However, as noted previously, the ability for STdeconvolve to deconvolve neuronal subtypes may not be representative of its expected performance in distinguishing between cell sub-types under less biased gene selection.

A**B****C**
Supplementary Figure S21. Comparison of STdeconvolve cell-types to ground truth neuronal subtypes of simulated MERFISH MPOA ST data. A) Heatmap of Pearson's correlations between the transcriptional profiles of the 76 ground truth cell-types and neuronal subtypes in the MERFISH MPOA data and STdeconvolve cell-types that matched to excitatory and inhibitory neuronal major cell-type types. B) Heatmap of Pearson's correlations between the transcriptional profiles of the 76 ground truth cell-types and neuronal subtypes in the MERFISH MPOA data and 76 deconvolved cell-types by STdeconvolve. C) Heatmap of Pearson's correlations between the cell-type pixel proportions of the 76 ground truth cell-types and neuronal subtypes in the MERFISH MPOA data and 76 deconvolved cell-types by STdeconvolve.

Minor Comment 2-2: It is very hard to see the H&Es under the scatter-pie chart visualizations (E.g. in figure 3b). Can the authors provide a different visualization or quantification of that their cell types align with the histological structures?

We agree with the reviewer that the scatter-pie charts are difficult to see when overlaid on top of the corresponding H&E images. Scatter-pie plots indicating pixel proportions of cell-types are no longer covering H&E images. In cases where it is relevant to show the actual tissue H&E image, the scatter pies have been placed next to the image. For example, Figure 5A-C now shows the deconvolved cell-type proportions for all 15 cell-types, as well as cell-type X15 specifically, next to the H&E stained tissue section.

Figure 5A-C. STdeconvolve characterizes the spatial organization of immune cells in real and simulated breast cancer ST data. A) An H&E-stained image of the breast cancer tissue with pathological annotations adapted from Yoosuf *et al.*²². B) Deconvolved cell-type pixel proportions for ST data of a breast cancer tissue section, represented as pie charts. Pixels are outlined with colors based on the pixel transcriptional cluster assignment corresponding to 3 pathological annotations. C) Highlight of deconvolved cell-type X15. Pixel proportion of deconvolved cell-type X15 are indicated as black slices in pie charts. Pixels are outlined with colors as in B). Select genes corresponding cell-type X15's select top marker genes are shown.

For quantification of the cell-type proportions and their respective alignment with the histological structure, we note that in the original publication, clustering analysis of the pixels aligned with histological structures and our deconvolved cell-types align with the pixel clusters based on proportional overlaps. We note this finding in the revised Results on lines 397-399 and have provided the text below for the reviewer's convenience.

Of these, deconvolved cell-types X3 and X13 pixel proportions corresponded spatially with pixels annotated as the non-malignant and DCIS regions, respectively (Supplementary Figure S16D).

D

Supplementary Figure S16D. Deconvolution of the breast cancer ST data by STdeconvolve. D) Pearson's correlation between the pixel proportions of 15 deconvolved cell-types by STdeconvolve and the breast cancer tissue annotations. Black boxes highlight tissue annotations with highest correlations to each deconvolved cell-type.

Reviewer 3

Overview 3-0: Miller et.al., in this work, have developed an unsupervised, reference-free approach for deconvolving multi-cellular pixel resolution of spatially resolved transcriptome data. In comparison to other recent approaches, their methodology is unique in that it is devoid of references. When sufficient single-cell references are available, the authors claim that their method recapitulates expected biology and gives comparable performance to existing supervised approaches, as well as potentially higher performance when acceptable single-cell references are not available. The paper is well written and could be accepted upon revision of the issues listed below.

Major Comment 3-1: STdeconvolve first feature selects for gene likely to be informative of transcriptionally distinct cell-types (line 45-46):

a. Is the approach sensitive to the number of genes (in the first step: feature selection of informative genes) How many minimum and maximum genes are required?

We thank the reviewer for raising this interesting point. We note that LDA seeks to represent latent “topics”, or cell-types, as ideally non-overlapping groups of co-expressed, or frequently co-occurring, genes in different pixels. Several parameters will likely affect the ability of LDA to achieve this grouping of genes into distinct cell-types, which include the number of cell-types to be deconvolved, how distinct they are transcriptionally, how variable cell-types are across ST pixels, and how the gene expression values were measured, e.g., targeted gene panel or transcriptome-wide profiling. For example, in our simulated ST data based on the single-cell resolution MERFISH data of the MPOA, a panel of 135 genes was specifically curated to optimally distinguish between 7 major classes of non-neuronal cell-types in addition to excitatory and inhibitory neuronal subtypes. In this particular situation, given only 135 genes, STdeconvolve was still able to deconvolve underlying cell-types. However, we note that real multi-cellular pixel resolution ST data is not limited to pre-selected gene panels known to optimally distribute between cell-types. These thoughts have been included as part of “**Supplementary Note 3**” (Lines 145-179 in Supplementary Materials) and relevant excerpts are included below for the reviewer’s reference.

For this MERFISH dataset, a specific panel of 135 genes previously chosen to optimally distinguish primarily between neuronal subtypes. As such, only 33 remaining genes were available to distinguish between all major cell-types⁸. Therefore, these chosen genes represent a specific subset of overdispersed genes rather than all overdispersed genes that are skewed towards distinguishing between neurons, even though many more overdispersed genes may exist for distinguishing microglia and pericytes, for example. In this manner, neuronal subtypes may appear more transcriptionally distinct than certain cell-types for which few markers were included. Indeed, when we applied STdeconvolve to the simulated MERFISH MPOA ST data limited to only these 135 genes, deconvolved cell-types such as cell-types X2 and X8 both matched to excitatory neurons while cell-types X4 and X7 both matched to inhibitory neurons. Given that excitatory and inhibitory neurons could previously be further subdivided into finer neuronal sub-types, we sought to evaluate whether these deconvolved cell-types that matched to the excitatory and inhibitory neurons could represent these additional finer neuronal sub-types. To test this hypothesis, we further partitioned the ground-truth excitatory and inhibitory cell-types into additional sub-types based on previous annotations, resulting in 76 total non-neuronal cell-types and neuronal subtypes⁸. Comparing the deconvolved transcriptional profiles of X2, X4, X7, and X8 to the ground truth transcriptional profiles of the 76 non-neuronal cell-types and neuronal subtypes, we indeed observed a correlation between the deconvolved transcriptional profiles with the ground truth transcriptional profiles of neuronal sub-types (Supplementary Figure S21A). We then sought to evaluate whether increasing the number of deconvolved cell-types could recover the finer neuronal sub-types. We therefore applied STdeconvolve with $K=76$ and were able to identify deconvolved cell-types that were highly correlated in terms of both transcriptional profiles and pixel proportions to finer neuronal subtypes as well as rare cell-types such as pericytes and microglia (Supplementary Figure 21B-C). However, as noted previously, the ability for STdeconvolve to deconvolve neuronal subtypes may not be representative of its expected performance in distinguishing between cell sub-types under less biased gene selection.

A**B****C**
Supplementary Figure S21. Comparison of STdeconvolve cell-types to ground truth neuronal subtypes of simulated MERFISH MPOA ST data. A) Heatmap of Pearson's correlations between the transcriptional profiles of the 76 ground truth cell-types and neuronal subtypes in the MERFISH MPOA data and STdeconvolve cell-types that matched to excitatory and inhibitory neuronal major cell-type types. B) Heatmap of Pearson's correlations between the transcriptional profiles of the 76 ground truth cell-types and neuronal subtypes in the MERFISH MPOA data and 76 deconvolved cell-types by STdeconvolve. C) Heatmap of Pearson's correlations between the cell-type pixel proportions of the 76 ground truth cell-types and neuronal subtypes in the MERFISH MPOA data and 76 deconvolved cell-types by STdeconvolve.

Real multi-cellular pixel resolution ST data typically provide transcriptome-wide gene expression profiling. Again, as latent cell-types are best discovered by LDA modeling if cell-type specific marker genes are included in the input ST data while genes whose expression is shared across cell-types are excluded, STdeconvolve filters for genes that are more likely to be specifically expressed in particular cell-types by identifying overdispersed genes across pixels in the dataset, or genes with higher-than-expected expression variance across pixels²³.

To explore whether this feature selection step is sensitive to the number of genes, we have now simulated ST datasets using mixtures of single cells and then evaluated the performance of STdeconvolve in deconvolving these cell mixtures given differing numbers of genes. Specifically, we generated simulated ST datasets in which pixels were composed of mixtures of luminal cells, pericytes, and macrophages sampled from a scRNA-seq dataset²⁴. We sampled random cells of each cell-type into mixtures within multi-cellular pixels based on a Dirichlet distribution with a sparse shape parameter ($\alpha = 0.4$), and the gene counts of cells in each pixel were collapsed. From this simulated ST dataset, we feature selected for overdispersed genes, and from this list, subsampled different numbers of overdispersed genes. With each panel of selected genes, we deconvolved $K=3$ cell-types and computed the RMSE across the pixels. More details are provided in the Supplementary Methods section “**Accuracy of STdeconvolve**

with respect to number of feature selected genes” (Lines 513-526 in Supplementary Materials). We repeated this process for each subsampling three times. Overall, we find that RMSE decreases and thus deconvolution performance improves as the number of genes increase. However, the reduction in RMSE also diminishes as the number of genes increases, suggesting that after a certain point, adding additional genes provides no further improvement. In this manner, these results suggest that our approach may be sensitive to particularly small numbers of informative features, though performance stabilizes for larger numbers of informative features. Given this, we have now also set STdeconvolve to use the top 1000 overdispersed genes by default, though users may still modulate this number depending on their knowledge of the dataset such as regarding the transcriptional distinctness of the cell-types and how many are to be deconvolved. We now indicate this in the revised Methods section **“Selection of genes for LDA model”** (lines 550-554) and have provided the relevant text below for the reviewer’s convenience.

By default, only the top 1000 most overdispersed genes are retained in the input ST data because we note that deconvolution accuracy in general stabilizes for larger numbers of informative features (Supplementary Figure S22).

Additional gene filtering or cell-type specific marker genes to include in the input ST data may also be augmented by the user.

Supplementary Figure S22. Deconvolution accuracy by number of randomly sampled feature selected genes. Each point represents the mean RMSE based on the deconvolved versus ground truth cell-type proportions across 900 simulated pixels for 3 replicates. Vertical bars represent the standard deviation from the mean. Genes were sampled from the top 1000 most significant overdispersed genes.

Major Comment 3-2: Further partitioning the ground-truth excitatory and inhibitory cell-types into additional subtypes (76 total) based on the previous annotation found that these deconvolved cell types correlated with specific combinations of neuronal subtypes (line 65-68)

a. It seems that the MERFISH paper identified ~70 different neuronal populations (PMID:30385464). It is not clear how the 9-ground truth cell-types (Figure 2D) were selected if the original data (ground truth) has 76.

We thank the reviewer for raising this point of clarification. The simulated ST data of the MPOA was built using single-cell resolution MERFISH data. In the original publication, single-cell RNA-seq and clustering analysis was first performed to identify cell-types in the MPOA. From there, a panel of 135 genes was specifically chosen to optimally distinct between 7 major classes of non-neuronal cell-types in addition to excitatory and inhibitory neuronal subtypes. Spatially resolved MERFISH data was then collected for this select panel of 135 genes and clustering analysis was subsequently performed to identify 9 major cell-types. In this manner, the ground truth cell-types were determined based on the original publication’s clustering analysis. Subsequently, the neuronal cells were

reclustered to identify additional subtypes. We now clarify this point on lines 95-105 and provide the relevant text below for the reviewer's reference.

We simulated ST data by aggregating the gene expression of cells from single-cell resolution multiplex error-robust fluorescence *in situ* hybridization (MERFISH) data of the mouse medial pre-optic area (MPOA)⁸ within spatially contiguous pixels. Previously, MERFISH was previously applied to map the spatial distribution of 135 select genes within MPOA brain tissue. These select 135 genes were chosen to distinguish between major non-neuronal cell-types as well as neuronal subtypes. Imaging-based cell segmentation was performed and the counts of genes per cell were quantified to achieve single-cell resolution spatially resolved transcriptomic profiling. Subsequent transcriptional clustering analysis on the single-cell resolution gene expression measurements identified 9 major cell-types, including excitatory and inhibitory neurons.

b. For figure 3E, can the cluster (right vertical labels) be identified only based on correlation? If so, then it's hard to judge which K value (K=9 or K=76) is performing better? For instance, X7 which is an inhibitory neuron appears to be similarly correlated to both excitatory and inhibitory (Figure S3 D). Similarly, X3 rather than X9 looks more like astrocytes (Figure S3 B). Can a graph similar to Figure S14 be provided for different Ks? Can the cluster identification (in addition to correlation) be marker-assisted? For instance, *Gad1* and *Gad2* profile for the inhibitory cluster. *Sl17a7* for Excitatory cluster.

We thank the reviewer for raising this point of clarification. We would like to point out that our ground truth cell-type assignment of the deconvolved cell-types is actually determined by the enrichment of ground truth marker genes in the deconvolved transcriptional profiles much like the reviewer suggests. To do this, we first determined differentially expressed marker genes for each of the ground truth cell-types followed by rank-based gene set enrichment analysis of the ground truth gene sets in each deconvolved cell-type transcriptional profile. A match to a ground truth cell-type was confirmed and assigned if the ground truth gene set had the lowest gene set enrichment adjusted *p*-value that was at least < 0.05 , followed by the highest positive edge score²⁵, and then highest positive enrichment score to break ties. We have now updated the Methods to clarify this under “**Annotation and matching of deconvolved and ground truth cell-types**” (Lines 626-643) with relevant excerpts selected below for the reviewer's reference:

The assignment of deconvolved cell-types to ground truth cell-types was confirmed by testing for enrichment of differentially upregulated genes of the ground truth cell-types in the deconvolved cell-type transcriptional profiles. To determine the differentially upregulated genes of the ground truth cell-types, ground truth transcriptional profiles were converted to counts per thousand and low expressed genes, defined as those with average expression values less than 5, were removed. For each ground truth cell-type, the \log_2 fold-change of each remaining gene with respect to the average expression across the other ground-truth cell-types was computed. Differentially upregulated genes were those with \log_2 fold-change > 1 . We performed rank-based gene set enrichment analysis of the ground truth upregulated gene sets in each deconvolved cell-type transcriptional profile using the `liger` R package²⁶. A match to a ground truth cell-type was confirmed and assigned if the ground truth gene set had the lowest gene set enrichment adjusted *p*-value that was at least < 0.05 , followed by the highest positive edge score²⁵, and then highest positive enrichment score to break ties.

We note that cell-type X7 is enriched in both inhibitory and excitatory genes but based on our cell-type assignment was slightly more enriched in excitatory genes. Cell-type X3 was highly enriched in ependymal genes compared to astrocytes, and likewise cell-type X9 was enriched for astrocyte genes but not ependymal genes. However, astrocytes and ependymal cell-types are correlated in terms of their transcriptional profiles overall (Supplementary Figure S2C), which likely explains why cell-types X3 and X9 transcriptionally could be mistaken for the other, thus demonstrating the importance of our annotation and validation procedure. Further, with these assignments, cell-type X3 also correlates most strongly with the ependymal cells compared to the other ground truth cell-types in terms of cell-type pixel proportions, as does cell-type X9 with respect to the ground truth astrocytes. Finally, given the select 135 gene panel used in the MERFISH data, one of the top genes in the deconvolved transcriptional profile of cell-type X8, assigned to the excitatory neurons, is *Slc17a6*. Likewise, the top gene for cell-type X4, assigned to inhibitory neurons, is *Gad1*.

Supplementary Figure S2C. Comparison of deconvolved cell-types from STdeconvolve to ground truth cell-types of simulated MERFISH MPOA ST data. C) Pearson's correlation between the transcriptional profiles of the 9 ground truth cell-types.

Major Comment 3-3: We evaluated the performance of each approach using the root-mean-square-error of the deconvolved cell-type proportion to ground truth across simulated pixels. In general, we find the performance of STdeconvolve to be comparable to SPOTlight and RCTD (lines 79-82).

a. Can you provide the p-value of the difference between the three methods in Figure S5?

We thank the reviewer for this point and we have now calculated statistical differences using a one-sided Diebold-Mariano Test²⁷ and report *p*-values between distributions of predicted cell-type pixel proportion RMSEs between STdeconvolve and the other approaches.

How does the RSME plot look when you consider $K=76$ and ST dataset at 20um (described in lines 51-76)?

We thank the reviewer for this suggestion. The RMSE distribution of cell-type pixel proportions serves as a quantitative means to compare different approaches in terms of their deconvolution accuracy on the simulated MPOA ST data. To this end, an important point we try to emphasize is that reference-based deconvolution approaches are sensitive to the reference they are trained on. We have tried to demonstrate this point by deconvolving the simulated MERFISH MPOA ST data using various single-cell transcriptomics references that comprise these 9 major cell-types. One difficult consideration in repeating this analysis with the 76 cell-types would be how to deal with the independent mouse brain scRNA-seq reference, which does not contain up to 76 finer subtype annotations. Thus, the reference dependent methods would already be at a disadvantage due inherently not having enough reference cell-types to match to the 76 ground truth cell-types if the neuronal subtypes were used in the ground truth.

However, in terms of deconvolving simulated datasets at different resolutions, we now assess STdeconvolve's performance at additional simulated resolutions of the simulated ST MPOA dataset at 100, 50, 20, and 10 μm^2 . Here, using $K=9$, we do compute the RMSE at each resolution and find that both the deconvolved transcriptional profiles and predicted pixel proportions of the 9 deconvolved cell-types are highly correlated with the 9 major ground truth cell-types. Additionally, the RMSEs between STdeconvolve predicted pixel proportions of 9 major cell-types across the different pixel resolutions are similar and do not increase, suggesting that STdeconvolve is accurate even at high multi-cellular pixel resolution. These results are now included in a new section **"STdeconvolve is applicable across diverse ST dataset resolutions and technologies"** (Lines 320-384) and the relevant sections are provided below for the reviewer's reference.

First, to simulate higher resolution ST data, we again aggregated single-cell resolution MERFISH data of the MPOA into 50, 20, and 10 μm^2 resolution pixels. Applying STdeconvolve, we observed similarly strong correlations between the deconvolved cell-type transcriptional profiles and proportions with the ground truth (Supplementary Figure S11A-D).

Supplementary Figure S11. Comparison of STdeconvolve cell-types to ground truth cell-types in high resolution pixel MERFISH MPOA ST datasets. A) Deconvolution accuracy in 50 μm^2 simulated pixels. Left: The ranking of each gene based on its expression level in the transcriptional profiles of the deconvolved cell-types,

compared to its gene rank in the transcriptional profile of the matched ground truth cell-type. Middle: Pearson's correlation between the transcriptional profiles of the 9 ground truth cell-types in MERFISH MPOA data and 9 deconvolved cell-types by STdeconvolve. Right: Pearson's correlation between the pixel proportions of the 9 ground truth cell-types in the MERFISH MPOA data and 9 deconvolved cell-types by STdeconvolve. B) Deconvolution accuracy in $20 \mu\text{m}^2$ simulated pixels. C) Deconvolution accuracy in $10 \mu\text{m}^2$ simulated pixels. D) Root-mean-square-error (RMSE) of the deconvolved cell-type pixels proportions compared to ground truth cell-types at each resolution of the simulated ST dataset. E) Distributions of number of individual cells contained within pixels of each simulated resolution of the MERFISH MPOA ST dataset. Numbers above each dataset indicate the total number of pixels for each simulated ST data. F) Stacked bar plots showing the fraction of pixels that contain a distinct number of different cell-types for each simulated resolution of the MERFISH MPOA ST dataset. This was done for pixels that contain at 2 or more individual cells. Numbers within each box indicate the total number of pixels with that many distinct cell-types and number above the bars indicate the total number of pixels with 2 or more individual cells.

b. Please show all figures for SPOTlight and RCTD similar to Figure S3 and S4.

We thank the reviewer for the suggestion. Heatmaps of Pearson's correlations between deconvolved and ground truth cell-type pixel proportions in the simulated MPOA ST dataset are now shown in Supplementary Figure S3. As the supervised deconvolution approaches do not deconvolve transcriptional profiles, those Pearson's correlations are not available. We also show these for each method after being trained with a different single-cell reference and are provided below for the reviewer's convenience.

Supplementary Figure S3. Cell-type predictions of supervised and semi-supervised deconvolution approaches across simulated MERFISH MPOA ST dataset pixels with different single-cell transcriptomics references. A) Deconvolved pixel proportions of cell-types by each supervised and semi-supervised deconvolution approach using

the MERFISH MPOA single-cell transcriptomics data as the reference. B) Deconvolved pixel proportions of cell-types by each supervised and semi-supervised deconvolution approach using the MERFISH MPOA single-cell transcriptomics data with missing neurons as the reference. C) Deconvolved pixel proportions of cell-types by each supervised and semi-supervised deconvolution approach using the MERFISH MPOA single-cell transcriptomics data with missing endodermal cells as the reference.

c. Recently, graph-based semi-supervised algorithms have been developed (PMID:33480403, 33303016). Please show how STdeconvolve stacks up against them.

We thank the reviewer for raising this important comparison. Of the two suggested methods, we have now included an assessment of DSTG to deconvolve the 9 major ground truth cell-types in our simulated MPOA ST dataset. We used the initial single-cell MERFISH data at the single-cell reference and performed the default pipeline to generate pseudo-mixtures of ground truth cells, which were used to train the model and deconvolve cell-types in the simulated ST dataset. However, we found that DSTG did not deconvolve any distinct cell-types in our MPOA dataset. The results of this are shown in Supplementary Figure S3A. This was also the case when we used the MOB scRNA-seq reference to train DSTG to deconvolve cell-types in the real MOB ST dataset. The second suggested approach, GCNG, was developed to primarily predict gene interaction networks and therefore we did not feel it was appropriate for comparison with respect to cell-type deconvolution of multi-cellular ST data.

d. Although there are budgetary constraints (line 40), the availability of numerous scRNAseq datasets online makes it simple to obtain reference data for SPOTlight and RCTD. Naturally, the references will not be as ideal as the ones used by the authors (i.e., from the same source). In this regard, STdeconvolve, being reference-free, will be of great advantage. The authors must demonstrate, however, that STdeconvolve surpasses SPOTlight and RCTD when the source reference is not available (i.e., reference scRNAseq data not from the same study).

We thank the reviewer for this important point. To this end, we have now expanded the comparison between STdeconvolve and reference-dependent methods to also include the case in which an ideal reference that optimally matches the cell-types in the ST dataset is not available. For this, we have now trained the reference-dependent methods using a scRNA-seq reference from a mouse brain atlasing effort followed by deconvolution of the simulated ST MPOA dataset. We believe this scRNA-seq reference presents a more realistic use case given the availability of numerous scRNA-seq datasets online that are not from the same source as the ST dataset to be deconvolved as the reviewer notes. Indeed, we find that the performance for all reference-dependent methods resulted in a significantly higher RMSE than STdeconvolve. We hope that this now more clearly demonstrates how STdeconvolve may surpass reference-based deconvolution approaches when a source reference is not available. The results of this comparison are in the main Results in section “**STdeconvolve achieves competitive performance to reference-based, supervised deconvolution approaches**” (Lines 124-169) with selected excerpts provided below for the reviewer’s reference.

Likewise, such an ideal single-cell transcriptomics reference that optimally matches the cell-type composition and measurement sensitivities of the ST data to be deconvolved may not be available. Therefore, this ideal MERFISH MPOA single-cell transcriptomics reference likely provides an upper bound on performance for reference-based deconvolution approaches. To provide a more realistic evaluation of performance for reference-based deconvolution approaches, we sought to deconvolve our simulated ST data of the MPOA using a scRNA-seq reference from a mouse brain atlasing effort¹². Again, as a reference-free deconvolution approach, the performance of STdeconvolve does not change. However, again, the performance for all reference-dependent methods resulted in a significantly higher RMSE (Diebold-Mariano p -value $< 2.2 \times 10^{-16}$) than STdeconvolve (Figure 2H, Supplementary Methods). Thus, STdeconvolve achieves comparable performance to reference-based, supervised deconvolution approaches when an ideal single-cell transcriptomics reference is used, and potentially better performance when an ideal single-cell transcriptomics reference is not used.

Figure 2E-H. Deconvolution of simulated ST data. E) Root-mean-square-error (RMSE) of the deconvolved cell-type proportions compared to ground truth for STdeconvolve, F) for supervised deconvolution approaches using the ideal single cell transcriptomics MRFISH MPOA reference, G) for supervised deconvolution approaches using the single cell transcriptomics MRFISH MPOA reference with missing neurons, and H) for supervised deconvolution approaches using a brain single-cell RNA-seq reference.

Reviewer 4

Overview 4-0: This paper proposes to apply Latent Dirichlet Allocation (LDA) to perform an unsupervised deconvolution of spatial transcriptomics data without any single-cell reference data. It is noteworthy that they accurately inferred the presence of the rostral migratory stream in the olfactory bulb that other methods previously failed to detect. However, applying LDA to decompose bulk RNA-seq data has been already considered by Dey et al. (PLoS Genetics, 2017, <https://doi.org/10.1371/journal.pgen.1006599>), so the novelty of this manuscript is questionable. The authors should cite this work and explain the novelty of their contribution. Furthermore, as detailed below, some of the assumptions of LDA do not seem to hold in the spatial transcriptomics case, but these issues are not addressed in the manuscript. Lastly, more comprehensive benchmarking needs to be performed to demonstrate the improvement over other approaches.

We thank the reviewer for bringing up previous applications of LDA in biological data and its applicability to ST data. While LDA has been used to deconvolve bulk RNA-seq data, we note that there are inherent differences between ST data and bulk RNA-seq, of which we find ST data to be particularly amenable to LDA. Briefly, these include i) the limited number of cells and cell-types represented in each ST pixel, ii) the large number of pixels compared to cell-types, iii) the likely heterogeneity of cell-type proportional distributions across pixels in tissues, and iv) the limited impact of batch effects on the measured gene expression across pixels measured in the same ST experiment. In contrast, when comparing collections of bulk samples, the total number of samples can be smaller than the large number of pixels expected in ST datasets. Bulk RNA-seq samples could either vary substantially from each other in that each bulk sample may contain distinct cell-types not present in any of the other samples, which would violate the mixed membership structure of LDA. Or, conversely, a collection of bulk RNA-seq samples may be very similar in cell-type proportional composition, if for example they were all peripheral blood samples. Here, there would be minimal heterogeneity in cell-type composition. Finally, each bulk sample collected independently of the others means that there are inherent batch effects, of which sequencing depth is a major contributor because samples may vary by orders of magnitude in read depth. In this context, one could image two cell-types that express the same genes but at multiples of each other and be interpreted as the same cell-type by LDA. We have added these thoughts about the amenability of ST data for LDA as well as mention LDA's previous applications to bulk RNA-seq in the revised Introduction (lines 36-88) and have included the relevant text below for the reviewer's reference.

In the context of ST data, given a count matrix of gene expression in multi-cellular ST pixels, STdeconvolve applies LDA to infer the putative transcriptional profile for each cell-type and the proportional representation of each cell-type in each multi-cellular ST pixel (Methods). While LDA has previously been applied in the context of deconvolving cell-types in bulk RNA-seq data²⁸, STdeconvolve leverages several unique features of ST data that make this application of LDA particularly amenable (Supplementary Note 1). Briefly, these include i) the limited number of cells and cell-types represented in each ST pixel, ii) the limited impact of batch effects on the measured gene expression across pixels, iii) the large number of pixels compared to cell-types, and iv) the likely heterogeneity of cell-type proportional distribution across pixels in tissues.

The suitability of ST data for LDA has also been expanded upon in **Supplementary Note 1** (lines 94-124) in the Supplementary Materials.

With respect to the novelty of our approach, to improve the application of LDA towards ST data, we have implemented a feature selection step to identify genes informative of latent cell-types without reliance on external single-cell or cell sorted bulk RNA-seq references as is done in various bulk deconvolution approaches. In addition, previous bulk deconvolution approaches such as the one from Dey et al. have relied on analysts to specify the number of K cell-types to be deconvolved, such as by exploring multiple values of K and manually evaluating stability. STdeconvolve provides several data-driven metrics to guide the estimation of an appropriate K , which are summarized in the Methods in section **"Selection of LDA model with optimal number of cell-types"** (lines 555-587). We provide relevant text below for the reviewer's convenience.

The number of cell-types K in the LDA model must be chosen *a-priori*. To determine the optimal number of cell-types K to choose for a given dataset, we fit a set of LDA models using different values for K over a user defined range of positive integers greater than 1. We then compute the perplexity of each fitted model:

$$\text{Perplexity}(D) = \exp \left\{ -\frac{\log(p(D))}{\sum_{d=1}^D \sum_{n=1}^N c_{d,n}} \right\}$$

Where $p(D)$ is the likelihood of the dataset and $c_{d,n}$ is the gene count, or expression level, of gene n in pixel d . We can interpret $p(D)$ as the posterior likelihood of the dataset conditional on the cell-type assignments using the final estimated θ and β . The lower the perplexity, the better the model represents the real dataset. Thus, the trend between choice of K and the respective model perplexity can then serve as a guide. By default, the perplexity is computed by comparing $p(D)$ to the entire input dataset used to estimate θ and β .

In addition, STdeconvolve also reports the trend between K and the number of deconvolved cell-types with mean pixel proportions $< 5\%$ (as default). We chose this default threshold based on the difficulty of STdeconvolve and reference-based deconvolution approaches to deconvolve cell-types at low proportions, (i.e., “rare” cell-types) (Supplementary Note 2). We note that as K is increased for fitted LDA models, the number of such “rare” cell-types generally increases. Such rare deconvolved cell-types are often distinguished by fewer distinct transcriptional patterns in the data and may represent non-relevant or spurious subdivisions of primary cell-types. We can use this metric to help set an upper bound on K .

Generally, perplexity decreases and the number of “rare” deconvolved cell-types increases as K increases. Given these model perplexities and number of “rare” deconvolved cell-types for each tested K , the optimal K can then be determined by choosing the maximum K with the lowest perplexity while minimizing number of “rare” deconvolved cell-types. To further guide the choice of K , an inflection point (“knee”) is derived from the maximum second derivative of the plotted K versus perplexity plot and K versus number of “rare” deconvolved cell-types.

Still, for a given K , the fitted LDA model may fail to identify distinct cell-types e.g., the distribution of cell-type proportions in each pixel is uniform. In such a situation, the Dirichlet distribution shape parameter α of the LDA model will be ≥ 1 and STdeconvolve will indicate to the user that the fitted LDA model for a particular K has an α above this threshold by greying out these K s in the trend plot.

Ultimately, the choice of K is left up to the user and can be chosen taking into consideration prior knowledge of the biological system.

We have now included a citation to Dey et al.’s previous work and subsequently clarified the novel features of STdeconvolve in the main text, provided below for the reviewer’s reference (lines 75-78):

While LDA has previously been applied in the context of deconvolving cell-types in bulk RNA-seq data²⁸, STdeconvolve leverages several unique features of ST data that make this application of LDA particularly amenable (Supplementary Note 1).

Finally, we agree with the reviewer that additional benchmarking is needed and to this end we have compared STdeconvolve to additional approaches, different types of real ST datasets at different resolutions, and have performed an assessment of runtime and total memory usage with respect to ST dataset, which we go into further detail in the responses below.

Major Comment 4-1: The method has been applied to only MERFISH data, so it is unclear how it would perform when applied to whole-transcriptome spatial data with much more features. Benchmarking using single-cell-resolution whole-transcriptome data from XYSeq or sci-Space would be worthwhile. Also, the method could be benchmarked on multi-cellular whole-transcriptome data from Slide-Seq or 10X Visium.

We thank the reviewer for raising this important point. In addition to simulated ST data using single-cell MERFISH, we have now applied STdeconvolve to newer ST technologies, which include 10X Visium at 50 μm^2 pixel resolution, DBiT-seq at 25 μm^2 pixel resolution, and Slide-seq at 10 μm^2 pixel resolution. We show that our deconvolution analysis is able to recapitulate previously reported findings and identify known biological features, indicating that STdeconvolve is applicable to whole-transcriptome spatial data with much more features. These results are now presented in a new section called “**STdeconvolve is applicable across diverse ST dataset**”

resolutions and technologies” (Lines 320 to 384) in the revised manuscript, with relevant text provided below for the reviewer’s reference.

We anticipate that continual technological improvements will enhance the resolution of ST data. Already, ST technologies such as Visium (10X Genomics), Slide-seq², and DBiT-seq³ have achieved resolution that can range from 50 μm^2 to 10 μm^2 . Therefore, we sought to evaluate the performance of STdeconvolve on higher resolution ST data using both simulated as well as real ST data from higher resolution ST technologies including Visium, Slide-seq, and DBiT-seq...Encouraged by STdeconvolve’s ability to recover cell-types in simulated high-resolution ST data, we then applied STdeconvolve to real high-resolution multi-cellular ST data from several different technologies. First, we applied STdeconvolve to 50 μm^2 resolution ST data of a coronal section of the mouse brain from 10X Visium⁴. Briefly, for 10XVisium, mRNAs from tissue sections are captured onto an array of DNA barcoded spots, resulting in RNA-sequencing measurements with gridded 2D spatial positional information. We applied STdeconvolve to identify $K=20$ cell-types that exhibit spatially distinct patterns that demarcate known brain structures such as the isocortex and fiber tracts (Figure 4E, Supplementary Figure S12, Supplementary Methods).

Figure 4E. Deconvolution of ST data of varying resolution from multiple technologies by STdeconvolve. E) Deconvolved cell-type proportions for Visium data of the mouse brain.

A X13 and X15**B** isocortex
X19

fiber tract

Supplementary Figure S12. STdeconvolve pixel proportion of select deconvolved cell-types in 10X Visium data of the mouse brain. A) STdeconvolve pixel proportions for select deconvolved cell-types. B) Visually corresponding annotated brain region from the Allen Brain Atlas⁵.

We next applied STdeconvolve to 25 μm^2 resolution ST data of the lower body of the E11 mouse embryo from DBiT-seq. Briefly, for DBiT-seq, parallel microfluidic channels are used to deliver DNA barcodes to the surface of a tissue to enable direct barcoding of mRNAs *in situ*, resulting in RNA-sequencing measurements in a 2D mosaic of spatial pixels. Previously, the authors identified 13 transcriptionally and spatially distinct features in the E11 mouse embryo including the atrium, ventricle, liver, and blood vessels containing erythrocyte coagulation. Applying STdeconvolve with $K=13$, we identify deconvolved cell-types that corresponded with similar spatially distinct features in agreement with the original findings (Figure 4F, Supplementary Figure S13A, Supplementary

Methods). Moreover, the top genes in the deconvolved cell-type specific transcriptional profiles contained the expected marker genes of the matching features, such as *Myh6* for the atrium, *Myh7* for the ventricle, *Apoa2* for the liver, and *Hba.a2* for the blood vessels containing erythrocyte coagulation in agreement with the original findings (Supplementary Figure S13B).

Figure 4F. Deconvolution of ST data of varying resolution from multiple technologies by STdeconvolve. F) Deconvolved cell-type proportions for DBiT-seq data of the lower body of an E11 mouse embryo.

Supplementary Figure S13. STdeconvolve pixel proportions and transcriptional profiles of select deconvolved cell-types in DBiT-seq data of an E11 mouse embryo lower tail section. A) Pixel proportions of select deconvolved cell-types X7, X11, X1, and X3 corresponding to the atrium, ventricle, fetal liver, and erythrocyte

coagulation, respectively, annotated in the previous publication³. B) Deconvolved transcriptional profiles for cell-types X7, X11, X1, and X3 with the expression of the top gene in each deconvolved transcriptional profile visualized in the original tissue (inset).

Finally, we applied STdeconvolve to 10 μm^2 resolution ST data of the mouse cerebellum from Slide-seq. Briefly, for Slide-seq, mRNAs from tissue sections are captured onto densely packed barcoded beads, resulting in RNA-sequencing measurements with 2D spatial positional information. Previously, RCTD was also applied to this Slide-seq dataset with a matched Drop-seq scRNA-seq reference of the mouse cerebellum⁶ to identify beads representing a distinct layers of Purkinje neurons and Bergmann glia. Applying STdeconvolve, we identified $K=14$ cell-types (Figure 4G, Supplementary Methods) whose transcriptional profiles correlated strongly with cell-types from the scRNA-seq dataset of the mouse cerebellum (Supplementary Figure S14A). In particular, we found that the deconvolved transcriptional profiles of cell-type X4 and cell-type X2 correlated strongly with the transcriptional profiles of Purkinje neurons and Bergmann glia. Likewise, the deconvolved proportional representation of cell-type X4 and cell-type X2 also agreed significantly (Fisher's Exact p -value $< 2.2 \times 10^{-16}$) with the predicted proportions of Purkinje neuron and Bergmann glia from RCTD (Supplementary Figure S14B-C). Taken together, these results indicate that STdeconvolve can be applicable to a range of multi-cellular resolution ST technologies.

Figure 4G. Deconvolution of ST data of varying resolution from multiple technologies by STdeconvolve. G) Deconvolved cell-type proportions for Slide-seq data of the mouse cerebellum.

Supplementary Figure S14. STdeconvolve predicted cell-type proportions of Purkinje neurons and Bergmann glia in Slide-seq data of the mouse cerebellum compared to RCTD. A) Pearson's correlation between the deconvolved transcriptional profiles of the 14 STdeconvolve cell-types and best matched cell-types in the mouse cerebellum scRNA-seq reference⁶. B) Beads predicted to contain Purkinje neurons (cyan) or Bergmann glia (magenta) by either RCTD (left) or STdeconvolve (right). C) Stacked bar plots indicating the fractions of beads predicted to contain Purkinje neurons (left) or Bergmann glia (right) by RCTD and STdeconvolve.

Major Comment 4-2: In addition to SPOTlight and RCTD, there are other deconvolution methods, including cell2location, SpatialDWLS, stereoscope, and DSTG. The benchmarking should be expanded to include these methods and other settings/data types.

We thank the reviewer for bringing up these additional approaches. We have now expanded the comparison between STdeconvolve and other deconvolution approaches to include spatialDWLS and DSTG. We note that spatialDWLS has already extensively been compared to stereoscope so we did not repeat the comparison with stereoscope. Cell2location remains in development and unpublished and so was therefore omitted. We found that DSTG was not able to deconvolve distinct cell-types and was omitted from most of the quantitative comparisons, but we present the results in Supplementary Figure S3, which are shown below for reference.

Supplementary Figure S3. Cell-type predictions of supervised and semi-supervised deconvolution approaches across simulated MERFISH MPOA ST dataset pixels with different single-cell transcriptomics references. A) Deconvolved pixel proportions of cell-types by each supervised and semi-supervised deconvolution approach using

the MERFISH MPOA single-cell transcriptomics data as the reference. B) Deconvolved pixel proportions of cell-types by each supervised and semi-supervised deconvolution approach using the MERFISH MPOA single-cell transcriptomics data with missing neurons as the reference. C) Deconvolved pixel proportions of cell-types by each supervised and semi-supervised deconvolution approach using the MERFISH MPOA single-cell transcriptomics data with missing ependymal cells as the reference.

The benchmarking of the remaining approaches, STdeconvolve and the reference-dependent methods SPOTlight, RCTD, and spatialDWLS has been expanded in the following ways:

To compare STdeconvolve with these reference-based deconvolution approaches, we again used our simulated 100 μm^2 pixel resolution ST data using our single-cell resolution MERFISH data of the MPOA. The ground truth cell-type proportions of each multi-cellular pixel was based on the 9 major cell-types previously annotated in the original MERFISH publication. We quantitated the deconvolution performance of the different deconvolution approaches by calculating the Root-Mean-Squared-Error (RMSE) of a given approach's predicted cell-type proportions for each pixel in a manner similar to how benchmarking was done in the spatialDWLS paper. With respect to these reference-based deconvolution approaches however, a single-cell transcriptomics reference was required for deconvolution. We therefore trained these approaches using three different sets of references and compared the results to our reference-free approach.

The first reference used was the original single-cell MERFISH data used to construct the simulated ST data, which optimally matched the ST dataset and served as an upper bound of performance. The second set of references was the single-cell MERFISH data but with missing cell-types. The third was an independent scRNA-seq reference, more akin to what would typically be available for such deconvolution scenarios. When we use the ideal reference, supervised approaches RCTD and spatialDWLS had slightly smaller RMSEs than STdeconvolve, however, STdeconvolve had a lower RMSE than SPOTlight. DSTG was unable to recover distinct cell-types and was omitted from further analysis. When we use less ideal single cell transcriptomics references such as references with missing cell-types or an independent scRNA-seq reference, the RMSE of all reference-based deconvolution approaches significantly increased and were higher than that of STdeconvolve. This suggests that STdeconvolve provides potentially superior performance when suitable single-cell references are not available. We have added these findings to the Results under section **"STdeconvolve achieves competitive performance to reference-based, supervised deconvolution approaches"** (Lines 124-169) and have provided the relevant text below for the reviewer's reference.

We next sought to compare the performance of STdeconvolve to existing reference-based, supervised and semi-supervised deconvolution approaches SPOTlight, RCTD, spatialDWLS, and DSTG using our simulated 100 μm^2 resolution ST data of the MPOA. As described previously, these approaches require a single-cell transcriptomics reference for deconvolution. As an ideal single-cell transcriptomics reference, we used the original single-cell MERFISH data that was used to construct the simulated ST data (Supplementary Figure S3A, Supplementary Methods). We again quantified the performance of each approach using the RMSE of the deconvolved cell-type proportions compared to ground truth across simulated pixels. DSTG was unable to deconvolve distinct cell-types in the data and was omitted from further comparison (Supplementary Figure S3A). In general, we find the performance of STdeconvolve to be comparable to these reference-based deconvolution approaches when such an ideal single-cell transcriptomics reference is used (Figure 2E-F).

One potential limitation of such existing reference-based deconvolution approaches is their reliance on a suitable single-cell transcriptomics reference. We thus sought to evaluate the performance of these reference-based deconvolution approaches when a suitable single-cell reference is not available. To this end, we removed excitatory and inhibitory neuronal cell-types to simulate a less suitable single-cell transcriptomics reference (Supplementary Methods). We then deconvolved the simulated ST data of the MPOA using each reference-based deconvolution approach with this new reference and computed the RMSE across pixels. Because STdeconvolve does not use a reference, its performance does not change. However, the performance for all reference-based deconvolution approaches resulted in a significantly higher RMSE (Diebold-Mariano p -value $< 2.2 \times 10^{-16}$) than STdeconvolve (Figure 2G). Likewise, pixels previously comprised of neurons were now erroneously predicted by reference-based deconvolution approaches to be comprised primarily of immature oligodendrocytes (Supplementary Figure S3B). In addition, we evaluated the performance of each reference-based deconvolution approach after removing rarer ependymal cells from the single-cell transcriptomics reference. Again, given this less suitable single-cell

transcriptomics reference, pixels previously comprised of ependymal cells were now erroneously predicted by reference-based deconvolution approaches to be comprised primarily of astrocytes (Supplementary Figure S3C). Thus, the performance of reference-based deconvolution approaches is sensitive to differences in cell-type composition between the ST data and the single-cell transcriptomics reference used.

Likewise, such an ideal single-cell transcriptomics reference that optimally matches the cell-type composition and measurement sensitivities of the ST data to be deconvolved may not be available. Therefore, this ideal MERFISH MPOA single-cell transcriptomics reference likely provides an upper bound on performance for reference-based deconvolution approaches. To provide a more realistic evaluation of performance for reference-based deconvolution approaches, we sought to deconvolve our simulated ST data of the MPOA using a scRNA-seq reference from a mouse brain atlasing effort¹². Again, as a reference-free deconvolution approach, the performance of STdeconvolve does not change. However, again, the performance for all reference-dependent methods resulted in a significantly higher RMSE (Diebold-Mariano p -value $< 2.2 \times 10^{-16}$) than STdeconvolve (Figure 2H, Supplementary Methods). Thus, STdeconvolve achieves comparable performance to reference-based, supervised deconvolution approaches when an ideal single-cell transcriptomics reference is used, and potentially better performance when an ideal single-cell transcriptomics reference is not used.

Figure 2E-H. Deconvolution of simulated ST data. E) Root-mean-square-error (RMSE) of the deconvolved cell-type proportions compared to ground truth for STdeconvolve, F) for supervised deconvolution approaches using the ideal single cell transcriptomics MERFISH MPOA reference, G) for supervised deconvolution approaches using the single cell transcriptomics MERFISH MPOA reference with missing neurons, and H) for supervised deconvolution approaches using a brain single-cell RNA-seq reference.

Major Comment 4-3: Is the spatial structure of ST data used anywhere? If not, the problem that STdeconvolve attempts to solve seems to be in the purview (except for possibly minor sample size considerations) of general unsupervised reference-free joint deconvolution of multiple bulks, for which several methods exist (e.g., CDSeq, TOAST, UNDO, BayesCCE) and against which STdeconvolve should be benchmarked. (BayesCCE was developed for methylation data, but features essentially the same generative model as it applies for RNA-Seq, and so should be tried.) Moreover, if spatial structure is indeed not accounted for, then any kind of single-cell dataset (not only ST) could be artificially divided up into different pixels, used as ground truth data and hence benchmarked against.

These simulated pixels could also be manually tuned to satisfy more strongly or weakly LDA's assumptions (e.g., sparsity), with resultant effect on performance analyzed.

We thank the reviewer for bringing up this point. We note that STdeconvolve does not take into account the spatial relationship between pixels. We appreciate the reviewer's idea to take advantage of this aspect in generating additional simulations to assess the performance and other characteristics of STdeconvolve using single-cell RNA-seq data. As the reviewer suggested, we have now included additional simulations using mixtures of single-cell RNA-seq data to simulate ST data. We have now included additional simulations in the Results demonstrating that deconvolution can identify distinct insights compared to clustering in section "Deconvolution provides distinct insights compared to clustering analysis" (lines 203-244), that STdeconvolve can recover perturbation specific gene expression in section "STdeconvolve recovers perturbation specific gene expression profiles" (lines 171-201), that

STdeconvolve can distinguish excluded and infiltrating immune cells when simulating spatial organizations of excluded and infiltrating immune infiltrates in section “STdeconvolve identifies immune infiltrates in breast cancer” (lines 386-426)”. Additionally, we also demonstrated limitations of STdeconvolve through simulations of deconvolution failures, noted in the Discussion on lines 460-463. Instructions for recapitulating these simulations are included in the revised Supplementary Methods in section “**Simulating ST data using mixtures of cells from scRNA-seq**” (lines 232-301), which we have included below for the reviewer’s convenience.

Simulated ST datasets of 900 pixels were generated in which each pixel was the combined gene counts of a mixture of different cell-types up to 8 total individual cells. Individual cells were sampled from cell-types belonging to a scRNA-seq dataset using the original provided annotations²⁴ (GSE150580).

Simulating age perturbed ST data

Individual cells were sampled to generate a simulated ST dataset of “young” luminal cells and macrophages and a simulated ST dataset of “aged” luminal cells and macrophages. Genes detected in more than 5% but less than 100% of pixels were removed. Feature selection was performed to select genes that were significantly overdispersed using a general additive model with a basis of 5 and an adjusted p -value cutoff of $1e^{-4}$, which resulted in 877 overdispersed genes for the “young” input corpus. For the “aged” input corpus, overdispersed genes with an adjusted p -value cutoff of $1e^{-10}$ were chosen, which resulted in 726 overdispersed genes. STdeconvolve was applied to the “aged” and “young” simulated ST datasets using $K=2$ cell-types and the resulting deconvolved transcriptional profiles were correlated with the average gene expression profiles of the ground truth cell-types of the scRNA-seq dataset using Pearson’s correlation. Differentially expressed genes between the annotated deconvolved “young” and “aged” deconvolved macrophage cell-types were determined by dividing the deconvolved transcriptional profiles of the “aged” macrophages by the “young” macrophages and then \log_2 -transforming the data.

Deconvolution compared to clustering analysis of simulated cell-type mixtures

A simulated ST dataset containing pixels with mixtures of 3 cell-types was generated by sampling individual cells from “young” luminal cells, macrophages, and pericytes cell-types of the scRNA-seq reference and combining the gene counts. Genes with less than 10 total reads and detected in less than 10 pixels were removed. For the transcriptional clustering, gene counts were normalized to counts-per-million and \log_{10} transformed with a pseudo-count of 1. Dimensionality reduction using PCA was performed. Graph-based cluster detection using Louvain clustering²⁹ was performed using the top 30 principal components with the maximum number of nearest neighbors equal to 300, resulting in the assignment of pixels to 2 clusters. For deconvolution with STdeconvolve, we feature selected for genes detected in more than 5% but less than 100% of pixels and were significantly overdispersed using a general additive model with a basis of 5 and an adjusted p -value cutoff of 0.05 and used the top 1000 overdispersed genes with the lowest adjusted p -values. STdeconvolve was then applied with $K=3$ to deconvolve 3 cell-types. Deconvolved transcriptional profiles were correlated with the average gene expression profiles of the ground truth cell-types of the scRNA-seq dataset using Pearson’s correlation and cell-types were matched to ground truth cell-types that had the highest correlations.

Simulating and deconvolving immune infiltrated vs excluded ST data

Two simulated ST datasets containing pixels with mixtures of luminal and macrophage cells were generated by sampling individual cells from “young” luminal and “young” macrophage cell-types of the scRNA-seq reference and combining the gene counts. In the “infiltrated” ST simulation, cells were mixed such that the proportion of macrophage followed a gradient from 100% on either the left and right side of the dataset and converged to 0% at the middle. In the “excluded” ST simulation, macrophages were enriched in the 5 columns of pixels on either the left or right and 0% otherwise. For each row, where present, macrophage proportions alternated from 100% to 75%. For both simulations, genes with less than 100 gene counts detected were removed in addition to genes detected in more than 5% but less than 100% of pixels. Feature selection was performed to select genes that were significantly overdispersed using a general additive model with a basis of 5 and an adjusted p -value cutoff of 0.05 and the top 800 or 1000 most significant overdispersed genes were kept for either the infiltrated or excluded simulation, respectively. STdeconvolve was then applied to each simulation with $K=2$ to deconvolve 2 cell-types. Deconvolved transcriptional profiles were correlated with the average gene expression profiles of the ground truth cell-types of the scRNA-seq dataset using Pearson’s correlation and cell-types were matched to ground truth cell-types that had the highest correlations.

Simulating and deconvolving ST data with uniform cell-type distributions

A simulated ST dataset containing pixels with uniform proportions of 2 cell-types was generated by sampling individual cells from “young” luminal cells and macrophages of the scRNA-seq reference and combining the gene counts. Genes with less than 100 total reads and detected in less than 100 pixels were removed. For deconvolution with STdeconvolve, we feature selected for genes detected in more than 5% but less than 100% of pixels and were significantly overdispersed using a general additive model with a basis of 5 and an adjusted p -value cutoff of 0.05 and used the top 1000 overdispersed genes with the lowest adjusted p -value. STdeconvolve was then applied with $K=2$ to deconvolve 2 cell-types. Deconvolved transcriptional profiles were correlated with the average gene expression profiles of the ground truth cell-types of the scRNA-seq dataset using Pearson’s correlation and cell-types were matched to ground truth cell-types that had the highest correlations.

We reiterate that ST data has many assumptions and features distinct from bulk RNA-seq. This also extends to our simulations, where we limit the number of cells in each simulated pixel to only 8 cells to be more consistent with the number of cells more typically captured in each spatially resolved pixel in ST data.

One notable difference is that ST data is UMI-based whereas bulk RNA-seq quantifications are biased by gene length. Because of this gene length bias, previous LDA-based deconvolution approaches such as CDSeq have included corrections in their models to accommodate these differences in effective gene length on the observed gene expression measurements³⁰. As such, when we apply CDSeq to our immune “infiltrated” simulated ST data, we observe a poor correlation between the deconvolved gene expression profiles and the ground truth gene expression profiles (**Reviewer Figure 1**).

Reviewer Figure 1. CDSeq deconvolved versus ground truth transcriptional profiles of macrophage and luminal cells. Pearson’s correlation coefficient (R) between the two transcriptional profiles are indicated above each plot.

Further, as the reviewer notes, there are sample size differences between ST data and bulk RNA-seq deconvolution tasks. We have now also greatly expanded the range of ST datasets analyzed, including real ST data with as many as 2702 pixels. We have further now included a benchmark of runtime on these large datasets to show that STdeconvolve is amenable to analyzing them. In contrast, bulk RNA-seq analysis approaches understandably did not face such large sample size challenges. As such, applying CDSeq to deconvolve our simulated ST data with 900 pixels (a typical size for ST data) took over 2.5 hours compared to approximately 30 seconds by STdeconvolve. However, given that CDSeq was not intended for analysis of ST data, we did not feel it would be fair to include these results in the revised manuscript.

Finally, we agree with the reviewer that this new simulation framework could allow the simulated pixels to be manually tuned to evaluate resultant effects on performance. To this end, we have now included a simulation where LDA’s assumptions are indeed violated to demonstrate a case where STdeconvolve fails. Specifically, we show that if genes do not exhibit variability across pixels due to a homogeneous or uniform proportional representation of cell-

types across pixels STdeconvolve may fail to deconvolve distinct cell-types. For this, we simulated ST pixels in which macrophages and luminal cells were mixed at uniform proportions across the dataset. We find that indeed STdeconvolve struggles to achieve high deconvolution accuracy (Supplemental Figure S20). These findings have been included in the discussion on lines 460-463 with relevant text provided below for the reviewer’s convenience.

Further, LDA modeling attempts to identify tightly occurring, and ideally non-overlapping groups of genes in the pixels as cell-types. In this manner, if genes do not exhibit variability across pixels due to a homogeneous or uniform proportional representation of cell-types across pixels (Supplementary Figure S20), STdeconvolve may fail to deconvolve distinct cell-types.

Supplementary Figure S20. Deconvolution failures. A) Pie charts indicating the ground truth pixel proportions of luminal and macrophage cells with simulated uniform proportions across pixels. B) STdeconvolve predicted pixel proportions across pixels. C) The ranking of each gene based on its expression level in the transcriptional profiles of

the deconvolved cell-types, compared to its gene rank in the transcriptional profile of the matched ground truth cell-types. D) Heatmap of Pearson's correlations between the transcriptional profiles of the ground truth cell-types and deconvolved cell-types. E-F) Observed ground truth pixel proportions compared to deconvolved pixel proportions for the macrophage and luminal cells.

Major Comment 4-4: The spatial correlation between pixels violates the LDA assumption. This issue should be addressed.

We thank the author for their comment. We clarify that the spatial structure of the ST data itself is not used in our approach.

Major Comment 4-5: The theta and beta inference appear to be inconsistent; that is, even in the limit of infinite data, they might be misestimated. Intuitively, this should be due to the normalization of all cell types' expression vectors onto the simplex (i.e., normalizing their library sizes to unity)—in the worst case two cell types are perfect multiples of each other (in which case every gene is differentially expressed!), but the current LDA setup seems unable to tell them apart. More generally, in the large data limit, the fraction of reads coming from cell type k for a fixed gene g should be (for a single pixel d):

$$\theta_{d(k)} e_g(k) / \left(\sum_{j=1}^K \theta_{d(j)} e_g(j) \right),$$

where $e(k)$ is the unnormalized expression vector of cell type k . Since the true $\beta(k) = e(k) / \|e(k)\|_1$, the same quantity under LDA becomes

$$\left(\theta_{d(k)} e_g(k) / \|e(k)\|_1 \right) / \left(\sum_{j=1}^K \theta_{d(j)} e_g(j) / \|e(j)\|_1 \right),$$

which (unless the library sizes are all identical to begin with) is different from the above.

We thank the reviewer for this insightful point. We remark that the reviewer is correct in noting that in our current setup, two cells that express the same genes but are multiples of each other would indeed be interpreted as the same cell-type and LDA would be unable to tell them apart. We have now included a sentence noting this limitation in the Discussion (lines 460-463), provided below for the reviewer's reference.

In this manner, if genes do not exhibit variability across pixels due to a homogeneous or uniform proportional representation of cell-types across pixels (Supplementary Figure S20), STdeconvolve may fail to deconvolve distinct cell-types.

However, we would like to clarify that in ST data, all multicellular pixels are sequenced in the same experiment and thus generally result in comparable sequencing depth and library size within the same order of magnitude across all pixels. This contrasts with bulk RNA-seq where each bulk RNA-seq library may be sequenced to very different read depths resulting in library sizes that differ by multiple orders of magnitude. We have mentioned our assumptions about LDA applied to ST data and that pixels will have been approximately sampled to the same depth in the Introduction (Lines 77-81).

Briefly, these include i) the limited number of cells and cell-types represented in each ST pixel, ii) the limited impact of batch effects on the measured gene expression across pixels, iii) the large number of pixels compared to cell-types, and iv) the likely heterogeneity of cell-type proportional distribution across pixels in tissues.

Major Comment 4-6: Analyzing the robustness of supervised methods to perturbations is important, and its results should indeed inform the value of unsupervised methods, but might profit from a more thorough discussion. E.g., if I understand correctly, Figure S5B kicks the two most abundant cell types out of the reference, for which significant reductions in accuracy aren't surprising. What happens if less abundant cells are removed?

We thank the reviewer for this great point and have expanded our analysis of the limitations of supervised methods in a number of ways. As mentioned previously, we compared STdeconvolve to the supervised deconvolution approaches using our simulated ST dataset of the MPOA using single-cell MERFISH data as the ground truth. Supervised approaches were assessed using three different sets of single-cell references. One of these involved training the supervised deconvolution approaches with the ideal single-cell MERFISH reference after removing neuronal cells. Here, we note that immature oligodendrocytes were erroneously assigned at much higher abundances across pixels. Immature oligodendrocytes have transcriptional profiles that positively correlate with the neuronal cells, which may explain why the supervised methods assigned them in place of the missing excitatory and inhibitory cells (Supplementary Figure 2C). We have now also assessed the supervised approaches after removing less abundant ependymal cells as the reviewer has suggested. Likewise, we note that pixels previously comprised of ependymal cells are now erroneously predicted to be primarily composed of astrocytes. Again, astrocytes are also transcriptionally correlated with ependymal cells. These findings are reported in the revised Results section **“STdeconvolve achieves competitive performance to reference-based, supervised deconvolution approaches”** (Lines 124-169).

Additionally, we also note that while supervised reference-dependent methods may recover cell-type proportions in ST data, they currently do not deconvolve cell-type-specific gene expression profiles. This can be limiting in cases where perturbations may induce cell-type-specific transcriptional changes in ST data that would not be identifiable by current reference-based deconvolution approaches unless perturbation-matched single-cell transcriptomics references are used. Further, while the availability of numerous scRNA-seq datasets online, these datasets primarily represent collections of healthy tissues. In contrast, STdeconvolve being reference-free can estimate cell-type transcriptional profiles in a manner that is not constrained by the expression profiles of specific cell-types defined in single-cell transcriptomics references. To demonstrate this, we have now explored the potential of STdeconvolve in detecting these perturbation-driven cell-type-specific gene expression changes using simulated ST data from mixtures of single cells assayed by scRNA-seq. We simulated ST data of aged tissue using mixtures of aged macrophages and other luminal cells and ST data of young tissue using mixtures of young macrophages and other luminal cells. We found that STdeconvolve recovered transcriptional profiles were highly correlated with the matched ground truth gene expression profiles from scRNA-seq in all cases. Comparison of the deconvolved transcriptional profiles identified differentially expressed genes between young and aged macrophages consistent with the previous study. We hope that this new simulation and analysis better demonstrates the value of unsupervised deconvolution approaches in the context of perturbations. We have included these findings in the revised Results section **“STdeconvolve recovers perturbation specific gene expression profiles”** (Lines 171-201) and have included the relevant excerpts below for the reviewer’s reference.

Though reference-based deconvolution approaches may accurately recover cell-type proportions in ST data, they currently do not deconvolve cell-type specific gene expression profiles. Nonetheless, perturbations may induce cell-type-specific transcriptional changes in ST data that would not be identifiable by current reference-based deconvolution approaches unless perturbation-matched single-cell transcriptomics references are used. While the availability of scRNA-seq references grows due to single-cell atlasing initiatives, these datasets primarily represent collections of healthy tissues^{5,31-34}. As such, there is a particular scarcity of suitable scRNA-seq references available for reference-based deconvolution of ST data in the context of disease and other perturbations.

In contrast to current reference-based deconvolution approaches, STdeconvolve can estimate cell-type transcriptional profiles in a manner that is not constrained by the expression profiles of specific cell-types defined in single-cell transcriptomics references. We therefore sought to explore the potential of STdeconvolve in detecting these perturbation-driven cell-type-specific gene expression changes using simulated ST data from mixtures of single cells assayed by scRNA-seq (Figure 3A, Supplementary Methods). Briefly, we took advantage of scRNA-seq data previously collected from mammary tissues of aged and young mice²⁴. Previous transcriptional clustering analysis revealed a subpopulation of macrophages with age-associated gene expression changes. Specifically, aged macrophages upregulated *Cd274* and *Clec4d*, and downregulated *Coro1a* compared to young macrophages. Therefore, we simulated ST data of aged tissue using mixtures of aged macrophages and other luminal cells and ST data of young tissue using mixtures of young macrophages and other luminal cells (Figure 3B). We then sought to evaluate the ability of STdeconvolve to recover these age-associated gene expression changes in macrophages (Supplementary Methods). Applying STdeconvolve using $K=2$ cell-types to the simulated ST data of both aged and young tissue, we found that the deconvolved transcriptional profiles were highly correlated with the matched ground truth gene expression profiles from scRNA-seq in all cases (Supplementary Figure S4A-B). Further, when we

compared the deconvolved transcriptional profiles of aged versus young macrophages, we were able to identify upregulated genes included *Cd274* and *Clec4d*, and downregulated genes included *Coro1a*, consistent with the original study (Figure 3C). Thus, STdeconvolve can potentially recover perturbation-driven cell-type-specific gene expression changes in ST data.

Figure 3A-C. Comparing clustering versus deconvolution analysis for ST data. A) Overview of simulation approach. Starting from a single-cell RNA-seq clustering result visualized as a 2D tSNE embedding with cells colored by cell-type (top), gene expression counts from cells are combined to simulate cell-type mixtures, with the proportions of cell-types are represented as pie charts for each arbitrary spatial pixel (bottom). B) Simulated ST datasets of aged and young tissues using mixtures of aged macrophages with aged luminal cells and young macrophages with young luminal cells respectively represented as pie charts for each simulated ST pixel. C) Bar chart of log₂ fold-change for deconvolved aged macrophage versus deconvolved young macrophage gene expression. Select genes are highlighted in red.

Supplementary Figure S4. STdeconvolve deconvolves simulated ST data of aged and young tissues. A) The ranking of each gene based on its expression level in the transcriptional profiles of the deconvolved cell-types,

compared to its gene rank in the transcriptional profile of the matched ground truth cell-type for simulated ST data of young (top) and aged (bottom) tissues. B) Heatmap of Pearson's correlations between the transcriptional profiles of the ground truth cell-types and deconvolved cell-types from the simulated ST data of young (top) and aged (bottom) tissues.

Major Comment 4-7: There is no discussion of runtimes or the computational resources needed to run the algorithm.

We thank the reviewer for bringing up this important point about runtime. To assess the computational resources needed to run STdeconvolve for ST datasets of varying sizes, we have now assessed both the runtime and total memory usage of STdeconvolve in terms of varying the number of genes and number of pixels. We find that the runtime increases linearly with either the number of pixels or genes, as well as the number of cell-types to be deconvolved. As a specific example, deconvolution of 12 cell-types using all 2072 pixels and 1000 feature selected genes in a typical deconvolution analysis of a 10X Visium dataset completed in approximately 30 minutes on a personal computer, which we believe is a reasonable time cost and is comparable with previously published deconvolution approaches⁷. Further, we now allow for parallelization, which may further improve scalability. Importantly, the total memory cost was also minimal, with total memory reaching approximately 170 MBs for the full sized 10X Visium analysis. These results are now presented as Supplementary Figure S15 and we have provided the relevant text below for the reviewer's reference.

As the resolution of ST data improves, the number of spatially resolved pixels and cell-types represented in the data will presumably also increase. We therefore sought to evaluate the scalability of STdeconvolve in anticipation of these increasingly larger datasets. To this end, we benchmarked the runtime and total memory usage by STdeconvolve when deconvolving varying numbers of cell-types using varying numbers of genes across varying numbers of pixels (Methods). We found that both the runtime and memory usage by STdeconvolve increased linearly with the number of pixels and genes in the input dataset (Supplementary Figure S15A) and is comparable to existing reference-based deconvolution methods when applied to current ST datasets⁷. Likewise, runtime scales with the number of deconvolved cell-types K in the input dataset though memory usage remains stable (Supplementary Figure S15B). To enhance runtime efficiency, STdeconvolve has built in parallelization. In this manner, we anticipate that STdeconvolve will be amenable to larger ST data.

Supplementary Figure S15. Runtime and memory usage by STdeconvolve. A) Runtime as a function of dataset size in terms of the number of pixels (top) or genes (bottom), and number of cell-types K . For scaling pixels, the top 1000 most significant overdispersed genes were selected and kept constant. For scaling genes, all 2702 pixels in the dataset were used and kept constant. B) Memory usage as a function of dataset size in terms of scaling pixels (top) or genes (bottom), and number of cell-types K . Again, for scaling pixels, the top 1000 most significant overdispersed genes were selected and kept constant. For scaling genes, all 2702 pixels in the dataset were used and kept constant.

Major Comment 4-8: In the breast cancer application, would it be possible to apply RCTD and SPOTlight using single-cell reference datasets from a different breast cancer sample in order to quantitatively compare the performance, rather than qualitatively accounting for the tumor microenvironment heterogeneity?

We thank the reviewer for this interesting suggestion. We have now applied RCTD and SPOTlight using a single-cell RNA-seq reference dataset of DCIS from a different breast cancer sample to quantitatively compare performance as suggested. Briefly, we downloaded single-cell RNA-seq dataset³⁵ (GSE148673), which was also profiled on the Cancer Single-cell Expression Map³⁶ (Sample ID DCIS-028-01-1A). Following the default pipeline on the Cancer Single-cell Expression Map, 1114 single cells were profiled from a breast ductal carcinoma *in situ* tumor after filtering for cells with less than 5000 and more than 200 unique features. Clustering analysis was performed to identify 6 cell-types, which we recapitulate as shown below (**Reviewer Figure 2**). It is worth noting that most cells profiled in this tumor represent only one major transcriptionally distinct population of malignant cells. Further, no T cells and other lymphocytes were profiled.

Reviewer Figure 2. Clustering analysis of 1114 cells in a single-cell RNA-seq reference dataset of DCIS (GSE148673).

We then used these annotations and single-cell transcriptional profiles as the reference for RCTD and SPOTlight. Consistent with our deconvolution, RCTD and SPOTlight find that monocytes are enriched within the pixels corresponding to our deconvolved cell-type 15 (**Reviewer Figure 3**). However, as shown in our other analyses, reference-based approaches are sensitive to missing cell-types in the reference. Therefore, it is unclear whether monocytes are truly localized to those pixels or whether monocytes represent the cell-type most transcriptionally similar in the reference to the true cell-type.

Reviewer Figure 3. Heatmap of Pearson's correlations between the deconvolved cell-type pixel proportions of supervised methods and STdeconvolve across 1029 pixels of the BCL ST dataset.

Further, RCTD and SPOTlight fail to distinguish between the two known malignant subtypes (ductal and invasive) in this ST data (**Reviewer Figure 4**). This is likely because the provided single-cell reference only contains one population of malignant cells.

Reviewer Figure 4. Heatmap of Pearson’s correlations between the deconvolved cell-type pixel proportions of supervised methods and ground truth pathologist pixels annotations across 1029 pixels of the BCL ST dataset.

As such, results from RCTD and SPOTlight using a single-cell reference dataset from a different breast cancer sample appear consistent with STdeconvolve with respect to the spatial distribution of an inflammatory signature, but quantitative comparison is challenging due to a lack of ground truth.

Minor Comment 4-1: The total bifurcation of results in the main text and all methodological discussion in the supplement seems rather odd. I would prefer a short overview of the model and its strengths/weaknesses in the main text.

We thank the reviewer for pointing this point. The original format of the paper was a Brief Communication, which greatly restricted our ability to include a methodological discussion in the main text. We have now revised the manuscript as a full Research Article format and have included a short overview of the model in the main text (Lines 66-75), provided below for the reviewer reference.

Here, we developed STdeconvolve (available at <https://github.com/JEFworks-Lab/STdeconvolve> and as Supplementary Software) as a reference-free, unsupervised approach for deconvolving multi-cellular pixel resolution ST data (Figure 1). STdeconvolve builds on latent Dirichlet allocation (LDA), a generative statistical model commonly used in natural language processing for discovering latent topics in collections of documents. In the context of natural language processing, given a count matrix of words in documents, LDA infers the distribution of words for each topic and the distribution of topics in each document. In the context of ST data, given a count matrix of gene expression in multi-cellular ST pixels, STdeconvolve applies LDA to infer the putative transcriptional profile for each cell-type and the proportional representation of each cell-type in each multi-cellular ST pixel (Methods).

Figure 1. Overview of STdeconvolve. A) STdeconvolve takes as input a spatial transcriptomics (ST) gene counts matrix of D pixels (rows) by N genes (columns). A matrix of spatial coordinates for each of the D pixel can also be used for visualization. B) STdeconvolve first feature selects genes for deconvolution, such as genes with counts in more than 5% and less than 95% of the pixels, and overdispersed across the pixels. STdeconvolve then guides the selection of the optimal number of cell-types to be deconvolved, K . STdeconvolve finally applies LDA modeling. A graph representation of LDA modeling and the parameters to be learned is shown. Shaded circle indicates observed variables and clear circles indicate latent variables. C) STdeconvolve outputs two matrices: (1) β , the deconvolved transcriptional profile matrix of K cell-types over N' feature selected genes, and (2) θ , the proportions of K cell-types across the D pixels. The proportion of deconvolved cell-types can then be visualized across the pixels.

We have also now included a discussion on the weaknesses of our model in the Discussion of the main text (Lines 450-467), provided below for the reviewer's reference.

Though we have shown that STdeconvolve can effectively recover cell-type proportions and transcriptional profiles in simulated and real ST data, its use of LDA modeling relies on several underlying assumptions, which may present limitations when these assumptions are not satisfied. Notably, the performance of LDA in accurately deconvolving cell-types depends on the size of the dataset with respect to the number of pixels and the number of genes¹¹. As such, deconvolution accuracy generally decreases for ST data containing fewer than 10 pixels (Supplementary Figure S19). While we have generally found the number of pixels in most ST datasets to be well beyond 10 pixels after quality control filtering, the application of ST to profile tissue slivers or other thin structures covering only a few pixels may present challenges to deconvolution by STdeconvolve. Further, LDA modeling attempts to identify tightly occurring, and ideally non-overlapping groups of genes in the pixels as cell-types. In this manner, if genes do not exhibit variability across pixels due to a homogeneous or uniform proportional representation of cell-types across pixels (Supplementary Figure S20), STdeconvolve may fail to deconvolve distinct cell-types. Likewise, if the gene expression in the ST data is too sparse with high rates of stochastic drop-outs³⁷, then the LDA model may struggle to identify distinct groups of co-expressed genes and as such STdeconvolve may also struggle to deconvolve distinct cell-types as well. Still, when such failures happen, STdeconvolve will indicate to users when distinct cell-types are not detected.

Supplementary Figure S19. Accuracy of deconvolution by STdeconvolve based on the number of pixels in the input dataset. Each point is a replicate in which a random sample of pixels from the simulated MERFISH MPOA ST was used as the input into STdeconvolve. For each replicate, the mean RMSE was computed by averaging the RMSEs of the difference in deconvolved cell-type and ground truth cell-type proportions for each pixel. The solid line represents the mean RMSE of the replicates for each given number of pixels sampled.

Supplementary Figure S20. Deconvolution failures. A) Pie charts indicating the ground truth pixel proportions of luminal and macrophage cells with simulated uniform proportions across pixels. B) STdeconvolve predicted pixel proportions across pixels. C) The ranking of each gene based on its expression level in the transcriptional profiles of

the deconvolved cell-types, compared to its gene rank in the transcriptional profile of the matched ground truth cell-types. D) Heatmap of Pearson’s correlations between the transcriptional profiles of the ground truth cell-types and deconvolved cell-types. E-F) Observed ground truth pixel proportions compared to deconvolved pixel proportions for the macrophage and luminal cells.

Minor Comment 4-2: Please clarify how genes are selected for LDA. How were spatial variability and its expectation calculated? Based on the description in “Gene selection for the LDA model,” it seems that only the genes with global spatial expression variability are selected, but genes with local spatial expression variability could be important as well, and the selection of genes should be based on their cell-type-specific DE, rather than spatial DE, if the goal is to deconvolve cell types.

We thank the author for this point of clarification. First, we do not use the spatial structure of the ST data and thus are not looking for genes with global or local spatial expression variability. Instead, we assume that the proportion of cell-types will vary across pixels and thus differences in their cell-type-specific transcriptional profiles manifest as overdispersed genes across pixels in the dataset. Because we do not use a reference and therefore do not know what cell-types may be present in the ST dataset, STdeconvolve feature selects for significantly overdispersed genes, or genes with higher-than-expected expression variance across pixels, as a means to detect transcriptionally distinct cell-types²³. These details are now clarified in the Methods section “**Selection of genes for LDA model**” (Lines 534-553) provided below for the reviewer’s reference.

Latent cell-types are best discovered by LDA modeling if cell-type specific marker genes are included in the input ST data while genes whose expression is shared across cell-types are excluded. Therefore, to filter for genes that are more likely to be specifically expressed in particular cell-types to improve cell-type deconvolution by LDA, STdeconvolve first removes genes that are not detected in a sufficient number of pixels. By default, genes detected in less than 5% of pixels are removed. Likewise, STdeconvolve also removes genes that are expressed in all pixels. By default, genes detected in 100% of pixels are removed. STdeconvolve then selects for significantly overdispersed genes, or genes with higher-than-expected expression variance across pixels, as a means to detect transcriptionally distinct cell-types²³. We assume that the proportion of cell-types will vary across pixels and thus differences in their cell-type-specific transcriptional profiles manifest as overdispersed genes across pixels in the dataset.

If there are too many genes included in the input ST data, LDA may also struggle to identify non-overlapping clusters composed of distinct combinations of co-expressed genes. In these circumstances, users may modulate the number of informative genes included in the input matrix to ensure LDA convergence. By default, only the top 1000 most overdispersed genes are retained in the input ST data because we note that deconvolution accuracy in general stabilizes for larger numbers of informative features (Supplementary Figure S22). Additional gene filtering or cell-type specific marker genes to include in the input ST data may also be augmented by the user.

Supplementary Figure S22. Deconvolution accuracy by number of randomly sampled feature selected genes.

Each point represents the mean RMSE based on the deconvolved versus ground truth cell-type proportions across

900 simulated pixels for 3 replicates. Vertical bars represent the standard deviation from the mean. Genes were sampled from the top 1000 most significant overdispersed genes.

Minor Comment 4-3: How many genes do we need to distinguish between cell types accurately?

We thank the reviewer for raising this interesting point. We note that LDA seeks to represent latent “topics”, or cell-types, as ideally non-overlapping groups of co-expressed, or frequently co-occurring, genes in different pixels. Several parameters will likely affect the ability of LDA to achieve this grouping of genes into distinct cell-types, which include the number of cell-types to be deconvolved, how distinct they are transcriptionally, how variable cell-types are across ST pixels, and how the gene expression values were measured, e.g., targeted gene panel or transcriptome-wide profiling. For example, in our simulated ST data based on the single-cell resolution MERFISH data of the MPOA, a panel of 135 genes was specifically curated to optimally distinguish between 7 major classes of non-neuronal cell-types in addition to excitatory and inhibitory neuronal subtypes. In this particular situation, given only 135 genes, STdeconvolve was still able to deconvolve underlying cell-types. However, we note that real multi-cellular pixel resolution ST data is not limited to pre-selected gene panels known to optimally distribute between cell-types. These thoughts have been included as part of “**Supplementary Note 3**” (Lines 145-179 in Supplementary Materials) and relevant excerpts are included below for the reviewer’s reference.

For this MERFISH dataset, a specific panel of 135 genes previously chosen to optimally distinguish primarily between neuronal subtypes. As such, only 33 remaining genes were available to distinguish between all major cell-types⁸. Therefore, these chosen genes represent a specific subset of overdispersed genes rather than all overdispersed genes that are skewed towards distinguishing between neurons, even though many more overdispersed genes may exist for distinguishing microglia and pericytes, for example. In this manner, neuronal subtypes may appear more transcriptionally distinct than certain cell-types for which few markers were included. Indeed, when we applied STdeconvolve to the simulated MERFISH MPOA ST data limited to only these 135 genes, deconvolved cell-types such as cell-types X2 and X8 both matched to excitatory neurons while cell-types X4 and X7 both matched to inhibitory neurons. Given that excitatory and inhibitory neurons could previously be further subdivided into finer neuronal sub-types, we sought to evaluate whether these deconvolved cell-types that matched to the excitatory and inhibitory neurons could represent these additional finer neuronal sub-types. To test this hypothesis, we further partitioned the ground-truth excitatory and inhibitory cell-types into additional sub-types based on previous annotations, resulting in 76 total non-neuronal cell-types and neuronal subtypes⁸. Comparing the deconvolved transcriptional profiles of X2, X4, X7, and X8 to the ground truth transcriptional profiles of the 76 non-neuronal cell-types and neuronal subtypes, we indeed observed a correlation between the deconvolved transcriptional profiles with the ground truth transcriptional profiles of neuronal sub-types (Supplementary Figure S21A). We then sought to evaluate whether increasing the number of deconvolved cell-types could recover the finer neuronal sub-types. We therefore applied STdeconvolve with $K=76$ and were able to identify deconvolved cell-types that were highly correlated in terms of both transcriptional profiles and pixel proportions to finer neuronal subtypes as well as rare cell-types such as pericytes and microglia (Supplementary Figure 21B-C). However, as noted previously, the ability for STdeconvolve to deconvolve neuronal subtypes may not be representative of its expected performance in distinguishing between cell sub-types under less biased gene selection.

A**B****C**
Supplementary Figure S21. Comparison of STdeconvolve cell-types to ground truth neuronal subtypes of simulated MERFISH MPOA ST data. A) Heatmap of Pearson's correlations between the transcriptional profiles of the 76 ground truth cell-types and neuronal subtypes in the MERFISH MPOA data and STdeconvolve cell-types that matched to excitatory and inhibitory neuronal major cell-type types. B) Heatmap of Pearson's correlations between the transcriptional profiles of the 76 ground truth cell-types and neuronal subtypes in the MERFISH MPOA data and 76 deconvolved cell-types by STdeconvolve. C) Heatmap of Pearson's correlations between the cell-type pixel proportions of the 76 ground truth cell-types and neuronal subtypes in the MERFISH MPOA data and 76 deconvolved cell-types by STdeconvolve.

Real multi-cellular pixel resolution ST data typically provide transcriptome-wide gene expression profiling. Again, as latent cell-types are best discovered by LDA modeling if cell-type specific marker genes are included in the input ST data while genes whose expression is shared across cell-types are excluded, STdeconvolve filters for genes that are more likely to be specifically expressed in particular cell-types by identifying overdispersed genes across pixels in the dataset, or genes with higher-than-expected expression variance across pixels²³.

To explore whether this feature selection step is sensitive to the number of genes, we have now simulated ST datasets using mixtures of single cells and then evaluated the performance of STdeconvolve in deconvolving these cell mixtures given differing numbers of genes. Specifically, we generated simulated ST datasets in which pixels were composed of mixtures of luminal cells, pericytes, and macrophages sampled from a scRNA-seq dataset²⁴. We sampled random cells of each cell-type into mixtures within multi-cellular pixels based on a Dirichlet distribution with a sparse shape parameter ($\alpha = 0.4$), and the gene counts of cells in each pixel were collapsed. From this simulated ST dataset, we feature selected for overdispersed genes, and from this list, subsampled different numbers of overdispersed genes. With each panel of selected genes, we deconvolved $K=3$ cell-types and computed the RMSE across the pixels. More details are provided in the Supplementary Methods section “**Accuracy of STdeconvolve with respect to number of feature selected genes**” (Lines 513-526 in Supplementary Materials). We repeated this

process for each subsampling three times. Overall, we find that RMSE decreases and thus deconvolution performance improves as the number of genes increase. However, the reduction in RMSE also diminishes as the number of genes increases, suggesting that after a certain point, adding additional genes provides no further improvement. In this manner, these results suggest that our approach may be sensitive to particularly small numbers of informative features, though performance stabilizes for larger numbers of informative features. Given this, we have now also set STdeconvolve to use the top 1000 overdispersed genes by default, though users may still modulate this number depending on their knowledge of the dataset such as regarding the transcriptional distinctness of the cell-types and how many are to be deconvolved. We now indicate this in the revised Methods section “**Selection of genes for LDA model**” (lines 550-554) and have provided the relevant text below for the reviewer’s convenience.

By default, only the top 1000 most overdispersed genes are retained in the input ST data because we note that deconvolution accuracy in general stabilizes for larger numbers of informative features (Supplementary Figure S22).

Additional gene filtering or cell-type specific marker genes to include in the input ST data may also be augmented by the user.

Supplementary Figure S22. Deconvolution accuracy by number of randomly sampled feature selected genes.

Each point represents the mean RMSE based on the deconvolved versus ground truth cell-type proportions across 900 simulated pixels for 3 replicates. Vertical bars represent the standard deviation from the mean. Genes were sampled from the top 1000 most significant overdispersed genes.

Minor Comment 4-4: For the MOB dataset, are there particular reasons why replicates 2,5,8 and 12 were chosen? Why not deconvolve and report summary statistics for all replicates?

We thank the reviewer for this point of clarification. We picked replicates that had varied in the number of features selected as overdispersed genes to test the stability of deconvolved cell-types between different biological replicates. Additionally, we also restricted the replicates we selected to some of the most symmetric and complete ones. Further, we also anticipate most people will run on one sample so analyzing one sample provides a more realistic evaluation of the performance.

Minor Comment 4-5: In a similar vein, aggregating multiple replicates would be an interesting way to explore sample size related concerns somewhat orthogonal to the varying of pixel numbers done in supplementary note 2.

We thank the reviewer for mentioning this point. We have now applied STdeconvolve to ST datasets with varied pixel numbers in several ways. First, we explored the deconvolution of STdeconvolve with respect to the size of the ST dataset. For this, we took subsets of pixels from the 100 μm^2 simulated MPOA ST dataset, we found that deconvolution accuracy generally decreased when the input dataset was less than 10 pixels. While we have generally found the number of pixels in most ST datasets to be well beyond 10 pixels after quality control filtering, the application of ST to profile tissue slivers or other thin structures covering only a few pixels may present challenges to deconvolution by STdeconvolve. These thoughts have been added to the Discussion (Line 453-459) and we have also provided below for the reviewer's reference.

Notably, the performance of LDA in accurately deconvolving cell-types depends on the size of the dataset with respect to the number of pixels and the number of genes¹¹. As such, deconvolution accuracy generally decreases for ST data containing fewer than 10 pixels (Supplementary Figure S19). While we have generally found the number of pixels in most ST datasets to be well beyond 10 pixels after quality control filtering, the application of ST to profile tissue slivers or other thin structures covering only a few pixels may present challenges to deconvolution by STdeconvolve.

Supplementary Figure S19. Accuracy of deconvolution by STdeconvolve based on the number of pixels in the input dataset. Each point is a replicate in which a random sample of pixels from the simulated MERFISH MPOA ST was used as the input into STdeconvolve. For each replicate, the mean RMSE was computed by averaging the RMSEs of the difference in deconvolved cell-type and ground truth cell-type proportions for each pixel. The solid line represents the mean RMSE of the replicates for each given number of pixels sampled.

Additionally, increasing resolution of the simulated MERFISH MPOA ST datasets also increased the number of pixels. For example, the 100, 50, 20, and 10 μm^2 resolution datasets had a total of 3072, 13477, 45978, and 57397 pixels, with the same underlying individual cells combined into the respective pixels.

Lastly, as mentioned previously, we anticipate most users will analyze on one sample instead of combining biological or technical replicates to increase the number of pixels in the input dataset.

Minor Comment 4-6: When comparing proportions, the mean L1 metric might be both more intuitive (mean estimation error per cell type) and rigorous (through its relation to the TV distance) than RMSE.

We thank the reviewer for the suggestion. We have now evaluated the mean L1 metric (mean absolute error or MAE) for comparing proportions and have found the general trends to be essentially the same as RMSE, provided below for the reviewer's reference.

Reviewer Figure 5. Comparing the cell-type pixel proportions between different deconvolution approaches using RMSE or MAE to calculate predicted and ground truth cell-type differences.

Given that the trends are the same, our conclusions are not changed and therefore we have opted not to include the mean L1 metric in the revised manuscript.

Minor Comment 4-7: It would be helpful to see how the error in proportion inference depends on space or other properties of the ground truth distributions. I.e., is there any property of the ground truth mixture that guarantees accurate inference?

We thank the reviewer for this suggestion and have added additional simulations to explore this. To this end, we have now included a simulation to demonstrate properties of the ground truth mixture in which deconvolution fails. Particularly, this can happen if the cell-type proportions are uniform across pixels. Here, STdeconvolve has difficulty deconvolving distinct cell-types and we reference this caveat in the revised Discussion (lines 428-487).

Minor Comment 4-8: Are the N2 and OEC clusters transcriptionally similar? If they are, then the behavior of the supervised methods is expected and, indeed, arguably desirable.

We thank the reviewer for raising this point. N2 and OEC clusters are indeed transcriptionally similar. However, we note that just because two cell-types are transcriptionally similar does not mean that they are transcriptionally identical or have similar functions. While it may be expected for reference-based deconvolution approaches would assign N2 cells to where OECs are given a reference lacking OECs, we argue that this behavior is actually not desirable. In particular, the two cell-types exhibit significant transcriptionally differences. For instance, top differentially upregulated genes in OECs are highly expressed in the olfactory nerve layer (Supplementary Figure S8C) whereas top differentially upregulated genes in N2 cells are not well detected in the olfactory nerve layer (Supplementary Figure S9D). Additionally, when a scRNA-seq reference with OECs was used, reference-based deconvolution approaches generally estimated N2 cells to be relatively rare. However, when a scRNA-seq reference

without OECs was used, reference-based deconvolution approaches substantially increased their estimated abundance of N2 cells (Supplementary Figure S9B).

We have included these results in the revised Results section “STdeconvolve characterizes the spatial organization of transcriptionally distinct cell-types in real ST data” (Lines 246-318) and include relevant text below for the reviewer’s reference.

Next, to simulate a less suitable scRNA-seq reference, we removed OECs from the MOB scRNA-seq reference and again evaluated the performance of reference-based deconvolution approaches given this new scRNA-seq reference without OECs (Supplementary Methods). Again, as a reference-free deconvolution approach, the results of STdeconvolve do not change. However, for some reference-based deconvolution approaches, given this new reference without OECs, pixels in the olfactory nerve layer previously comprised of OECs were now predicted to be comprised of N2 cells (Supplementary Figure S9A). Although we do not know the ground truth cell-type composition of this olfactory nerve layer, we have reasons to believe that this placement of N2 cells is erroneous. First, when a scRNA-seq reference with OECs was used, reference-based deconvolution approaches generally estimated N2 cells to be relatively rare. However, when a scRNA-seq reference without OECs was used, reference-based deconvolution approaches substantially increased their estimated abundance of N2 cells (Supplementary Figure S9B). Second, while the transcriptional profiles of OECs and N2 cells are highly correlated (Supplementary Figure S9C), the two cell-types exhibit significant transcriptional differences. For example, top differentially upregulated genes in OECs are highly expressed in the olfactory nerve layer (Supplementary Figure S8C) whereas top differentially upregulated genes in N2 cells are not well detected in the olfactory nerve layer (Supplementary Figure S9D). This lack of detection of N2 cell marker genes within the olfactory nerve layer coupled with the rarity of N2 cells in the original reference-based deconvolution with OECs suggests that the placement of N2 cells in the olfactory nerve layer by reference-based deconvolution approaches when using a reference without OECs is erroneous.

Supplementary Figure S9. Effect of missing OEC cell cluster in MOB reference on supervised deconvolution approaches. A) Pixels of the MOB ST dataset represented as pie charts indicating the deconvolved pixel

proportions of OEC clusters (blue) and the N2 cell cluster (red), by supervised deconvolution approaches, trained with either the full MOB scRNA-seq reference or the reference missing OEC clusters. B) Deconvolved pixel proportions of N2 cell cluster by either SPOTlight (blue), RCTD (red), or spatialDWLS (green) trained with either the full MOB scRNA-seq reference or the reference missing OEC clusters. Numbers below each boxplot indicate the number of pixels in the MOB ST dataset for which N2 cluster was predicted. C) Pearson's correlations between transcriptional profiles of cell-types in the MOB scRNA-seq reference. D) Gene counts in each pixel of the MOB ST dataset of top putative marker genes for N2 cell cluster.

Minor Comment 4-9: The model performance measured by transcriptional profile correlation in Fig S3B seems rather strange. Why does ground truth immature OD have a relatively large correlation with every deconvolved cell type?

We thank the reviewer for pointing this out. The immature oligodendrocytes are transcriptionally similar to the neuronal cell-types when we correlate the ground truth cell-type transcriptional profiles (Supplementary Figure S2C) and thus correlate transcriptionally with deconvolved cell-types that are representative of neuronal cell-types (Supplementary Figure S2B). While correlated transcriptionally, however, the immature oligodendrocytes are not correlated in terms of cell-type pixel proportions with any of the neuronal deconvolved cell-types (Figure 2D).

Supplementary Figure S2B-C. Comparison of deconvolved cell-types from STdeconvolve to ground truth cell-types of simulated MERFISH MPOA ST data. B) Pearson's correlation between the transcriptional profiles of the 9 ground truth cell-types in the MERFISH MPOA data and the 9 deconvolved cell-types by STdeconvolve. C) Pearson's correlation between the transcriptional profiles of the 9 ground truth cell-types.

Figure 2D. Deconvolution of simulated ST data. D) Ground truth cell-types are ordered by their frequencies in the ground truth dataset. Matched deconvolved and ground truth cell-types are boxed.

Minor Comment 4-10: Does Fig 2C correspond to some aggregation of 9 cell types (if so, how was the aggregation of gene ranks performed) or just a single cell type?

We thank the reviewer for this point of clarification. The reviewer is correct that Fig 2C does correct to an aggregation of all 9 cell-types. In this manner, we had hoped to show a summarized performance across all 9 cell-types in one plot, rather than making 9 independent plots. Briefly, each deconvolved cell-type was matched to its corresponding ground truth cell-type. Then the gene ranks compared of each pair were compared. The shared gene rankings between deconvolved and ground truth for all cell-types are plotted. This has now been clarified in the figure legends and provided below for the reviewer’s reference.

Minor Comment 4-11: The model performance quantified by the pixel cell-type proportion in Fig 2D is not as good as that in Fig 1 of the RCTD paper (Cable et al., 2021). Could you explain why astrocyte and pericyte are mapping to X3 together with ependymal, while Microglia does not have a mapped prediction?

We thank the reviewer for pointing this out. However, we note that in Figure 1 in Cable *et al.*³⁸ demonstrates the challenges of platform effects when training models using one type of data and applying them to another. For example, Figure 1D in Cable *et al.*³⁸ shows the classification performance of a non-negative least squares regression model on snRNA-seq data after being trained using the same data type. Figure 1E in Cable *et al.* shows the classification performance of the same non-negative least squares regression tested on scRNA-seq data after being trained on snRNA-seq data. While Figure 2C in Cable *et al.* does show the classification performance of RCTD on scRNA-seq data after being trained with snRNA-seq data, although this particular heat map is not spatial transcriptomics data.

With respect to STdeconvolve’s own deconvolution results, we can explain the results in the following way. Cell-type X3 was annotated as ependymal cells because its transcriptional profile was highly enriched in ependymal upregulated genes (Methods, “**Annotation and matching of deconvolved and ground truth cell-types**”, Lines 625-642). As such, it is highly correlated with ependymal cells in terms of its transcriptional and cell-type pixel proportional correlation. Some astrocytes are also located in pixels with ependymal cells, which can explain the weak correlation between cell-type X3 and astrocytes. We note that the ground truth astrocytes have a weak positive correlation in terms of their cell-type pixel proportions with several deconvolved cell-types likely for this reason –

they are also present across many multi-cell pixels (Figure 2A and D). With respect to pericytes, this cell-type is also present in some of the pixels with ependymal cells. Pericytes are also very rare in this dataset and so the few that are present in the ependymal pixels may be enough to be positively correlated with cell-type X3.

Figure 2A and D. Deconvolution of simulated ST data. A) Ground truth single-cell resolution MERFISH data of one section of the MPOA partitioned into $100 \mu\text{m}^2$ pixels (black dashed squares). Each dot is a single cell colored by its ground truth cell-type label. D) Ground truth cell-types are ordered by their frequencies in the ground truth dataset. Matched deconvolved and ground truth cell-types are boxed.

Additionally, we note that the curated 135 genes in the MERFISH data that was used to construct the simulated ST MPOA dataset were primarily chosen to identify different neuronal subtypes, with only 33 non-neuronal genes selected. Thus, when deconvolving $K=9$ cell-types, LDA assigned cell-types to split up the neuronal cell-types first because there was more transcriptional variation measured there with the many neuronal genes versus the 33 non-neuronal genes. However, when we expanded the number of deconvolved cell-types to include the number of neuronal subtypes, we deconvolved cell-types that were highly correlated with microglia, as well as the other non-neuronal cell-types, and finer neuronal subtypes. These findings are further described in **Supplementary Note 3** in the Supplementary Materials (Lines 145-179).

Minor Comment 4-12: In order to properly evaluate the correlation structures in Figure 2D, S3B, etc., it would be helpful to display the correlations between true cell types as well.

We thank the reviewer for this suggestion and have now included heatmaps indicating Pearson’s correlations between the ground truth cell-types transcriptional profiles (Supplementary Figure S2C), which we have provided below for the reviewer’s convenience.

Supplementary Figure S2C. Comparison of deconvolved cell-types from STdeconvolve to ground truth cell-types of simulated MERFISH MPOA ST data. C) Pearson's correlation between the transcriptional profiles of the 9 ground truth cell-types.

Minor Comment 4-13: Instead of Fig 3C and Fig S6E, it might be better to show volcano plots with p-values to indicate the DE significance.

We thank the reviewer for the suggestion. Because the differentially expressed genes are based on one cell-type's deconvolved transcriptional profile versus the mean expression of the others, and not for a group of cells, we are unable to estimate the variances around these gene expression distributions and therefore are unable to estimate p-values that would enable a volcano plot visualization.

Minor Comment 4-14: Could you include the "proportion" evaluation of Figure S3C for the two supervised methods, too?

We thank the reviewer for the suggestion. We now quantitate the deconvolution performance of the different deconvolution approaches including the supervised methods by calculating the RMSE of a given approach's predicted cell-type proportions for each pixel compared to the ground truth. In this manner, approaches can then be compared by testing for statistical differences in the resulting RMSE distributions using the Diebold-Mariano test. We believe the new RMSE evaluation will provide a better way to compare performance across deconvolution approaches and have thus opted to remove the previous proportion evaluations.

Minor Comment 4-15: Figure S6D: Does deconvolving the data with $K=5$ reproduce the ground truth clusters?

We thank the reviewer for raising this question. When we deconvolve $K=5$ cell-types, we do recapitulate the ground truth clusters well, as we would expect given that the tissue layers themselves are made up of distinct cell-types. Indeed, when we do deconvolve $K=12$ cell-types, the deconvolved cell-types correlate strongly with specific transcriptional clusters (Supplementary Figure S6D) and further partition the 5-cell layers instead (Figure 4A).

Supplementary Figure S6D. Additional of the MOB. D) Pearson's correlation between the pixel proportions of 12 deconvolved cell-types by STdeconvolve and the MOB pixel transcriptional cluster memberships.

Figure 4A. Deconvolution of ST data of varying resolution from multiple technologies by STdeconvolve. A) Deconvolved cell-type proportions for ST data of the MOB, represented as pie charts for each ST pixel. Pixels are outlined with colors based on the pixel transcriptional cluster assignment corresponding to MOB coarse cell layers.

Minor Comment 4-16: Producing Figure S7 for the MERFISH data as well might help to interpret it.

We thank the reviewer for the comment. Because the MOB does not have a known ground truth reference, the purpose of this previous figure was to demonstrate that different deconvolution approaches were at least consistent in deconvolving cell-types whose pixel proportions corresponded to similar spatial locations. Conversely, because we know the ground truth of the MPOA, we can instead compare the approaches to each other by directly comparing their deconvolved cell-type pixel proportions to the known ground truth cell-type proportions and spatial locations.

Minor Comment 4-17: Fig S13 is difficult to interpret and it makes STdeconvolve look poor in comparison to the other methods. I felt this was somewhat insufficient to explore the implications of deconvolving with closely related cell types.

We agree with the reviewer that this figure was insufficient to assess the deconvolution accuracy of rare cell-types. We have removed it from this revised version of the manuscript.

Minor Comment 4-18: In lines 174-180 of Online Methods describes a method comparison based on RMSE of individual cell types: I may have missed where its results are reported.

We note that this particular calculation of RMSE was in order to normalize the RMSE between cell-types when calculating the RMSE of each cell-type across pixels for the previous Supplemental Figure S13 mentioned above. However, because we no longer do this calculation in this revised version of the manuscript, it has been removed. We still compute the RMSE for each pixel by comparing the predicted proportions to the ground truth proportions across all cell-types, which is in the Methods section “**Comparison of deconvolution approaches**” (lines 650-659) and we provide below for the reviewer’s convenience.

How each supervised and semi-supervised deconvolution approach was run is further detailed in the Supplementary Methods. To compare the performance between deconvolution methods, the root mean squared error (RMSE) was computed for each pixel between the deconvolved and matched ground truth cell-type proportions for each pixel in the ST dataset:

$$RMSE = \sqrt{\frac{\sum_{k=1}^K (\hat{y}_k - y_k)^2}{K}}$$

where K is the number of cell-types, \hat{y}_k is the predicted cell-type proportion for the cell-type k , and y_k is the ground truth cell-type proportion for the cell-type k . To assess whether the distribution of pixel RMSEs was significantly lower for STdeconvolve compared to other methods, a one-sided Diebold-Mariano Test²⁷ was used.

We have also now provided a table of contents in the Supplementary/Online Methods that we hope will help direct readers to the appropriate methods section.

References

- 1 Levy-Jurgenson, A., Tekpli, X., Kristensen, V. N. & Yakhini, Z. Spatial transcriptomics inferred from pathology whole-slide images links tumor heterogeneity to survival in breast and lung cancer. *Sci Rep* **10**, 18802, doi:10.1038/s41598-020-75708-z (2020).
- 2 Rodrigues, S. G. *et al.* Slide-seq: A scalable technology for measuring genome-wide expression at high spatial resolution. *Science* **363**, 1463-1467, doi:10.1126/science.aaw1219 (2019).
- 3 Liu, Y. *et al.* High-Spatial-Resolution Multi-Omics Sequencing via Deterministic Barcoding in Tissue. *Cell* **183**, 1665-1681 e1618, doi:10.1016/j.cell.2020.10.026 (2020).
- 4 <https://www.10xgenomics.com/resources/datasets>.
- 5 Lein, E. S. *et al.* Genome-wide atlas of gene expression in the adult mouse brain. *Nature* **445**, 168-176, doi:10.1038/nature05453 (2007).
- 6 Saunders, A. *et al.* Molecular Diversity and Specializations among the Cells of the Adult Mouse Brain. *Cell* **174**, 1015-1030 e1016, doi:10.1016/j.cell.2018.07.028 (2018).
- 7 Dong, R. & Yuan, G. C. SpatialDWLS: accurate deconvolution of spatial transcriptomic data. *Genome Biol* **22**, 145, doi:10.1186/s13059-021-02362-7 (2021).
- 8 Moffitt, J. R. *et al.* Molecular, spatial, and functional single-cell profiling of the hypothalamic preoptic region. *Science* **362**, doi:10.1126/science.aau5324 (2018).
- 9 Wang, C. *et al.* Identification and characterization of neuroblasts in the subventricular zone and rostral migratory stream of the adult human brain. *Cell Res* **21**, 1534-1550, doi:10.1038/cr.2011.83 (2011).
- 10 Hintiryan, H. *et al.* Comprehensive connectivity of the mouse main olfactory bulb: analysis and online digital atlas. *Front Neuroanat* **6**, 30, doi:10.3389/fnana.2012.00030 (2012).
- 11 Jian, T., Zhaoshi, M., Xuanlong, N., Qiaozhu, M. & Ming, Z. 190-198 (PMLR, 2014).
- 12 Tasic, B. *et al.* Adult mouse cortical cell taxonomy revealed by single cell transcriptomics. *Nat Neurosci* **19**, 335-346, doi:10.1038/nn.4216 (2016).
- 13 Vizgen Data Release V1.0. May 2021.
- 14 Zhao, E. *et al.* Spatial transcriptomics at subspot resolution with BayesSpace. *Nat Biotechnol*, doi:10.1038/s41587-021-00935-2 (2021).
- 15 Eng, C. L. *et al.* Transcriptome-scale super-resolved imaging in tissues by RNA seqFISH. *Nature* **568**, 235-239, doi:10.1038/s41586-019-1049-y (2019).
- 16 Xia, C., Fan, J., Emanuel, G., Hao, J. & Zhuang, X. Spatial transcriptome profiling by MERFISH reveals subcellular RNA compartmentalization and cell cycle-dependent gene expression. *Proc Natl Acad Sci U S A* **116**, 19490-19499, doi:10.1073/pnas.1912459116 (2019).
- 17 Wang, X. *et al.* Three-dimensional intact-tissue sequencing of single-cell transcriptional states. *Science* **361**, doi:10.1126/science.aat5691 (2018).
- 18 Wang, F. *et al.* RNAscope: a novel in situ RNA analysis platform for formalin-fixed, paraffin-embedded tissues. *J Mol Diagn* **14**, 22-29, doi:10.1016/j.jmoldx.2011.08.002 (2012).
- 19 Codeluppi, S. *et al.* Spatial organization of the somatosensory cortex revealed by osmFISH. *Nat Methods* **15**, 932-935, doi:10.1038/s41592-018-0175-z (2018).
- 20 Kim, M. *et al.* Immune microenvironment in ductal carcinoma in situ: a comparison with invasive carcinoma of the breast. *Breast Cancer Res* **22**, 32, doi:10.1186/s13058-020-01267-w (2020).
- 21 Beguinot, M. *et al.* Analysis of tumour-infiltrating lymphocytes reveals two new biologically different subgroups of breast ductal carcinoma in situ. *BMC Cancer* **18**, 129, doi:10.1186/s12885-018-4013-6 (2018).
- 22 Yoosuf, N., Navarro, J. F., Salmen, F., Stahl, P. L. & Daub, C. O. Identification and transfer of spatial transcriptomics signatures for cancer diagnosis. *Breast Cancer Res* **22**, 6, doi:10.1186/s13058-019-1242-9 (2020).
- 23 Fan, J. *et al.* Characterizing transcriptional heterogeneity through pathway and gene set overdispersion analysis. *Nat Methods* **13**, 241-244, doi:10.1038/nmeth.3734 (2016).
- 24 Li, C. M. *et al.* Aging-Associated Alterations in Mammary Epithelia and Stroma Revealed by Single-Cell RNA Sequencing. *Cell Rep* **33**, 108566, doi:10.1016/j.celrep.2020.108566 (2020).
- 25 Subramanian, A. *et al.* Gene set enrichment analysis: a knowledge-based approach for interpreting genome-wide expression profiles. *Proc Natl Acad Sci U S A* **102**, 15545-15550, doi:10.1073/pnas.0506580102 (2005).
- 26 Fan, J. Differential Pathway Analysis. *Methods Mol Biol* **1935**, 97-114, doi:10.1007/978-1-4939-9057-3_7 (2019).

- 27 Diebold, F. X. & Mariano, R. S. Comparing Predictive Accuracy. *Journal of Business & Economic Statistics* **13**, 253-263, doi:10.1080/07350015.1995.10524599 (1995).
- 28 Dey, K. K., Hsiao, C. J. & Stephens, M. Visualizing the structure of RNA-seq expression data using grade of membership models. *PLoS Genet* **13**, e1006599, doi:10.1371/journal.pgen.1006599 (2017).
- 29 Blondel, V. D., Guillaume, J.-L., Lambiotte, R. & Lefebvre, E. Fast unfolding of communities in large networks. *Journal of Statistical Mechanics: Theory and Experiment* **2008**, P10008, doi:10.1088/1742-5468/2008/10/p10008 (2008).
- 30 Kang, K. *et al.* CDSeq: A novel complete deconvolution method for dissecting heterogeneous samples using gene expression data. *PLoS Comput Biol* **15**, e1007510, doi:10.1371/journal.pcbi.1007510 (2019).
- 31 Hu, B. C. The human body at cellular resolution: the NIH Human Biomolecular Atlas Program. *Nature* **574**, 187-192, doi:10.1038/s41586-019-1629-x (2019).
- 32 Regev, A. *et al.* The Human Cell Atlas. *bioRxiv*, 121202, doi:10.1101/121202 (2017).
- 33 Network, B. I. C. C. A multimodal cell census and atlas of the mammalian primary motor cortex. *Nature* **598**, 86-102, doi:10.1038/s41586-021-03950-0 (2021).
- 34 Hawrylycz, M. J. *et al.* An anatomically comprehensive atlas of the adult human brain transcriptome. *Nature* **489**, 391-399, doi:10.1038/nature11405 (2012).
- 35 Gao, R. *et al.* Delineating copy number and clonal substructure in human tumors from single-cell transcriptomes. *Nat Biotechnol* **39**, 599-608, doi:10.1038/s41587-020-00795-2 (2021).
- 36 Zeng, J. *et al.* CancerSCEM: a database of single-cell expression map across various human cancers. *Nucleic Acids Res*, doi:10.1093/nar/gkab905 (2021).
- 37 Asp, M., Bergenstrahle, J. & Lundberg, J. Spatially Resolved Transcriptomes-Next Generation Tools for Tissue Exploration. *Bioessays* **42**, e1900221, doi:10.1002/bies.201900221 (2020).
- 38 Cable, D. M. *et al.* Robust decomposition of cell type mixtures in spatial transcriptomics. *Nat Biotechnol*, doi:10.1038/s41587-021-00830-w (2021).

Reviewers' Comments:

Reviewer #1:

Remarks to the Author:

The additional validation against existing datasets, such as 10x Visium that was compared to the Allen Brain Atlas mapping as well as the validation against the mouse cerebellum maps that were matched against a scRNA-seq reference, provide additional evidence in support of STdeconvolve. Along with the other evaluation methods (both existing and new from the rebuttal), these results strengthen the reader's confidence in that the method, due to its performance, may remain useful for the foreseeable future as a less expensive and scalable alternative to spatial single-cell technology, which is still in its infancy. Therefore - despite the fact that technology will advance towards single cell we do see value in the proposed approach, in light of its performance as demonstrated after the review.

In addition, the authors have addressed all of the other points to our full satisfaction, including scalability concerns. Their new discussion on the balance between resolution and throughput and STdeconvolve's accuracy as a function of resolution adds a lot to the manuscript.

Reviewer #2:

Remarks to the Author:

The authors have thoroughly addressed the reviewers' comments and included several additional analyses based on both simulated as well as real data sets in the manuscript. In addition to the previous points, the manuscript now includes a more thorough bench marking of STdeconvolve compared to related methods that rely on single-cell references as well as spatial subspot clustering-based methods. The authors now also included analyses of run-time and memory usage relative to the number of pixels and cell types. Further, the reviewers' questions regarding the relevance and performance of STdeconvolve in light of improving ST methods with smaller spot sizes were addressed both by analyses on several previously published data sets with smaller spot sizes and in the discussion. Lastly, the authors improved and clarified the analysis of the clinical breast cancer tissues and now added simulations to highlight the ability of STdeconvolve to distinguish immune excluded from immune infiltrated tumor microenvironments. The authors have made great efforts to include all of the reviewers' points into the manuscript.

There are only very few minor comments:

1. Do the authors have an idea why DSTG performs so badly? It seems not to work at all. Is that because some assumptions are violated?
2. After the comparison to BayesSpace the authors conclude: "Taken together, deconvolution approaches such as STdeconvolve can provide distinct results from clustering and resolution enhanced clustering approaches when applied to multi-cellular pixel resolution data." The wording of "distinct results" is a bit confusing. It sounds like the methods yield distinct results when trying to achieve the same thing, whereas I think the authors mean to say that these methods provide different types of information. Is that correct? It would help to state in one sentence what these different types of information are.
3. As a side note: The title is a bit clunky now. It would be easier to read if you changed it to something like "Reference-free cell type deconvolution of multi-cellular pixels in spatially resolved transcriptomic data". But this is up to the authors and editors to decide.

Reviewer #3:

Remarks to the Author:

The authors have addressed all my concerns. There are several approaches available, but the one presented in this paper is unique and, more importantly, reference-free. All the codes in the GitHub link are functional, so the analysis performed in this paper is reproducible. I recommend this paper for publication in Nature Communication.

Reviewer #4:

Remarks to the Author:

Overall the authors have done a thorough job of addressing the comments from the previous round of review. In particular, they have carried out additional benchmarking studies, including (1) applying STdeconvolve to datasets sequenced at various resolutions by different spatial transcriptomic techniques, especially whole-transcriptome spatial data, (2) comparing STdeconvolve to other computational methods that identify cell types in spatial data, and (3) assessing the runtime and memory usage.

The author's response to Major Comment 4-4 is not satisfactory, however. The correlation of data at nearby pixels can make LDA unsuitable for modeling spatial transcriptomic data and the problem should become exacerbated as the strength of correlation increases. The author should have investigated this issue, e.g., by doing simulations.

Lastly, since the spatial structure of ST data is not taken into account in STdeconvolve, the proposed method does not seem all that different from other unsupervised, reference-free joint deconvolution methods.

Point by Point Response – Overview

We sincerely thank the editor and the reviewers for their insightful and constructive feedback in helping us improve this manuscript. We have now revised the manuscript to address all the points raised by the reviewers, organized herein as a point-by-point response. Throughout this point-by-point response, reviewer comments are shown in **blue**, with our responses in **green**, and changes to the manuscript in **black**.

Reviewer 1

The additional validation against existing datasets, such as 10x Visium that was compared to the Allen Brain Atlas mapping as well as the validation against the mouse cerebellum maps that were matched against a scRNA-seq reference, provide additional evidence in support of STdeconvolve. Along with the other evaluation methods (both existing and new from the rebuttal), these results strengthen the reader's confidence in that the method, due to its performance, may remain useful for the foreseeable future as a less expensive and scalable alternative to spatial single-cell technology, which is still in its infancy. Therefore - despite the fact that technology will advance towards single cell we do see value in the proposed approach, in light of its performance as demonstrated after the review.

In addition, the authors have addressed all of the other points to our full satisfaction, including scalability concerns. Their new discussion on the balance between resolution and throughput and STdeconvolve's accuracy as a function of resolution adds a lot to the manuscript.

We wholeheartedly thank the reviewer for taking the time to review this manuscript. Their helpful comments and suggestions have greatly improved the findings and we are pleased that the reviewer is satisfied with the revisions.

Reviewer 2

The authors have thoroughly addressed the reviewers' comments and included several additional analyses based on both simulated as well as real data sets in the manuscript. In addition to the previous points, the manuscript now includes a more thorough bench marking of STdeconvolve compared to related methods that rely on single-cell references as well as spatial subspot clustering-based methods. The authors now also included analyses of run-time and memory usage relative to the number of pixels and cell types. Further, the reviewers' questions regarding the relevance and performance of STdeconvolve in light of improving ST methods with smaller spot sizes were addressed both by analyses on several previously published data sets with smaller spot sizes and in the discussion. Lastly, the authors improved and clarified the analysis of the clinical breast cancer tissues and now added simulations to highlight the ability of STdeconvolve to distinguish immune excluded from immune infiltrated tumor microenvironments. The authors have made great efforts to include all of the reviewers' points into the manuscript.

We wholeheartedly thank the reviewer for taking the time to review this manuscript. Their helpful comments and suggestions have greatly improved the findings and we are pleased that the reviewer is satisfied with the revisions.

There are only very few minor comments:

1. Do the authors have an idea why DSTG performs so badly? It seems not to work at all. Is that because some assumptions are violated?

We thank the reviewer for raising this point. We were motivated by this comment to reach out to the authors and read through the DSTG source code to try and understand why it was not working and why we could not produce any reasonable deconvolution results. In reading through the DSTG source code, we were able to identify certain bugs and believe these bugs directly led to the poor performance by DSTG we saw previously. These bugs have since been communicated to the authors of DSTG.

After addressing these bugs, we were able to obtain more reasonable deconvolution results than previously when deconvolving the MPOA MERFISH simulated ST data using this fixed version of DSTG. Our conclusions remain unchanged considering this new result from the fixed version of DSTG in that, like other reference-based methods, DSTG's performance exhibits similar reference-dependent variation as expected (Reviewer Figure 1).

We therefore anticipate that this fixed version of DSTG will also perform similarly to other reference-based deconvolution approaches when applied to our other benchmark evaluations. However, given the difficulties we experienced in running the DSTG code, we did not feel confident in continuing to evaluate the performance of DSTG. We have therefore opted to remove DSTG altogether from the revised manuscript.

Reviewer Figure 1. Deconvolution of simulated ST data with DSTG. A) Left: Deconvolved pixel proportions of cell-types by DSTG using the MERFISH MPOA single-cell transcriptomics data as the reference. Right: Deconvolved pixel proportions of cell-types by DSTG using the MERFISH MPOA single-cell transcriptomics data with missing neurons as the reference. B) Root-mean-square-error (RMSE) of the deconvolved cell-type proportions compared to ground truth for STdeconvolve, C) for supervised and semi-supervised (i.e., DSTG) deconvolution approaches using the ideal single cell transcriptomics MERFISH MPOA reference, D) for supervised and semi-supervised (i.e., DSTG) deconvolution approaches using the single cell transcriptomics MERFISH MPOA reference with missing neurons, and E) for supervised and semi-supervised (i.e., DSTG) deconvolution approaches using a brain single-cell RNA-seq reference.

2. After the comparison to BayesSpace the authors conclude: “Taken together, deconvolution approaches such as STdeconvolve can provide distinct results from clustering and resolution enhanced clustering approaches when applied to multi-cellular pixel resolution data.” The wording of “distinct results” is a bit confusing. It sounds like the methods yield distinct results when trying to achieve the same thing, whereas I think the authors mean to say that these methods provide different types of information. Is that correct? It would help to state in one sentence what these different types of information are.

We thank the reviewer for bringing up this point of clarification. The reviewer is correct; we are trying to suggest these methods provide different types of information. Specifically, while clustering analysis may identify pixels that are transcriptionally similar, in a multi-cellular pixel setting where pixels contain mixtures of multiple cell-types, transcriptional similarity will also be impacted by the underlying distribution of cell-type proportions. As we demonstrated via simulation, in a multi-cellular pixel setting, pixels that contains cell-types A and B or B and C may be assigned to 2 different clusters via clustering analysis, however, deconvolution could show that there are in fact 3 different cell-type populations mixed at different proportions. To clarify this point, we have made the following changes to the concluding sentence of “**Deconvolution provides distinct insights compared to clustering analysis**” (lines 240 – 243), which we have included below for the reviewer’s convenience.

Taken together, deconvolution approaches such as STdeconvolve can reveal cell-types and patterns not evident through clustering and resolution enhanced clustering approaches alone when applied to multi-cellular pixel resolution data for pixels containing heterogeneous mixtures of cell-types.

3. As a side note: The title is a bit clunky now. It would be easier to read if you changed it to something like “Reference-free cell type deconvolution of multi-cellular pixels in spatially resolved transcriptomic data”. But this is up to the authors and editors to decide.

We thank the reviewer for the suggestion. Given that there is a variety of spatial transcriptomic technologies that measure gene expression at different resolutions, we have opted to keep the current title in order specifically and clearly indicate the type of spatially resolved transcriptomics data for which STdeconvolve is best suited, i.e. pixels, or regions of measured gene expression that contain more than one cell or “multi-cellular pixel-resolution spatially resolved transcriptomics data”

Reviewer 3

The authors have addressed all my concerns. There are several approaches available, but the one presented in this paper is unique and, more importantly, reference-free. All the codes in the GitHub link are functional, so the analysis performed in this paper is reproducible. I recommend this paper for publication in Nature Communication.

We wholeheartedly thank the reviewer for taking the time to review this manuscript. Their helpful comments and suggestions have greatly improved the findings and we are pleased that the reviewer is satisfied with the revisions.

Reviewer 4

Overall the authors have done a thorough job of addressing the comments from the previous round of review. In particular, they have carried out additional benchmarking studies, including (1) applying STdeconvolve to datasets sequenced at various resolutions by different spatial transcriptomic techniques, especially whole-transcriptome

spatial data, (2) comparing STdeconvolve to other computational methods that identify cell types in spatial data, and (3) assessing the runtime and memory usage.

We wholeheartedly thank the reviewer for taking the time to review this manuscript. Their helpful comments and suggestions have greatly improved the findings and we are pleased that the reviewer is satisfied with the revisions made thus far.

The author's response to Major Comment 4-4 is not satisfactory, however. The correlation of data at nearby pixels can make LDA unsuitable for modeling spatial transcriptomic data and the problem should become exacerbated as the strength of correlation increases. The author should have investigated this issue, e.g., by doing simulations.

We appreciate the reviewer's concern about how the correlation between nearby pixels may affect deconvolution accuracy.

As the reviewer hints, potential lateral diffusion of mRNAs originating in one pixel into adjacent pixels before mRNA capture and measurement could result in an artificially increased transcriptional correlation between nearby pixels that can make LDA unsuitable. In our application of LDA to ST data, we are inherently assuming that the gene expression levels we observe at each ST pixel are from the cells of that pixel, rather than from any lateral diffusion of genes from adjacent pixels. We believe this is a reasonable assumption because current pixel-resolution spatial transcriptomics technologies have sought to characterize this phenomenon and have reported it to occur at very low levels¹. However, we acknowledge future spatially resolved transcriptomics technologies may exhibit a greater degree of lateral diffusion and as such a greater degree of correlation of data at nearby pixels that can make LDA unsuitable. We have added a few sentences clarifying our assumptions to Supplemental Note 1 (**Suitability of ST data for LDA**, lines 123-126):

Finally, we assume ST data exhibit minimal lateral diffusion such that gene-level counts observed at each spatially resolved pixel come from the cells present within that pixel rather than through diffusion from nearby pixels^{1,2}. We believe this to be a reasonable assumption for ST data from current ST technologies.

Regardless of the possible technical issues of different ST technologies, we reiterate that the implementation of LDA in STdeconvolve does not consider the spatial relationship or correlation between pixels, and moreover, the positions of the pixels are not required for deconvolution by STdeconvolve but merely only for subsequent visualization purposes. While the dataset of pixels and the gene counts within them will determine the deconvolved cell-types, the defined pixels themselves can be theoretically positioned anywhere in space. Because cell-types are typically organized into higher order structures in tissues, we would expect adjacent or nearby pixels to be correlated in terms of their cell-type composition, but this is not required for STdeconvolve to accurately recover them.

To demonstrate this, we used a simulation by taking the MOB ST dataset and randomly shuffled the positions of the pixels to create an ST dataset where there is no spatial relationship or correlation between pixels. We then reapplied STdeconvolve and found that the results are identical (Reviewer Figure 1). In this way, the positions of pixels and their spatial relationship to one another does not affect the deconvolution accuracy of STdeconvolve.

Reviewer Figure 1. Deconvolution of MOB ST dataset by STdeconvolve before and after shuffling pixel positions. A) Left: scatter pie charts of the MOB ST dataset pixels indicating the transcriptional cluster membership of each pixel (top) and the deconvolved cell-type pixel proportions (bottom). Right: same as left panel but after randomly shuffling pixels. B) Correlation heatmaps of pixel cell-type proportions and transcriptional cluster assignments. There is no difference regardless of where pixels are initially located before deconvolution.

Lastly, since the spatial structure of ST data is not taken into account in STdeconvolve, the proposed method does not seem all that different from other unsupervised, reference-free joint deconvolution methods.

Although we do not take into account the spatial structure, STdeconvolve is distinct from previously published reference-free bulk deconvolution methods in several ways.

First, STdeconvolve includes several data driven approaches for identification of the optimal number of cell-types to deconvolve, which are not included in previously published reference-free bulk deconvolution approaches. Additionally, we also implement a data-driven feature selection approach to improve LDA not used by previously published reference-free bulk deconvolution approaches.

Second, to our knowledge, STdeconvolve is the first reference-free deconvolution approach that has been applied to ST data. Importantly, previous reference-free deconvolution methods were developed for bulk samples (microarray^{3,4}, mixing of two cellular compositions (tumor-stroma⁴) or bulk RNA-seq⁵, for example), in which there are several inherently different assumptions made. For example, one notable difference is that ST data is UMI-based whereas bulk RNA-seq quantifications are biased by gene length. Because of this gene length bias, previous LDA-based deconvolution approaches such as CDSeq have included corrections in their models to accommodate these differences in effective gene length on the observed gene expression measurements⁵. As such, when we apply CDSeq to our simulated ST data, we observe a poor correlation between the deconvolved gene expression profiles and the ground truth gene expression profiles as we showed previously. Additionally, there are sample size

differences between ST data and bulk RNA-seq deconvolution tasks, in which ST datasets can have considerably more pixels to deconvolve in contrast to collections of bulk RNA-seq data. Understandably, bulk RNA-seq analysis approaches did not face such large sample size challenges. As such, applying CDSeg to deconvolve our simulated ST data with 900 pixels (a typical size for ST data) took over 2.5 hours compared to approximately 30 seconds by STdeconvolve as we showed previously.

While STdeconvolve is the first case of a reference-free deconvolution approach applied to ST data, we recognize it is not the first application of LDA as a reference-free deconvolution approach to other types of high-throughput transcriptomic profiling data. As such, we have included additional citations referencing these previous applications of LDA approaches to bulk RNA-seq data noted by the reviewer, in which we have included the relevant text below for the reviewer's convenience (**Introduction**, lines 74-77).

While LDA has previously been applied in the context of deconvolving cell-types in bulk RNA-seq data^{5,6}, STdeconvolve leverages several unique features of ST data that make this application of LDA particularly amenable (Supplementary Note 1).

References

- 1 Liu, Y. *et al.* High-Spatial-Resolution Multi-Omics Sequencing via Deterministic Barcoding in Tissue. *Cell* **183**, 1665-1681 e1618, doi:10.1016/j.cell.2020.10.026 (2020).
- 2 Stahl, P. L. *et al.* Visualization and analysis of gene expression in tissue sections by spatial transcriptomics. *Science* **353**, 78-82, doi:10.1126/science.aaf2403 (2016).
- 3 Li, Z. & Wu, H. TOAST: improving reference-free cell composition estimation by cross-cell type differential analysis. *Genome Biol* **20**, 190, doi:10.1186/s13059-019-1778-0 (2019).
- 4 Wang, N. *et al.* UNDO: a Bioconductor R package for unsupervised deconvolution of mixed gene expressions in tumor samples. *Bioinformatics* **31**, 137-139, doi:10.1093/bioinformatics/btu607 (2015).
- 5 Kang, K. *et al.* CDSeq: A novel complete deconvolution method for dissecting heterogeneous samples using gene expression data. *PLoS Comput Biol* **15**, e1007510, doi:10.1371/journal.pcbi.1007510 (2019).
- 6 Dey, K. K., Hsiao, C. J. & Stephens, M. Visualizing the structure of RNA-seq expression data using grade of membership models. *PLoS Genet* **13**, e1006599, doi:10.1371/journal.pgen.1006599 (2017).